# Interplay of two small RNAs fine-tunes hierarchical flagella gene expression in *Campylobacter jejuni*

Fabian König [1], Sarah L. Svensson [1,2] & Cynthia M. Sharma [1] ✉

Like for many bacteria, flagella are crucial for *Campylobacter jejuni* motility and virulence. Biogenesis of the flagellar machinery requires hierarchical transcription of early, middle (RpoN-dependent), and late (FliA-dependent) genes. However, little is known about post-transcriptional regulation of flagellar biogenesis by small RNAs (sRNAs). Here, we characterized two sRNAs with opposing effects on *C. jejuni* filament assembly and motility. We demonstrate that CJnc230 sRNA (FlmE), encoded downstream of the flagellar hook protein, is processed from the RpoN-dependent *flgE* mRNA by RNase III, RNase Y, and PNPase. We identify mRNAs encoding a flagella-interaction regulator and the anti-sigma factor FlgM as direct targets of CJnc230 repression. CJnc230 overexpression upregulates late genes, including the flagellin *flaA*, culminating in longer flagella and increased motility. In contrast, overexpression of the FliA-dependent sRNA CJnc170 (FlmR) reduces flagellar length and motility. Overall, our study demonstrates how the interplay of two sRNAs post-transcriptionally fine-tunes flagellar biogenesis through balancing of the hierarchically-expressed components.

Flagella are crucial for colonization of the human gut by many bacterial pathogens. They mediate not only chemotaxis towards the epithelial lining, but also adhesion to host cells, immunogenicity, and even secretion of virulence factors[1–4]. The biogenesis of these complex, multi-component molecular machines follows a well characterized cascade of hierarchical transcriptional control to mediate assembly of the flagellum from the inside out in gammaproteobacterial model organisms such as *Escherichia coli* and *Salmonella enterica*[5]. In *E. coli* and *Salmonella*, early (class I) flagellar genes include the master transcriptional regulator FlhDC, which activates middle (class II) genes, mainly comprising hook-basal-body (HBB) complex components, as well as the sigma factor $\sigma^{28}$ (FliA) and its antagonist FlgM[6]. Upon completion of the HBB, FlgM is secreted, thereby releasing FliA for transcriptional activation of late flagellar genes (class III), including the filament component flagellin[5]. Despite flagella being a paradigm for studying hierarchical gene expression in bacteria, their regulation

varies among different species and little is known about post-transcriptional control in this complex process.

The Epsilonproteobacterium *Campylobacter jejuni* is currently the most prevalent cause of bacterial foodborne illness worldwide[7]. While its small ~1.6 megabase pair (Mbp) genome lacks classical virulence factors encoded by enteric pathogens such as dedicated type III secretion systems[8], *C. jejuni* secretes virulence-associated factors through its flagellar type III secretion system (fT3SS)[9]. Moreover, flagella-mediated motility is also a crucial *C. jejuni* virulence determinant[10–12], and the flagellum mediates host cell adhesion[13]. In contrast to *E. coli* or *Salmonella*, Epsilonproteobacteria do not encode a homolog of the master regulator FlhDC, and early genes in the hierarchical transcriptional cascade are expressed constitutively with the help of the housekeeping sigma factor RpoD ($\sigma^{70}$ in *C. jejuni*)[14]. Moreover, Epsilonproteobacteria have dedicated their sole additional sigma factors (RpoN and FliA) to drive expression of middle and late

[1]University of Würzburg, Institute of Molecular Infection Biology, Department of Molecular Infection Biology II, 97080 Würzburg, Germany. [2]The Center for Microbes, Development and Health, CAS Key Laboratory of Molecular Virology and Immunology, Shanghai Institute of Immunity and Infection, Chinese Academy of Sciences, Shanghai 200031, China. ✉e-mail: cynthia.sharma@uni-wuerzburg.de

flagellar genes (Fig. 1a)[15]. In *C. jejuni*, early genes encode components of the fT3SS, the cytoplasmic and inner membrane rings, as well as activators of class II gene expression, including the two-component system factors FlgS (histidine kinase) and FlgR ($\sigma^{54}$-associated transcriptional activator), RpoN ($\sigma^{54}$), and the FlhF GTPase[16–18]. RpoN promotes transcription of class II genes including rod and hook

components and the minor flagellin FlaB. In addition, RpoN activates transcription of FliA ($\sigma^{28}$) and FlgM[15] (Fig. 1a). Similar to Gammaproteobacteria, *H. pylori* and *C. jejuni* use a FlgM-FliA checkpoint inhibition strategy between expression of class II and class III genes to guarantee proper flagellar assembly[19,20]. Late flagellar gene transcription is promoted by FliA and includes the major flagellin FlaA and other

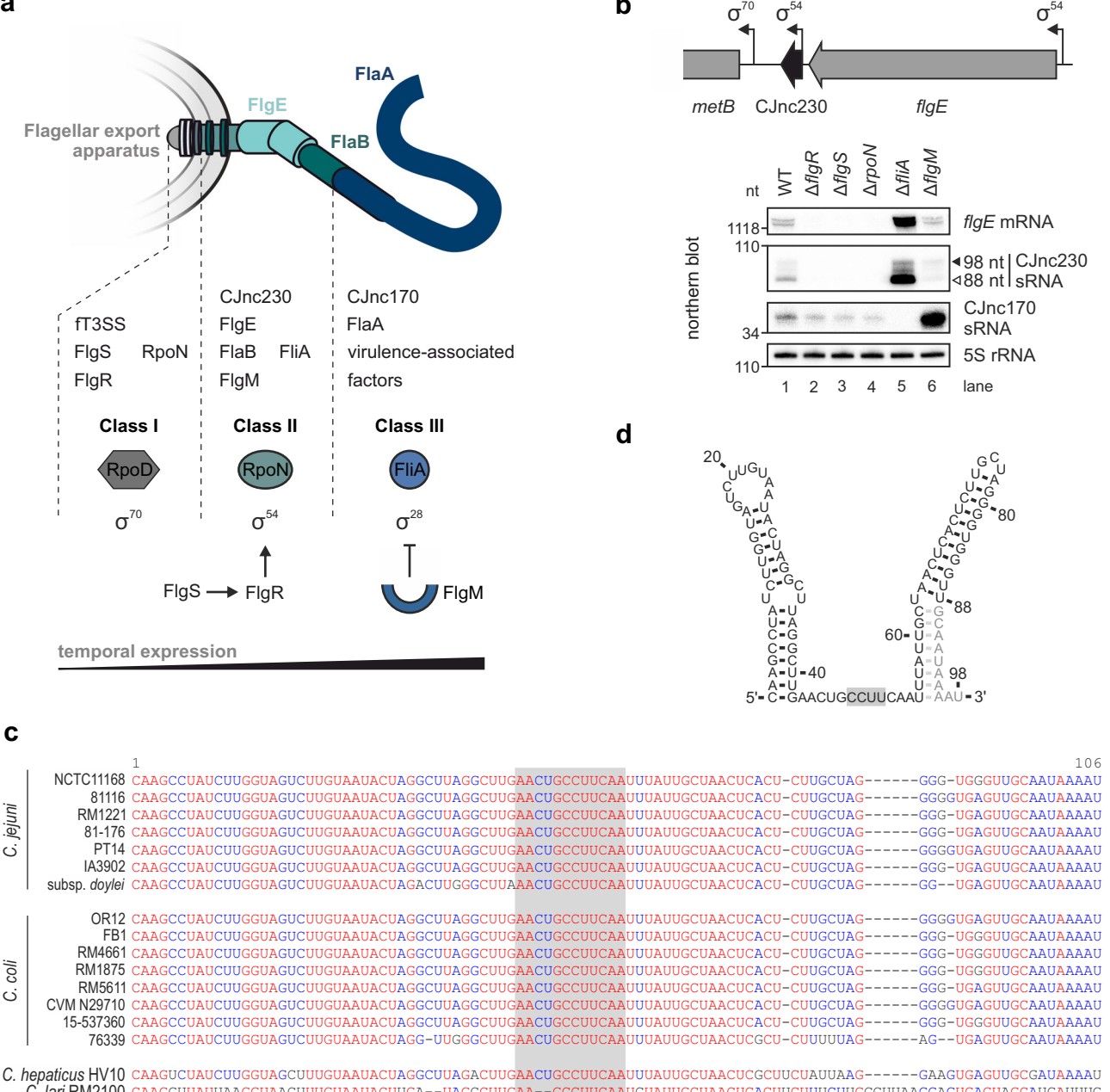

**Fig. 1 | CJnc230 sRNA is transcribed in an RpoN-dependent manner and conserved in diverse *Campylobacter* species. a** *C. jejuni* flagellum structure with flagellar export apparatus, hook (FlgE), filament components (FlaB & FlaA), and CJnc230/CJnc170 sRNAs. Below: temporal expression control by three sigma factors (RpoD, RpoN, FliA) of three classes of flagellar genes. Only proteins and sRNAs mentioned in the text are indicated. fT3SS: flagellar type three secretion system. **c** Alignment of corresponding 98-nt CJnc230 sequences from multiple *Campylobacter* species and strains using MultAlin[122]. Gray box: single-stranded RNA region (see panel **d**).
**b** (*Upper*) Genomic location of CJnc230 (black arrow) in *C. jejuni* strain NCTC11168 downstream of the RpoN ($\sigma^{54}$)-dependent *flgE* (Cj1729c) gene. Bent arrows: transcriptional start sites (TSSs) with associated promoter motifs[28,29]. (*Lower*) Northern blot analysis of total RNA from *C. jejuni* wildtype (WT) and flagella regulator deletion mutants at exponential growth phase. Black triangle: 98-nt previously

annotated full-length sRNA[28]. White triangle: 3'-truncated abundant 88-nt version identified in this study. The *flgE* mRNA was detected with oligonucleotide CSO-5136, CJnc230 sRNA was probed with CSO-0537, CJnc170 sRNA with CSO-0182, and 5S rRNA (CSO-0192) was detected as a loading control. **c** Alignment of corresponding 98-nt CJnc230 sequences from multiple *Campylobacter* species and strains using MultAlin[122]. Gray box: single-stranded RNA region (see panel **d**).
**d** Secondary structure of full-length CJnc230 (98 nt) predicted by RNAfold[39]. Black residues: 3'-truncated 88-nt version. Gray box: putative anti-Shine-Dalgarno motif.
Data in (**b**) are representative results of at least three independent experiments. Source data are provided as a Source Data File.

virulence-associated factors, such as flagellar co-expressed determinant (Fed) proteins[9,21].

In contrast to the well studied hierarchical transcriptional control of flagellar biogenesis in diverse bacterial species, much less is known about the role of post-transcriptional regulation via small regulatory RNAs (sRNAs) within this cascade. These riboregulators mostly act via base-pairing interactions with target mRNAs and play important roles in bacterial adaptation to diverse environmental stresses and fine-tuning of multiple physiological processes, including virulence[22–24]. So far, only a few gammaproteobacterial sRNAs are known to modulate flagellar biogenesis, mainly targeting the mRNA encoding FlhDC[25–27] and even less is known about the role of sRNAs in the flagellar bio-synthesis cascade outside of *E. coli* and *Salmonella*.

Several sRNA candidates identified based on global transcriptome analyses of *C. jejuni* have conserved motifs for flagellar sigma factors in their promoters[28–30], indicating they could be co-expressed and play a role in flagellar assembly. Here, we demonstrate that two distinct sRNAs are co-regulated with middle and late flagellar genes and post-transcriptionally fine-tune hierarchical control of flagellar gene expression in *C. jejuni*. CJnc230 sRNA is co-transcribed with the flagellar hook structural protein FlgE and processed by three ribonucleases (RNases). CJnc230 directly represses mRNAs encoding Cj1387c, a putative regulator of flagella-flagella interactions (CfiR)[31], as well as the anti-sigma factor FlgM. Overexpression of CJnc230 increases flagellar length and motility in part via translational inhibition of *flgM*, thereby de-repressing FliA-dependent late flagellar genes, including the major flagellin FlaA. While CJnc230 transcription is dependent on RpoN, we show FlgM repression also induces the FliA-dependent sRNA CJnc170, which has an opposite effect on filament length and swimming. Based on these observations, we suggest renaming CJnc230 and CJnc170 to FlmE and FlmR (flagellar length and motility enhancer/repressor), respectively. Due to their impact on the assembly of the flagellar filament and motility, they might ultimately also affect pathogenesis, as motility is crucial for *C. jejuni* host colonization[9].

## Results

### RpoN is required for expression of the conserved *C. jejuni* sRNA CJnc230

While RNA-seq-based transcriptome analysis revealed diverse sRNA candidates in *C. jejuni*[28,29,32], their functions and cellular targets remain largely unknown. To explore sRNAs that might regulate key *C. jejuni* phenotypes such as motility, we examined expression differences of several potential flagella co-regulated sRNAs in deletion mutants of characterized flagella regulatory genes by northern blotting. sRNAs were selected based on presence of flagellar sigma factor promoter motifs or close vicinity to flagella structural genes. This confirmed a previously suggested dependency of CJnc170 sRNA on FliA[30] and revealed that expression of CJnc230 sRNA, which is encoded downstream of the gene encoding the flagellar hook protein FlgE (Cj1729c)[28,29], is strongly affected by deletion of several regulators involved in flagellar biosynthesis (Fig. 1b). CJnc230 expression was completely lost when the gene encoding RpoN or components of its upstream activating FlgRS two-component system were deleted and increased in a FliA mutant (Fig. 1b). Our previous comparative differential RNA-seq (dRNA-seq) identified CJnc230 as a conserved ~98-nt long sRNA in *C. jejuni* strains NCTC11168, 81-176, 81116, and RM1221[28]. While CJnc230 overlaps with the predicted open reading frame Cj1728c, we confirmed CJnc230 to be non-coding based on ribosome profiling[33]. As CJnc230 expression was lost in the ΔrpoN mutant and transcription of the upstream encoded *flgE* is dependent on RpoN[14,34] (Fig. 1b), we hypothesized that CJnc230 might be a processed sRNA that is co-transcribed with *flgE*. In line with this, deletion of *fliA* strongly upregulated both *flgE* and two species (98/88 nt) of CJnc230 (Fig. 1b). This upregulation is consistent with previous reports for several other RpoN-dependent *C. jejuni* flagellar genes[35,36], although the underlying

mechanism is so far unclear. Moreover, in contrast to the strong upregulation of FliA-dependent CJnc170, CJnc230 and *flgE* levels slightly decreased in a mutant lacking the anti-σ[28] factor FlgM. This uniform downregulation further supports the co-expression of CJnc230 and upstream *flgE*.

To examine CJnc230 conservation, we performed sequence analysis using blastn and GLASSgo[37]. This revealed CJnc230 is highly conserved in diverse *C. jejuni* and *C. coli* strains (Fig. 1c), the two most common agents of *Campylobacter*-mediated gastroenteritis in humans[38]. Manual inspection revealed highly similar sequences also in more distant *C. hepaticus* (85.9% identity) and *C. lari* (69.6% identity) compared to *C. jejuni* strain NCTC11168. Secondary structure predictions for the full-length 98-nt sRNA with RNAfold[39] revealed two stem-loops flanking a single-stranded stretch of 12 nt (Fig. 1d). This single-stranded region contains a conserved sequence with a potential anti-Shine-Dalgarno sequence (CCUU), complementary to the *C. jejuni* ribosome binding site (RBS) consensus motif[33], suggesting a role as a *trans*-acting mRNA regulator. In all species where a homologous CJnc230 sequence was identified in, the sRNA was encoded downstream of the corresponding *flgE* ortholog (Fig. 1b and Supplementary Fig. 1a), which, together with its dependence on RpoN, points to a possible role of CJnc230 in *Campylobacter* flagellar biogenesis.

### RNase III, RNase Y, and PNPase are involved in processing of CJnc230

CJnc230 seemed to be coupled with *flgE* mRNA levels and was detected as multiple species (~98 and ~88 nt) on northern blots (Fig. 1b). Thus, we investigated its possible co-transcription with *flgE* and subsequent processing by RNases. Unlike other Gram-negative bacteria, *C. jejuni* does not encode RNase E, responsible for processing many 3′ untranslated region (UTR)-derived sRNAs in Gammaproteobacteria[40,41], and instead encodes RNase Y[8]. Double-strand-specific RNase III is also emerging as an important player in sRNA processing and regulation[42–45], and is required for ribosomal and CRISPR RNA biogenesis in *C. jejuni*[28,46,47]. We measured CJnc230 expression in total RNA from a panel of RNase-deficient *C. jejuni* deletion mutant strains by northern blot (Supplementary Fig. 1b), which revealed that RNase III (*rnc*, Cj1635c), RNase Y (*rny*, Cj1209), and the 3′-5′ exoribonuclease polynucleotide phosphorylase (PNPase, *pnp*, Cj1253) affect its expression (Fig. 2a and Supplementary Fig. 1b). In Δrnc, mature CJnc230 was not observed, and instead a transcript longer than 1000 nt, potentially comprising *flgE*-CJnc230, was detected (Fig. 2a, lane 2). Upon deletion of the gene encoding RNase Y, several transcripts of approximately 200-300 nt in length appeared in addition to the mature sRNA bands present in wildtype (WT) (Fig. 2a, lane 3), suggesting RNase Y also affects CJnc230 processing. Accumulation of the most abundant WT RNA species (~88 nt) was increased in the PNPase deletion mutant compared to WT (Fig. 2a, lane 4, white triangle). In an *rny/pnp* double deletion strain, additional species between 110 and 300 nt were also detected (Fig. 2a, lane 5). An oligonucleotide binding downstream of the annotated sRNA 3′ end[28] (3′-extended, CSO-4297) revealed that RNase Y is involved in 3′-end maturation of CJnc230 or degradation of longer RNA species, as only the higher molecular weight fragments were detected (Fig. 2a, lane 3). PNPase seems to also contribute to processing of some 3′-extended transcripts, since the additional fragments between 200 and 300 nt were also observed in the *rny/pnp* double mutant when using CSO-4297 for hybridization (Fig. 2a, lane 5).

To further investigate processing of CJnc230, we used primer extension to locate 5′ ends in the WT and RNase mutant strains. This confirmed the previously annotated CJnc230 5′ end in the WT strain (Fig. 2b, lane 1, black triangle). This also showed that this end is potentially RNase III-dependent, as the 5′ end was shifted upstream by 28 nt in the *rnc* deletion mutant (Fig. 2b, lane 1 vs. 2, gray triangle), despite it being annotated as a transcriptional start site (TSS)[28,29]. A

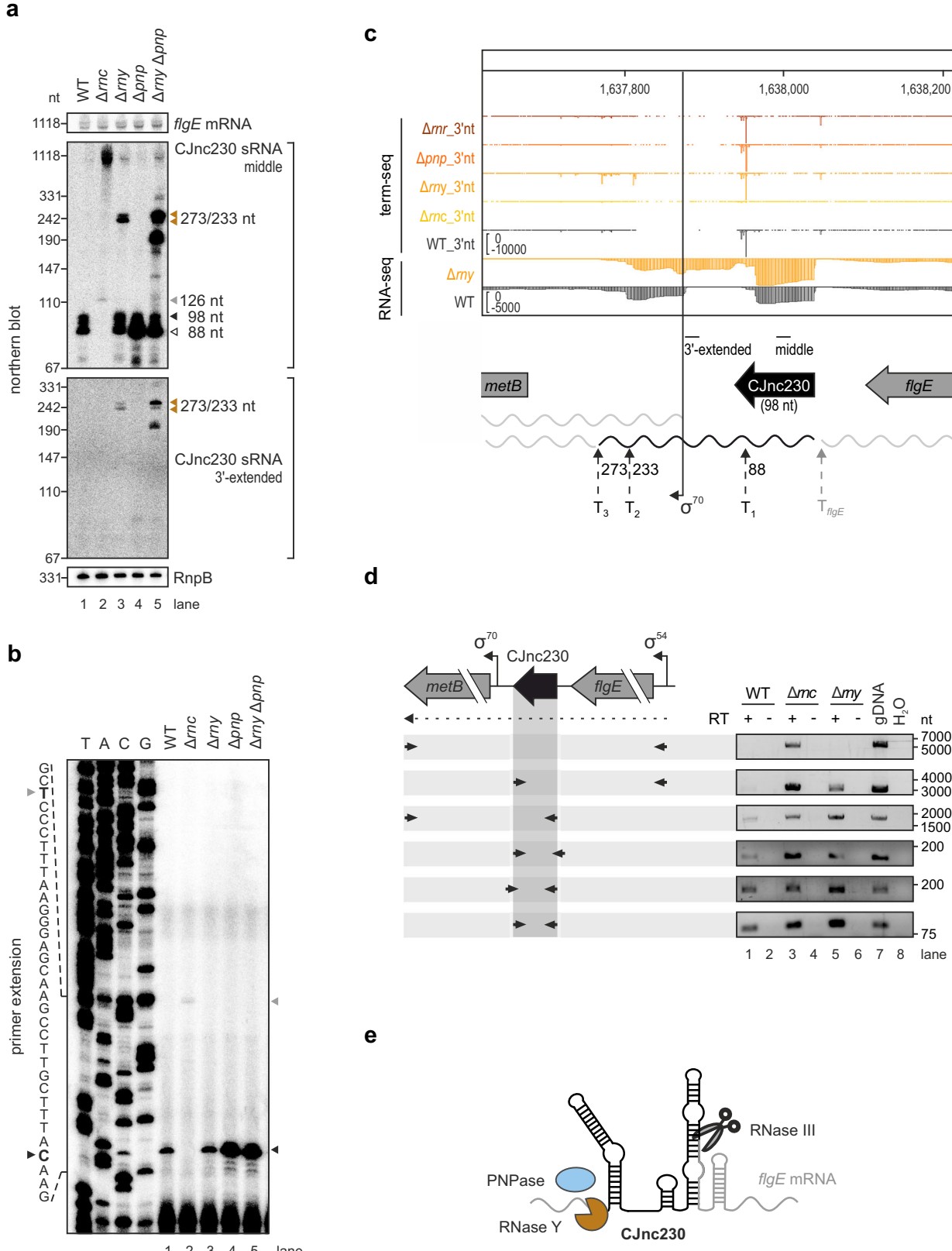

longer version of the sRNA (-126 nt) was also detected in Δ*rnc* total RNA by northern blot (Fig. 2a, lane 2, gray triangle). So far, it was unclear whether this 28 nt-extended 5' end originates from an independent promoter or from cleavage of the *flgE* promoter-derived transcript by a yet unknown RNase. All other nuclease deletion mutants tested did not show 5' end differences from WT (Fig. 2b, lane 1

vs. lanes 3-5, black triangle), although higher levels of the mature CJnc230 5' end were detected in strains without PNPase (Fig. 2b, lanes 4 and 5), comparable to northern blot observations (Fig. 2a). These results suggest that CJnc230 biogenesis involves three RNases: RNase III cleaves CJnc230 at the 5' end, while RNase Y and PNPase affect 3'-end processing of the sRNA.

**Fig. 2 | CJnc230 is co-transcribed with *flgE* and processed by RNase III, RNase Y, and PNPase. a** Northern blot analysis of total RNA from *C. jejuni* WT and ribonuclease (RNase) deletion mutant strains in exponential phase. Colored triangles: prominent CJnc230 transcripts determined by termination site sequencing (term-seq) and/or primer extension. The *flgE* mRNA was detected with CSO-5136 (binding the coding sequence (CDS)), CJnc230 sRNA was probed with CSO-0537 (middle of the sRNA) or CSO-4297 (3′-extended versions, binding ~70 nt downstream of the annotated 3′ end[28]). RnpB RNA (CSO-0497) served as a loading control. Oligonucleotide binding positions are indicated in (**c**). **b** Primer extension analysis of CJnc230 5′ ends from *C. jejuni* WT and RNase deletion mutant strains in exponential phase with the CJnc230 probe used for northern blots (CSO-0537) binding in the middle of the sRNA. A sequencing ladder generated with this probe is partially indicated on the left. Black triangle: CJnc230 5′ end in the WT (C residue in bold on the left). Gray triangle: alternative 5′ end 28 nt upstream in Δ*rnc* (T residue in bold). **c** Term-seq and total RNA-seq represented by relative cDNA read counts at the CJnc230 locus in *C. jejuni* WT and nuclease deletion mutants grown to exponential phase. One representative biological replicate each is shown. For term-seq libraries, coverage for the last base of each read was plotted (3′nt), while total RNA-seq tracks depict full read coverage. Bent arrow: *metB* transcriptional start site (TSS)[28]. Dashed arrows: prominent CJnc230 3′ ends (T_1, T_2, and T_3) with resulting sRNA lengths in nucleotides and RNA 3′ end downstream of *flgE* (T_{flgE}). Oligonucleotide binding positions used for northern blot in (**a**) are indicated. **d** Reverse transcription-polymerase chain reaction (RT-PCR) analysis of total RNA from *C. jejuni* WT, Δ*rnc*, and Δ*rny* harvested at exponential growth phase. (*Left*) The *flgE*-CJnc230-*metB* locus. Arrows below: RT-PCR primers. Bent arrows: TSSs & promoter motifs[28,29]. (*Right*) Agarose gels of RT-PCR reactions performed in the presence (+) or absence (-) of reverse transcriptase (RT). Reactions with genomic DNA (gDNA) of WT or water (H_2O) served as positive and negative controls, respectively. **e** Model of CJnc230 processing and maturation by three ribonucleases. RNase III potentially cleaves a predicted stem-loop in the *flgE*-CJnc230 transcript (Supplementary Fig. 3), while RNase Y and PNPase are involved in maturation of the sRNA 3′ end (full-length CJnc230, 98 nt, colored in black) or in degradation of precursor transcripts. Data in (**a**–**d**) are representative results of at least two independent experiments. Source data are provided as a Source Data File.

## Term-seq reveals the most abundant 3′ end of CJnc230

The CJnc230 expression pattern in WT bacteria on northern blots showed one highly abundant RNA species of lower molecular weight (~88 nt) and several slightly longer transcripts (up to ~98 nt in length) (Figs. 1b and 2a, lane 1). Since primer extension analysis identified only one 5′ end in the WT background (Fig. 2b), this pattern likely arises from differences at the sRNA 3′ end. To follow this up, we globally identified RNA 3′ ends using termination site sequencing (term-seq)[48] in *C. jejuni* NCTC11168 WT, Δ*rnc*, Δ*rny*, Δ*pnp*, and Δ*rnr* strains. A mutant of RNase R (*rnr*, Cj0631c) was included as this enzyme shows 3′-5′ exoribonuclease activity in *C. jejuni*[49]. Visual inspection of term-seq data in WT, Δ*rny*, Δ*pnp*, and Δ*rnr* revealed multiple CJnc230 3′ ends, resulting in transcripts from 87 to 98 nt in length according to the mature sRNA 5′ end (Fig. 2c). This 3′ end variability might explain the diversity of CJnc230 RNA species detected on northern blots (Figs. 1b and 2a, lane 1). The most prominent peak in WT, Δ*rny*, Δ*pnp*, and Δ*rnr* was located 88 nt downstream of the CJnc230 5′ end, consistent with the most abundant northern blot species and the primer extension-located 5′ end (Figs. 1b and 2a, 2b), and giving rise to a version of the sRNA with a predicted longer single-stranded stretch (Supplementary Fig. 2a). In contrast to the increase of the 88-nt version, the abundance of longer CJnc230 species was reduced ~2-fold over growth in WT (Supplementary Fig. 2b, panel 3′ end oligo, lanes 1, 5, and 9). The variation in CJnc230 3′ ends and the sRNA 5′ end was also confirmed by circular Rapid Amplification of cDNA Ends (cRACE) (Supplementary Fig. 2c). In agreement with northern blots, only low CJnc230 expression was observed in term-seq of Δ*rnc* bacteria and all prominent peaks were missing (Fig. 2c). Another peak was detected nine nucleotides upstream of the sRNA 5′ end in WT, which might represent a *flgE* transcript 3′ end. As this position was not immediately adjacent to the sRNA 5′ end, but was RNase III-dependent, this suggests that there is either further degradation of the *flgE* 3′ end or sRNA 5′ end.

Term-seq of Δ*rny* revealed two additional peaks 233 and 273 nt downstream of the mature sRNA 5′ end that was determined by primer extension (Fig. 2c). These positions (T_2 and T_3), also reflected by additional total RNA-seq coverage downstream of the sRNA 3′ end in the Δ*rny* library compared to WT (Fig. 2c), were consistent with two additional CJnc230 bands detected in this mutant on northern blots, thus confirming these as 3′-extended transcripts (Fig. 2a, lanes 3 and 5, orange triangles).

## CJnc230 is co-transcribed with *flgE* and *metB* mRNAs

The similar RpoN dependence of *flgE* mRNA and CJnc230 on northern blot (Fig. 1b) and the RNase III cleavage at the CJnc230 5′ end hinted towards co-transcription of *flgE* and CJnc230. We next used reverse transcription-PCR (RT-PCR) with different primer combinations to determine if they are part of the same RNA. Indeed, we found that CJnc230 is transcribed together with *flgE*, and also with the downstream gene *metB* (O-acetylhomoserine (thiol)-lyase, Cj1727c), which also has its own dedicated RpoD-dependent TSS (Fig. 2d). However, *flgE*-CJnc230-*metB* and *flgE*-CJnc230 transcripts were hardly detectable in the WT strain when RNase III was present (Fig. 2d, two upper boxes). RNase Y appears to play a role in 3′-maturation of CJnc230 (Fig. 2a), but it also seems to destabilize the *flgE*-CJnc230 transcript, as this fragment could be detected in the Δ*rny* background where RNase III is present (Fig. 2d, second box from top, lanes 1 and 5). Oligonucleotides binding the CJnc230 5′ or 3′ end were used to detect the sRNA only as a control (Fig. 2d, last box). Interestingly, a 5′-extended sRNA was also detected in the WT background, arguing for a version originating upstream of the RNase III cleavage site (Fig. 2d, third last box, lane 1). RNA species, which are extended further downstream beyond the CJnc230 3′ end, were detectable in all strain backgrounds, either only ~70 nt downstream of the 3′ end of the full-length sRNA (Fig. 2d, second last box) or until the *metB* stop codon (Fig. 2d, third box from top). A long stem-loop structure was predicted using the *flgE*-CJnc230 transcript as input (Supplementary Fig. 3), potentially serving as template for RNase III cleavage between positions 2732 and 2733 (between A-U and C-G base pairs) relative to the *flgE* TSS. Alternatively, CJnc230 maturation might also include other so-far unknown RNA partners binding to CJnc230, thereby promoting RNase III cleavage at the sRNA 5′ end.

In summary, based on northern blotting, primer extension, term-seq, cRACE, and RT-PCR, we could show that CJnc230 is generated by a complex biogenesis pathway. This involves transcription together with the RpoN-dependent gene *flgE*, cleavage downstream of the *flgE* coding sequence (CDS) by RNase III, and 3′ end maturation/degradation by RNase Y and PNPase (Fig. 2e).

## CJnc230 is also transcribed from an RpoD-promoter downstream of the *flgE* CDS

To determine whether the alternative CJnc230 5′ end observed in Δ*rnc* (Fig. 2b) originated from an independently transcribed version of the sRNA, in addition to processing from the *flgE* promoter-derived transcript, we globally mapped primary transcript 5′ ends with dRNA-seq in NCTC11168 WT and Δ*rnc*. This revealed a potential alternative TSS in Δ*rnc* with a putative RpoD-dependent promoter motif (5′-TATTGT-3′; -41 to -36 relative to annotated 5′ end[28]) (Supplementary Fig. 4), upstream of the annotated CJnc230 5′ end that was confirmed by primer extension in WT bacteria (Fig. 2b). This alternative TSS was consistent with the 28 nt-extended 5′ end detected in Δ*rnc* by primer extension (Fig. 2b). In the dRNA-seq experiment performed here we did not observe enrichment of the annotated CJnc230 5′ end[28] in the TEX-treated WT sample compared to the untreated sample. This suggests processing of the sRNA 5′ end, even though we also found a

conserved putative RpoN-dependent motif upstream of it (5′-GGxxxxxxxxTTGCTT-3′; position -18 to -3 relative to annotated 5′ end) (Supplementary Fig. 4).

To determine if there is an active promoter in this region, we generated a transcriptional reporter by fusing 150 nt upstream of the annotated sRNA 5′ end to the unrelated *metK* RBS and a superfolder GFP (sfGFP) gene[50] (Supplementary Fig. 5a). Western blot analysis of whole cell lysates confirmed that this region alone could drive transcription of the reporter. This transcription was independent of flagellar sigma factors (RpoN or FliA) (Supplementary Fig. 5a, upper panel of western blot, lanes 1 to 4), unlike the *flgE* promoter, which we confirmed is RpoN dependent (lower panel of western blot). This suggested that CJnc230 might also be transcribed from its own RpoD-dependent promoter in addition to transcription together with upstream *flgE*. Interestingly, we also observed that deletion of *fliA* increased activity of P$_{flgE}$, in line with increased *flgE* mRNA abundance in Δ*fliA* (Fig. 1b). This suggests increased *flgE* levels might originate from transcriptional feedback on class II flagellar genes via a yet unknown mechanism, as observed previously in microarray analyses[35,36].

Primer extension analysis located a 5′ end in Δ*rnc* that is in line with the alternative CJnc230 TSS 28 nt upstream of the 5′ end in WT (Fig. 2b and Supplementary Fig. 5b, lane 2, gray triangle), supporting the presence of an independently transcribed sRNA in addition to the *flgE* promoter-derived version. Analysis of strains deleted for *rpoN*, *flgE* (including its promoter), or both, also confirmed that the sRNA can be transcribed independently of *flgE*, possibly from its own RpoD-dependent promoter (black triangle). However, this transcript also appears to be further processed, as the 5′ end is the same as in WT bacteria (Supplementary Fig. 5b, lanes 6 to 8, black triangle). Strikingly, the RpoD-dependent version of CJnc230 seemed to be independent of RNase III, as the WT 5′ end and sRNA expression was detected in a *flgE/rnc* double mutant by primer extension or northern blot, respectively (Supplementary Fig. 5b, c; lane 9). Northern blot analysis confirmed the 28-nt 5′-extended transcript in Δ*rnc* samples (Fig. 2b and Supplementary Fig. 5c, lane 2, gray triangle), as well as very low levels of sRNA expression in *rpoN* and *flgE* single or double deletion mutants with a similar 88-nt length as in WT bacteria (Supplementary Fig. 5c, lanes 6 to 8, white triangle). Taken together, our data suggest that CJnc230 can also be generated by an additional biogenesis pathway, which is independent of *flgE* transcription and relies on its own RpoD-dependent promoter. However, this generates very low levels of CJnc230 under the conditions examined. The 126 nt-long precursor transcript generated from the promoter also seems further processed by a yet unknown RNase, as the sRNA 5′ end originating from either biogenesis pathway is the same.

## CJnc230 directly represses translation of Cj1387c and *flgM* mRNAs

Next, to investigate CJnc230 function and cellular targets in *C. jejuni*, we constructed deletion (Δ), complementation (C), and overexpression (OE) mutants in strain NCTC11168. Approximately 14- to 26-fold overexpression of CJnc230 was achieved during different growth phases (Supplementary Fig. 2b) by fusing the full-length sRNA (98 nt) to the RpoD-dependent *porA* promoter at the unrelated *rdxA* locus[51]. No significant difference in growth was observed for any of the mutant strains compared to WT bacteria (Supplementary Fig. 6a). However, inspection of whole cell protein samples of the mutant strains by SDS-PAGE revealed slight upregulation of bands consistent with the sizes of FlaA and FlaB flagellins[52] upon overexpression of CJnc230 (Supplementary Fig. 6b), suggesting CJnc230 might play a role in regulation of *C. jejuni* flagellar biogenesis.

To identify potential direct targets of CJnc230 in *C. jejuni* strain NCTC11168, we performed *in-silico* target predictions with IntaRNA[53] for both the 98-nt full-length and the 88-nt version of the sRNA. Three quarters of the predicted targets for full-length CJnc230 overlapped

with predictions for the shorter, more abundant 88-nt version, which has an almost two-fold longer single-stranded RNA region and also larger pool of predicted targets (Supplementary Datasets 1 and 2, Fig. 1d and Supplementary Fig. 2a). The top ten overlapping candidates included mRNAs encoding flagella-related proteins, of which *flgM* and Cj1387c mRNAs were predicted to interact with the single-stranded region of CJnc230 at their RBS (Fig. 3a), consistent with repression via competition for ribosome binding. While the anti-FliA factor FlgM (Cj1464) regulates late flagellar gene expression by sequestering σ$^{28}$ (FliA)[20] (Fig. 1a), Cj1387c was previously suggested to play a role in flagella-flagella interactions and thus named *Campylobacter* flagella interaction regulator (CfiR)[31]. In-line probing[54] supported the predicted structure of CJnc230 (Fig. 1d) and confirmed its interaction with the two targets (Fig. 3b). Addition of increasing concentrations of in-vitro transcribed, unlabeled mRNA fragments, including their RBS protected the single-stranded region of 5′-radiolabeled CJnc230 from spontaneous degradation (Fig. 3b, lanes 4 to 7 for Cj1387c and 8 to 11 for *flgM*). In contrast, a mutant sRNA with two base exchanges in the predicted interaction region (M1) was not protected, even with high concentrations of Cj1387c and *flgM* transcripts (Fig. 3b, lanes 15 to 17). These results were validated by a reciprocal experiment with 5′-radiolabeled Cj1387c and *flgM* mRNA leaders and unlabeled CJnc230 (Supplementary Fig. 7a).

Next, we confirmed direct translational repression of *flgM* and Cj1387c mRNAs by CJnc230 in an in-vitro system. Translational reporters for each target were generated by fusing their 5′UTR and first ten codons to *sfgfp* and in-vitro transcribed using T7 RNA polymerase. We observed dose-dependent translational repression of both reporters by CJnc230 on western blots (Fig. 3c, lanes 1 to 6). As a negative control, no repression was observed for a *flaA* translational reporter, which is not predicted to be a target of CJnc230 (Fig. 3c, lanes 7 and 8; Supplementary Datasets 1 and 2). When sRNA excess was further increased to 50-fold, a complete inhibition of Cj1387c and *flgM* reporter translation was observed (Supplementary Fig. 7b).

To demonstrate that CJnc230 can repress translation of Cj1387c and *flgM* in vivo, we chromosomally tagged both genes with a 3xFLAG epitope at their C-terminus. Western blot analysis of whole cell lysates revealed that while deletion of CJnc230 did not significantly affect protein levels, -26-30-fold overexpression of CJnc230 led to a significant reduction of Cj1387c- and FlgM-3xFLAG by -35% and -30%, respectively (Fig. 3d). Additionally, translational *sfgfp* reporters driven from an unrelated promoter (*metK*), fused to the 5′UTRs and first 10 codons of both targets were also downregulated in *C. jejuni* upon overexpression of the sRNA (Fig. 3e). GFP expression was significantly reduced only when WT CJnc230 was overexpressed (by -42% for *flgM* and -33% for Cj1387c), but not when overexpressing the M1 mutant, although both sRNA versions were overexpressed to similar levels (Fig. 3e). In summary, data from both in-vitro and in-vivo approaches validated post-transcriptional repression of Cj1387c and *flgM* by CJnc230, mediated via direct binding to the mRNA RBSs.

## CJnc230-mediated FlgM repression leads to increased FlaA expression, filament length, and motility

Due to its role as a central regulator and check-point in flagellar biogenesis, we further focused on exploring *flgM* mRNA as a CJnc230 target. Thus, we investigated the effect of CJnc230-mediated FlgM repression on flagellar genes downstream of FlgM. SDS-PAGE analysis of total proteins suggested that CJnc230 overexpression upregulates either the FlaA (FliA-dependent) or FlaB (RpoN-dependent) flagellins, which have a similar size (Supplementary Fig. 6b). To identify if FlaA or FlaB are affected by CJnc230, we measured levels of 3xFLAG-tagged flagellins in CJnc230 mutants by western blot analysis. While deletion of CJnc230 had no significant effect on FlaA- or FlaB-3xFLAG levels, -32-fold overexpression of the sRNA significantly upregulated epitope-tagged FlaA levels by -118% compared to the WT background (Fig. 4a

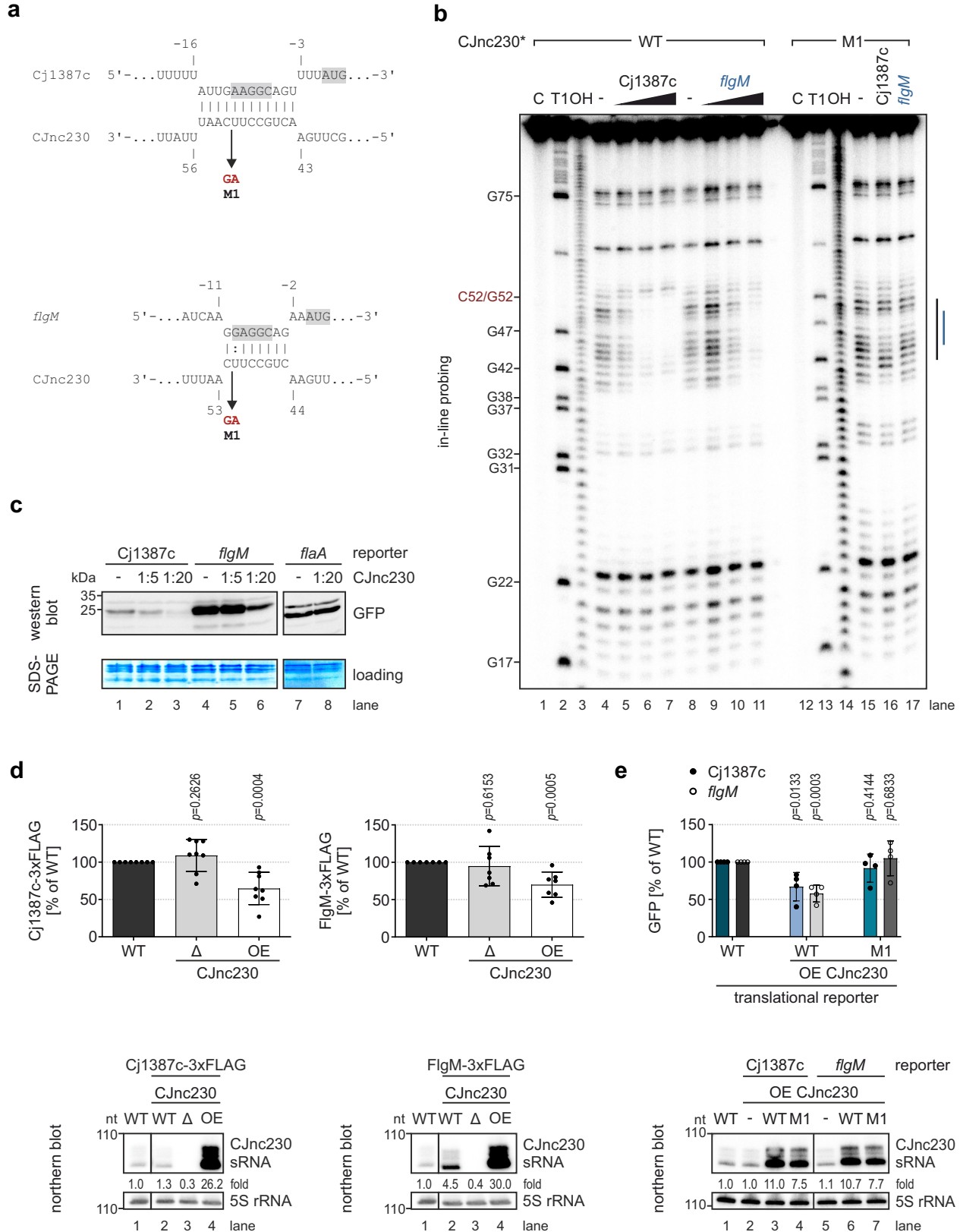

and Supplementary Fig. 8a). As FlaB-3xFLAG levels remained unchanged upon CJnc230 overexpression (Supplementary Fig. 8a), this argues for a specific effect of CJnc230 on class III flagellar genes via repression of FlgM. As a control, we tested if expression of the alternative sigma factors was altered by CJnc230 inhibition of *flgM*. Western blot analysis of CJnc230 deletion or overexpression strains in RpoN-3xFLAG or FliA-3xFLAG backgrounds showed that tagged protein levels were not affected by the sRNA (Supplementary Fig. 8b).

Because the major flagellin FlaA is the main component of the *Campylobacter* flagellar filament and is required for motility[9], we compared the swimming behavior of CJnc230 mutants to WT. While deletion of CJnc230 did not affect motility, overexpression of CJnc230

**Fig. 3 | CJnc230 interacts with the ribosome binding site (RBS) of Cj1387c and *flgM* mRNAs to repress their translation. a** Predicted interactions between the single-stranded region of CJnc230 sRNA and Cj1387c or *flgM* mRNAs (IntaRNA)[53]. Gray boxes: RBS and start codons. Arrow/M1: sRNA mutations. **b** In-line probing of 0.2 pmol [32]P-5′-end-labeled (marked with *) WT CJnc230 sRNA in the absence or presence of 0.02/0.2/2 pmol unlabeled Cj1387c or *flgM* WT leaders. Interaction sites for Cj1387c (black) and *flgM* (blue) are indicated on the right. 5′-end-labeled mutated sRNA (M1; C52G mutation in red on the left; 0.2 pmol) was incubated with 2 pmol of WT mRNA leaders. C - untreated control; T1 ladder - G residues (indicated on the left); OH - all positions (alkaline hydrolysis). **c** In-vitro translation of in-vitro transcribed mRNAs of Cj1387c/*flgM*-sfGFP reporters (4 pmol, 5′UTR and first 10 codons fused to *sfgfp*) in an *E. coli* cell-free system -/+ CJnc230 (1:5 or 1:20) detected by western blot. *flaA* 5′UTR reporter: negative control, separate western blot. PageBlue staining of the gel after transfer served as a loading control. **d** (*Upper*) Protein levels of C-terminally FLAG-tagged Cj1387c (*left*) and FlgM (*right*) in exponentially grown *C. jejuni* measured by western blot. Mean of independent biological replicates (n = 8 for Cj1387c-3xFLAG; n = 7 for FlgM-3xFLAG), error bars depict the standard deviation. Unpaired, two-tailed Student's *t*-test was used to compare the respective mutant to WT. (*Lower*) Northern blot analysis of total RNA from *C. jejuni* WT and CJnc230 deletion (Δ) and overexpression (OE) mutants in epitope-tagged backgrounds grown to exponential phase. CJnc230 was detected with CSO-0537 and 5S rRNA (CSO-0192) served as a loading control. Fold changes of CJnc230 expression relative to WT and normalized to 5S rRNA are indicated. Images in (**d**) were cut between lanes 1 and 2. **e** (*Upper*) Western blot analysis of *C. jejuni* translational reporters of Cj1387c (filled circles) and *flgM* (open circles) 5′UTRs (and first 10 codons) fused to *sfgfp*, harvested at exponential growth phase. sRNA WT or M1 mutant OE strains were compared to the 5′UTR reporters without sRNA overexpression. Mean of independent biological replicates (n = 4), error bars depict the standard deviation. Unpaired, two-tailed Student's *t*-test was used to compare the respective mutant to the WT reporter background. (*Lower*) Northern blot analysis of total RNA from *C. jejuni* WT and CJnc230 (WT/M1) OE mutants in translational reporter backgrounds grown to exponential phase. Expression of CJnc230 sRNA was detected with CSO-0537 and 5S rRNA (CSO-0192) served as a loading control. Fold changes of CJnc230 expression relative to the WT and normalized to 5S rRNA are indicated. Images in (**e**) were cut between lanes 4 and 5. Data in (**b**–**e**) are representative results of at least three independent experiments. Source data are provided as a Source Data File.

slightly, but significantly, increased the mean swim halo radius on soft agar compared to the WT strain (12 mm for WT vs. 14.4 mm for OE CJnc230) (Fig. 4b and Supplementary Fig. 9a). Additionally, as *flgM* mutants in different bacterial species, as well as another *C. jejuni* strain grown under different conditions, express longer flagella or have twice the number of flagella per cell[20,55,56], we determined whether flagellar morphology was also affected by CJnc230. Transmission electron microscopy revealed significantly shorter flagella in the CJnc230 deletion background and significantly longer flagella in the sRNA overexpression mutant compared to the mean filament length in WT bacteria (WT: 3064 nm; ΔCJnc230: 2614 nm; OE CJnc230: 3549 nm) (Fig. 4c and Supplementary Dataset 3). Increased flagellum length upon overexpression of CJnc230 was further validated by using a mutated version of FlaA (S395C) in combination with a maleimide-conjugated fluorophore, allowing for staining of the flagellar filament. Consistent with electron microscopy, we generally observed that CJnc230 overexpression mutants expressed longer flagella than ΔCJnc230 or WT cells using confocal microscopy (Fig. 4d). Bacteria with a *fliA* deletion were either aflagellate or had shorter flagella, possibly consisting of FlaB minor flagellin units only (class II, RpoN-dependent). This resulted in a mean flagellum length of 1100 nm (Fig. 4c and Supplementary Dataset 3), which is similar to previous reports for this mutant[57]. We did not observe a significantly different mean flagellum length of a Δ*flgM* strain compared to WT (2926 nm) (Fig. 4c and Supplementary Dataset 3). This indicates that there might be additional mechanisms compensating for the complete loss of FlgM in the Δ*flgM* deletion mutant, potentially leading to stronger feedback, that are absent in the CJnc230 overexpression strain, or that additional CJnc230 targets affect flagellum length. Taken together, our data suggest a connection between CJnc230 regulation of *flgM* and major flagellin protein expression and an increase in filament length (Fig. 4e).

## CJnc230 overexpression increases transcription of class III genes, including the sRNA CJnc170

Since we observed that CJnc230 overexpression increased *C. jejuni* NCTC11168 motility and flagellar length (Fig. 4b–d), we hypothesized that these phenotypes are due to increased FlaA levels as a result of FlgM repression (and FliA de-repression). Thus, to test whether CJnc230 overexpression de-represses transcription of FliA-dependent genes, we measured *flaA* mRNA levels by northern blot in WT and CJnc230 mutants. In addition, we also examined levels of the CJnc170 sRNA (Supplementary Fig. 10), which was previously shown to be an unprocessed FliA-dependent transcript[28,29] (Fig. 1b) and suggested to control multiple RpoN-dependent genes, such as *flgE* (Cj1729c), *flaB* (Cj1338c), and *flgP* (Cj1026c)[30]. Further, ribosome profiling analysis of *C. jejuni*

strain NCTC11168, the same isolate, grown in rich media confirmed the 47-nt long CJnc170 sRNA is non-coding[33]. Compared to WT, ΔCJnc230 did not show different levels of either *flaA* mRNA or CJnc170 sRNA (Fig. 5a, compare lanes 1 and 2). However, overexpression of CJnc230 increased the abundance of both transcripts (Fig. 5a, compare lanes 1 and 4). Transcript levels of the RpoN-dependent class II gene *flgE* were slightly decreased in the CJnc230 overexpression mutant and remained unchanged in deletion and complementation strains. Although expression of *flgE* mRNA (and CJnc230) was lost in the *rpoN* deletion mutant, confirming this transcript as class II gene, it was increased in the Δ*fliA* strain (Fig. 5a, compare lanes 1 and 6), suggesting that there is transcriptional feedback on class II flagellar genes (Supplementary Fig. 5a, lower panel). In contrast, *flaA* mRNA and CJnc170 sRNA were downregulated in Δ*rpoN* but completely lost upon deletion of *fliA*, as expected for class III genes (Fig. 5a).

To test whether increased *flaA* mRNA and CJnc170 sRNA levels observed upon CJnc230 overexpression arose from elevated transcription of class III promoters, we constructed transcriptional reporters including ~200 nt upstream of their respective TSSs fused to an unrelated RBS (*metK*) and *sfgfp* (Fig. 5b and Supplementary Fig. 11a). Deletion of *rpoN*, *fliA*, or *flgM* confirmed the functionality of the reporters, as expression measured on western blots and via flow cytometry was strongly reduced in Δ*rpoN* and Δ*fliA* backgrounds, and increased in Δ*flgM* (Fig. 5b, c and Supplementary Fig. 11). Deletion or overexpression of CJnc230 slightly reduced or significantly upregulated P*flaA* reporter expression, respectively, by western blot analysis, compared to WT (Fig. 5b). In contrast, deletion or overexpression (~7-fold, via the *porA* promoter) of FliA-dependent CJnc170 did not significantly affect *flaA* reporter expression. The significant upregulation of P*flaA*-sfGFP in OE CJnc230 was validated by flow cytometry, which further showed a significant reduction in P*flaA* reporter expression upon deletion of CJnc230 (Fig. 5c and Supplementary Fig. 11b). Similar trends as for P*flaA*-sfGFP were observed for the P*CJnc170* reporter. Overexpression of CJnc230 increased P*CJnc170*-sfGFP expression, but not quite to the levels of *flgM* deletion (Fig. 5c and Supplementary Fig. 11).

Overall, the transcriptional reporter analyses indicate overexpression of CJnc230 increases transcript levels of class III genes of the flagellar regulatory cascade, e.g., *flaA* and CJnc170. Moreover, our data show that this effect is due to increased FliA-dependent promoter activity, which is likely mediated by CJnc230 repression of the anti-σ[28] factor FlgM.

## CJnc170 and CJnc230 balance flagellar biosynthesis and motility

So far, we have established a model for CJnc230 function in *C. jejuni* flagellar biogenesis in which repression of the anti-σ[28] factor FlgM

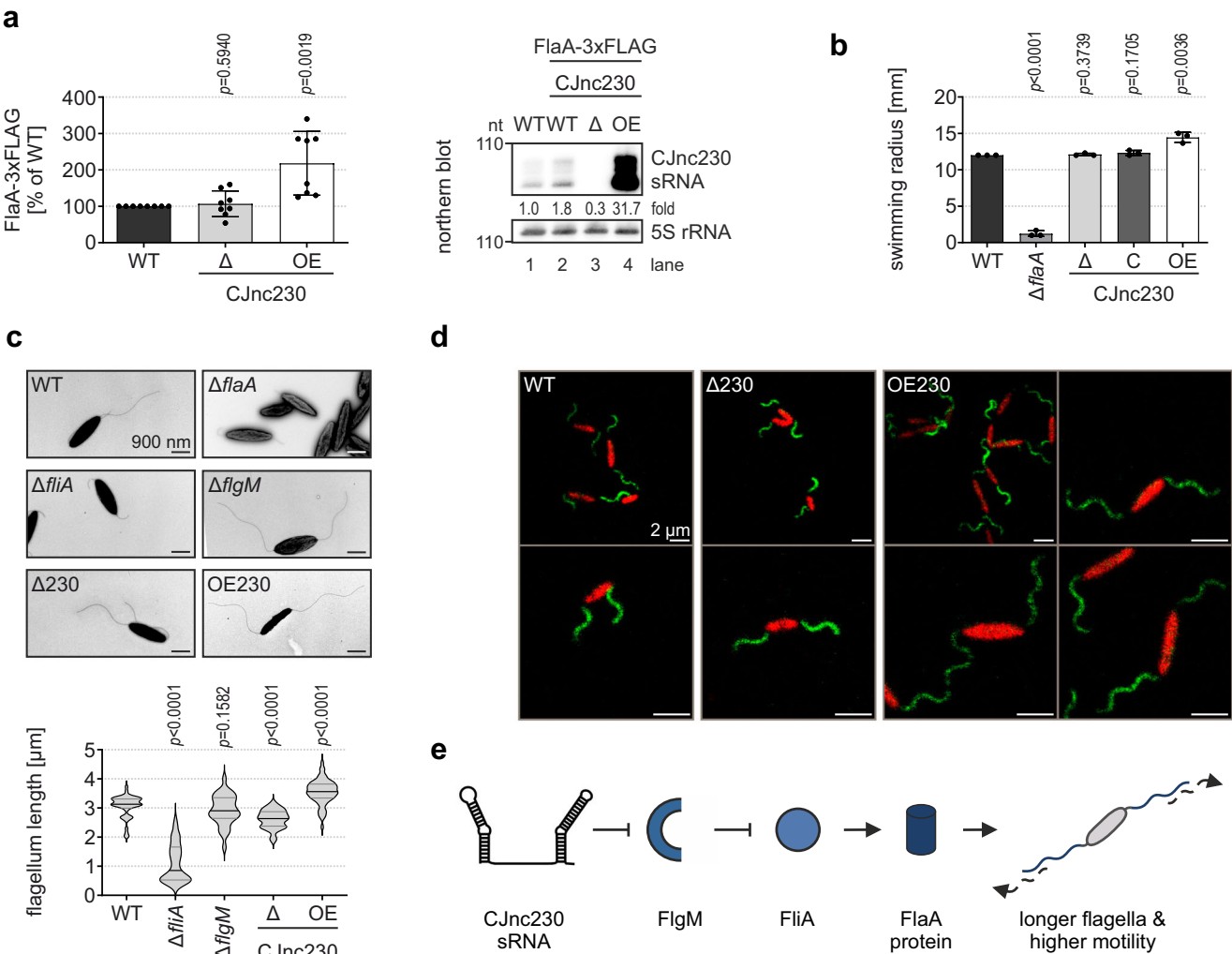

**Fig. 4 | CJnc230 overexpression increases major flagellin levels, filament length, and motility. a** (*Left*) FlaA-3xFLAG levels in *C. jejuni* WT, CJnc230 deletion (Δ), and overexpression (OE) mutants harvested at exponential growth phase and quantified by western blotting. Mean of independent biological replicates (n = 8), error bars depict the standard deviation. Unpaired, two-tailed Student's *t*-test was used to compare the respective mutant to WT. (*Right*) Northern blot analysis of total RNA from *C. jejuni* WT and CJnc230 mutants in the FlaA-3xFLAG background grown to exponential phase. CJnc230 sRNA was detected with CSO-0537 and 5S rRNA (CSO-0192) served as a loading control. Fold changes of CJnc230 expression relative to WT and normalized to 5S rRNA are indicated. **b** Motility assays of WT and mutant strains with ΔCJnc230, complementation in *trans* (C CJnc230), and OE CJnc230 in 0.4% soft agar BB plates quantified after 24 hrs of incubation. Δ*flaA*: non-motile control. Error bars: standard deviation of the mean of independent biological replicates (n = 3). Unpaired, two-tailed Student's *t*-test vs. WT. A representative plate image is shown in Supplementary Fig. 9a. **c** Transmission electron microscopy

(TEM) of *C. jejuni* WT, Δ*flaA*, Δ*fliA*, Δ*flgM*, and CJnc230 deletion (Δ230) or overexpression (OE230) mutants. (*Upper*) Representative TEM images at 6000 x magnification, scale bar = 900 nm. (*Lower*) Violin plots depict the median (black) and quartiles (gray) of at least 30 flagellum length measurements per strain. Unpaired, two-tailed Student's *t*-test was used to compare the respective mutant to WT. **d** Representative confocal microscopy images of *C. jejuni* expressing a second copy of *flaA* with an S395C mutation under control of its native promoter introduced at the *rdxA* locus. Bacteria were harvested at exponential growth phase and flagella were stained with DyLight™ 488 maleimide (green), while cell bodies were counterstained with eFluor™ 670 (red). Scale bar = 2 µm. **e** Model of CJnc230 function in *C. jejuni* strain NCTC11168. The sRNA represses translation of *flgM*, thereby derepressing FliA-dependent genes, e.g., *flaA*. This leads to increased FlaA protein levels, filament length, and motility. Data in (**a**–**d**) are representative results of at least two independent experiments. Source data are provided as a Source Data File.

transcriptionally activates FliA-dependent class III flagellar genes such as the major flagellin *flaA* mRNA. This also upregulates the CJnc170 sRNA. CJnc170, together with its paralog CJnc10, was previously suggested to regulate *C. jejuni* class II flagellar genes, including *flgE* upstream of the CJnc230 sRNA[30]. Although no motility or flagellar morphology phenotype had been described so far for CJnc10 and CJnc170 deletion or overexpression mutants, the predicted targets of these two sRNAs point to possible feedback regulation on flagellar biosynthesis[30]. Since levels of CJnc10 seemed to be very low throughout growth in NCTC11168[28,30] but CJnc170 was readily detectable on northern blot (Figs. 1b and 5a), we focused on CJnc170 for further experiments.

To explore a possible interplay between CJnc170 and CJnc230 in controlling *C. jejuni* motility, we constructed various combinations of sRNA single or double mutants. We observed that overexpression, but not deletion, of CJnc230 in *C. jejuni* strain NCTC11168 significantly increased bacterial motility (Figs. 4b and 6a, Supplementary Fig. 9a). In contrast, 21-fold overexpression of CJnc170 led to significantly reduced swimming motility compared to the WT (13.7 mm for WT vs. 8.7 mm for OE CJnc170), while ΔCJnc170 had no effect (Fig. 6a and Supplementary Fig. 9b). Strikingly, deletion of CJnc230 in combination with CJnc170 overexpression completely abolished bacterial motility to levels observed for a Δ*flaA* mutant, revealing a phenotype not observed with single sRNA mutants before. In contrast, both deletion

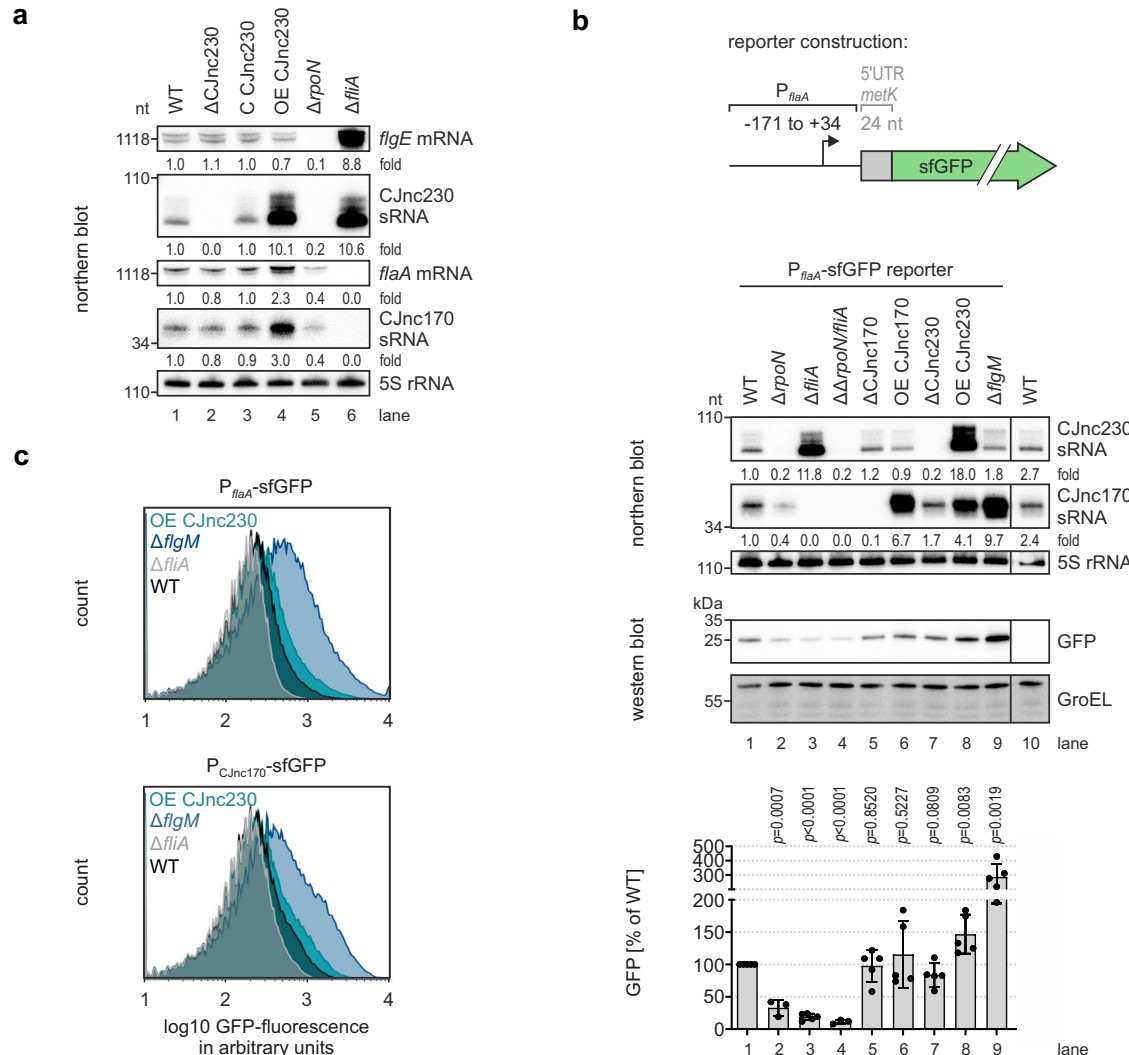

**Fig. 5 | CJnc230-mediated post-transcriptional repression of *flgM* increases transcription of class III flagellar genes. a** Northern blot analysis of total RNA from *C. jejuni* WT and CJnc230/sigma factor mutant strains grown to exponential phase. Transcripts were detected with the following radiolabeled oligonucleotides: *flgE* mRNA - CSO-5136, CJnc230 sRNA - CSO-0537, *flaA* mRNA - CSO-0486, CJnc170 sRNA - CSO-0182, 5S rRNA - CSO-0192 (loading control). Fold changes of sRNA/mRNA expression relative to WT and normalized to 5S rRNA are indicated. **b** Northern and western blots of *C. jejuni* WT, sigma factor, sRNA, or anti-sigma factor mutant strains with a transcriptional reporter of the *flaA* promoter region (nucleotide positions with respect to the *flaA* mRNA TSS[28,29] are indicated in the scheme on top) fused to an unrelated ribosome binding site (RBS) (*metK*) and *sfgfp*, harvested at exponential growth phase. (*Middle panel*) CJnc230 and CJnc170 sRNA expression was probed on the northern blot with CSO-0537 and CSO-0182,

respectively. 5S rRNA (CSO-0192): loading control. Fold changes of sRNA expression relative to the WT reporter and normalized to 5S rRNA are indicated. Representative images were cut between lanes 9 and 10. (*Lower panel*) Reporter expression measured by western blotting. GroEL was detected for normalization. One representative blot (cut between lanes 9 and 10) is shown with mean of independent biological replicates plotted below (n = 5 for WT, Δ*fliA*, ΔCJnc170, OE CJnc170, ΔCJnc230, OE CJnc230, and Δ*flgM*; n = 3 for Δ*rpoN* and Δ*rpoN*/Δ*fliA*). Error bars represent the standard deviation. Unpaired, two-tailed Student's *t*-test was used to compare the respective mutant to the WT reporter background (lane 1). **c** Flow cytometry analysis of P*flaA* (*upper*) and P$_{CJnc170}$ (*lower*) transcriptional fusions grown to exponential phase. Representative histograms are shown. See also Supplementary Fig. 11. Data in (**a**–**c**) are representative results of at least three independent experiments. Source data are provided as a Source Data File.

or overexpression of CJnc170 restored the increased motility phenotype of OE CJnc230 alone back to WT. Finally, the ΔCJnc170/ΔCJnc230 double deletion mutant showed similar motility compared to WT (Fig. 6a and Supplementary Fig. 9b).

Transmission electron microscopy analysis revealed that flagellar length partially correlated with motility of the respective mutant (Fig. 6b and Supplementary Dataset 4). We observed shorter filament length in the CJnc170 overexpression mutant, which might account for its reduced motility. Similarly, very short flagella were observed in the non-motile OE CJnc170/ΔCJnc230 mutant. However, significantly longer flagella were still found in CJnc230 overexpression mutants combined with CJnc170 deletion or overexpression (WT: 2776 nm; ΔCJnc170: 2765 nm; OE CJnc170: 972 nm; ΔCJnc170/ΔCJnc230:

2566 nm; OE CJnc170/ΔCJnc230: 555 nm; ΔCJnc170/OE CJnc230: 3373 nm; OE CJnc170/OE CJnc230: 3640 nm) (Fig. 6b and Supplementary Dataset 4). This suggests there are sRNA effects on motility that are independent of filament length and driven by yet unknown CJnc230 targets/functions. We also cannot rule out that there are even additional regulators (transcriptional or post-transcriptional) in the cascade, including under specific conditions. We have assayed steady-state expression levels of the two sRNAs under different growth phases in *C. jejuni* NCTC11168 WT as well as in a panel of non-essential transcriptional regulator deletion strains (Supplementary Fig. 12a, b). Both sRNAs were slightly upregulated in lag phase in minimal media (MEM) compared to rich Brucella broth (BB). They showed decreased abundance at stationary phase, which could be rescued by iron

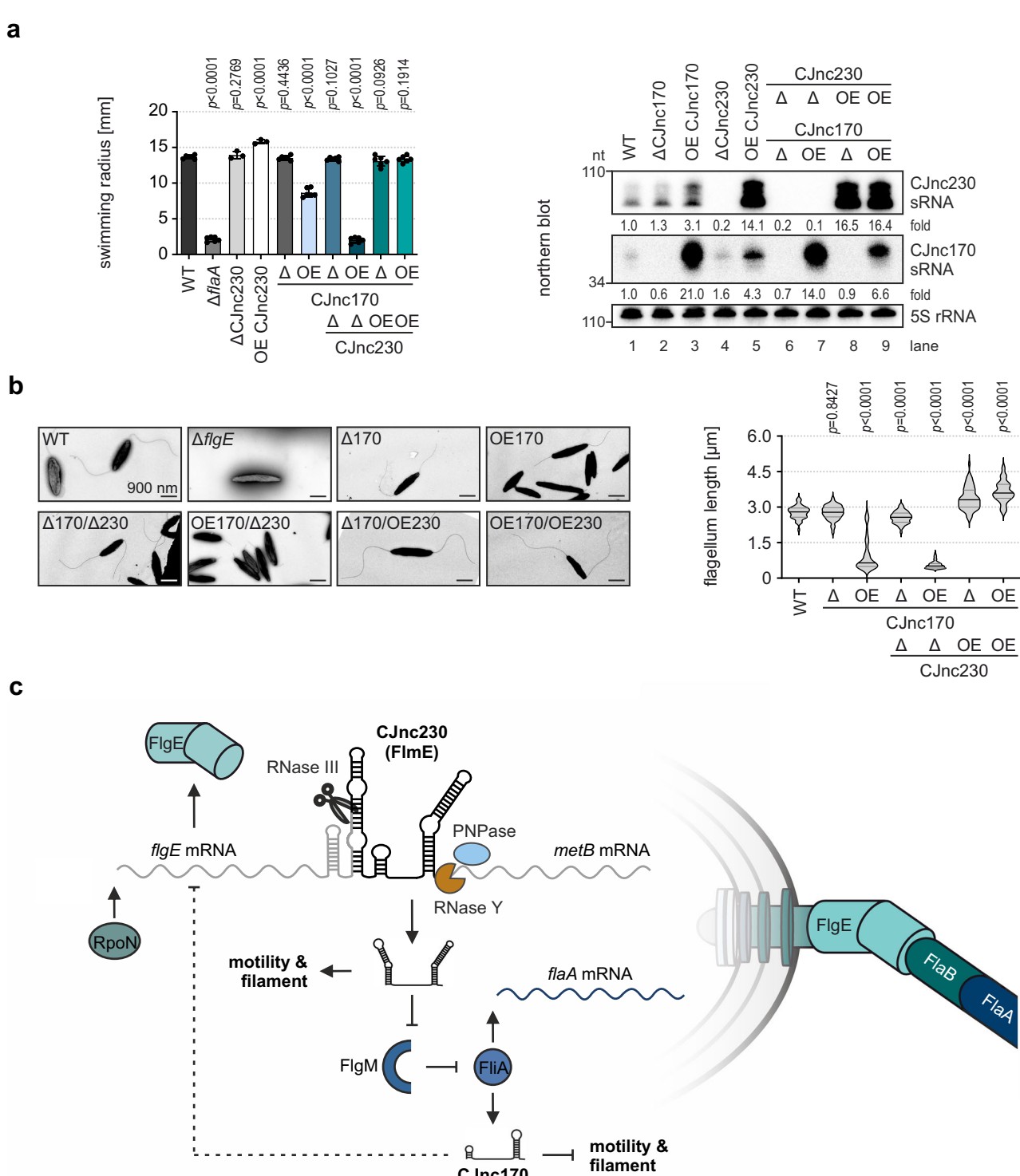

supplementation (MEM+Fe) (Supplementary Fig. 12a). In contrast to strong control by the flagellar regulators (Fig. 1b), neither sRNA was substantially affected by deletion of several transcriptional regulators (Supplementary Fig. 12b). Based on their apparently dedicated effects on filament length and bacterial motility, we propose renaming CJnc230/Jnc170 to FlmE/FlmR (flagellar length and motility enhancer/repressor). By uncoupling transcription of the sRNAs from their native position in the flagellar expression cascade, our data has shown that they have opposite effects on motility and filament length. This suggests that the two sRNAs balance each other's impact on expression through the cascade (summarized in Fig. 6c).

## Discussion

In this study we have characterized a pair of *C. jejuni* sRNAs that are themselves hierarchically regulated by two flagellar sigma factors and have opposing effects on flagellar biosynthesis. Overexpression of CJnc230 (FlmE), processed from the RpoN-dependent *flgE* mRNA by RNase III, appears to upregulate class III flagellar gene expression via translational repression of the anti-σ[28] factor FlgM. This increases FlaA levels, as well as flagellar length and motility. CJnc230 overexpression also influences expression of the class III sRNA CJnc170 (FlmR). In contrast to FlmE, overexpression of FlmR reduces flagellar length and motility, possibly via feedback on class II genes, including *flgE*

**Fig. 6 | RpoN-dependent CJnc230 and FliA-dependent CJnc170 have opposing effects on motility and flagellar biogenesis. a** (*Left*) Motility assays of *C. jejuni* WT and CJnc170 deletion (Δ) or overexpression (OE) mutants with or without deletion or overexpression of CJnc230 in 0.4% soft agar BB plates quantified after 24 hrs of incubation. Δ*flaA*: non-motile control. Error bars: standard deviation of the mean of independent biological replicates (n = 3 for CJnc230 deletion and overexpression, n = 6 for all other strains). Unpaired, two-tailed Student's *t*-test vs. WT. A representative plate image is shown in Supplementary Fig. 9b. (*Right*) Northern blot analysis of total RNA from *C. jejuni* WT and CJnc170/CJnc230 deletion or OE mutants with or without deletion or overexpression of the respective other sRNA, harvested at exponential growth phase. CJnc230 and CJnc170 sRNA expression was probed with CSO-0537 and CSO-0182, respectively. 5S rRNA (CSO-0192): loading control. Fold changes of sRNA expression relative to WT and normalized to 5S rRNA are indicated. **b** Transmission electron microscopy (TEM) of *C. jejuni* WT, Δ*flgE*, and CJnc170/CJnc230 single or double deletion and/or OE mutants. (*Left*) Representative TEM images at 6000 x magnification, scale bar = 900 nm. (*Right*) Violin plots

depict the median (black) and quartiles (gray) of at least 44 flagellum length measurements per strain. Flagella are completely absent on Δ*flgE* bacteria and, thus, could not be quantified. Unpaired, two-tailed Student's *t*-test was used to compare the respective mutant to the WT. **c** Model for CJnc230 (FlmE) and CJnc170 (FlmR) interplay in *C. jejuni* strain NCTC11168. CJnc230 is primarily transcribed together with the RpoN-dependent flagellar hook gene *flgE* and then processed downstream of the *flgE* CDS by RNase III. The sRNA 3' end is further matured by RNase Y and PNPase. Overexpression of full-length CJnc230 (FlmE, 98 nt) increases filament length and motility, potentially via repression of the anti-σ²⁸ factor FlgM, which de-represses class III flagellar genes such as *flaA* mRNA and the CJnc170 sRNA (dependent on FliA, σ²⁸). Overexpression of CJnc170 (FlmR), in contrast, decreases motility and flagellar length, possibly via repression of class II flagellar genes such as *flgE*[30] (dashed line). Overall, two hierarchically controlled sRNAs serve a regulatory circuit to control *C. jejuni* flagellar biogenesis. Data in (**a**, **b**) are representative results of at least three independent experiments. Source data are provided as a Source Data File.

upstream of FlmE[30]. Together, our data point to an additional layer of post-transcriptional regulation in *C. jejuni* flagellar biosynthesis mediated by sRNAs, whose role might be to fine-tune the cascade. Together with the RpoN and FliA sigma factors, FlmE and FlmR comprise a mixed regulatory circuit, integrating transcriptional and post-transcriptional control[58], and possibly contribute to a faster switch from middle to late flagellar gene expression at reduced metabolic cost compared to protein factors[59] (Fig. 6c).

Reminiscent of FlmE and FlmR, four UTR-derived, σ²⁸-dependent *E. coli* sRNAs (UhpU, MotR, FliX, and FlgO) have recently been described to have varied effects on flagellin protein levels, flagellar number, and motility[60]. These effects are mediated by regulation of diverse targets by the four sRNAs, including middle and late flagellar genes, a transcriptional regulator, and a ribosomal protein mRNA[60]. While there is no sequence homology between the *E. coli* and *C. jejuni* flagella sRNAs and there are differences in their targets as well as their biogenesis and hierarchical order of transcription, post-transcriptional fine-tuning of the flagellar biogenesis cascade involving counteracting regulatory RNAs appears to be a functionally conserved concept. Several other *E. coli* sRNAs post-transcriptionally regulate flagellar biosynthesis, but are controlled by environmental signals rather than flagellar sigma factors (e.g., EnvZ/OmpR and osmotic stress for OmrA/B)[61,62]. Also, OmrA/B, ArcZ, OxyS, and McaS modulate the master regulator FlhDC[25–27], which is absent in *C. jejuni*. Abundance of *flhD* mRNA also seems to be modulated by a *cis*-encoded RNA in conjunction with RNase III[63]. Only a handful of *E. coli* sRNAs target later points in the cascade. Like FlmE, *E. coli* OmrA/B sRNAs were recently shown to activate class III genes via repression of *flgM* mRNA[64], while the Esr41 sRNA of enterohemorrhagic *E. coli* activates *fliA* transcription[65–67]. In *C. jejuni*, paralogous CJnc10 and CJnc170 (FlmR) sRNAs were previously suggested to regulate class II flagellar genes but motility phenotypes associated with the sRNAs were not previously observed[30]. Due to differences in motility assay protocols and lower sRNA overexpression levels compared to our study, a motility phenotype might have been previously missed for FlmR[30]. In our study, fusion of CJnc230 or CJnc170 to the *porA* (Cj1259) promoter allowed for high sRNA expression and uncoupling of the flagella cascade (Fig. 6a). *C. jejuni* motility also appears to be subject to environmental control mediated by sRNAs[68]. Deletion of CJnc110 and CJnc140, which themselves do not appear to be directly regulated by flagellar sigma factors (based on promoter motif predictions)[29], increased bacterial swimming behavior of the highly virulent *C. jejuni* sheep abortion clone IA3902, although direct molecular targets were not identified[69,70].

FlmE, initially discovered during primary transcriptome analysis of *C. jejuni*[29], caught our attention while looking for sRNAs that might be controlled by flagellar regulators. Our in-depth 5'-end mapping and co-transcriptional analysis indicate that a minor fraction is transcribed from its own RpoD-dependent promoter, while most of FlmE

expression seems to be generated by processing from the same RpoN-dependent transcript as *flgE*, encoding the flagellar hook protein. FlmE adds to an expanding number of bacterial sRNAs that are processed by RNase III[71–74]. So far, examples are mainly restricted to bacteria that lack RNase E, such as Gram-positive *Staphylococcus aureus* (RsaC, via cleavage of the *mntABC* mRNA 3'UTR)[43,71]) and *C. jejuni* (processing of the antisense sRNA pair CJnc180/190[73]). While RNase III is required to generate the mature 5' end of *flgE*-derived FlmE (Fig. 2b), the exact mechanism remains unclear. RNase III potentially cleaves a predicted intramolecular stem-loop downstream of the *flgE* CDS (Supplementary Fig. 3), and maturation of the 5' end might additionally involve trimming by exonucleases such as RNase J, which is essential in *C. jejuni* NCTC11168[75], or yet unidentified RNases as in the case of the RpoD-dependent FlmE precursor (Supplementary Fig. 5b, c). *S. aureus* RNases J1/J2, for example, trim the 5' end of *uhpT* mRNA to generate the 3'UTR-derived sRNA RsaG[76], and *Bacillus subtilis* RNase J1 trims the signal recognition particle (SRP) RNA component scRNA (small cytoplasmic RNA) after processing by RNase III[77]. While 3'UTR-derived sRNAs control multiple physiological processes in diverse bacterial species[40,41], also with implications for larger regulatory networks such as the bacterial envelope stress response[78,79], processing of flagellar mRNAs to produce sRNAs that function in the same pathway is a relatively new concept. To the best of our knowledge, only two sRNAs that potentially repress flagellar biogenesis and that are also processed from flagellar mRNAs have been reported in *E. coli*: FlgO and FliX, which are processed from the 3'UTRs of FliA-dependent *flgL* and *fliC* mRNAs, respectively[60,80]. While FlgO has a minor impact on flagellar number and motility, FliX seems to repress these two traits via regulation of several middle and late flagellar genes and an mRNA encoding ribosomal proteins[60].

RNase Y and PNPase likely play a role in 3' end maturation/degradation of FlmE (Fig. 2a). In contrast to the complete absence of mature FlmE sRNA in Δ*rnc*, these RNases only affected the abundance of the most stable 88-nt species (Δ*pnp*) or additional higher molecular weight fragments (Δ*rny* and Δ*rny*/Δ*pnp*). This hints at a combined action of endo- and exonucleases in *C. jejuni* RNA maturation or degradation as observed in *Streptococcus pyogenes* or *B. subtilis*[81,82]. In Gram-positive bacteria such as *S. aureus* and *B. subtilis*, where RNase Y is highly conserved in place of RNase E[83,84], it has global effects on the transcriptome, with 10-30% of genes differentially expressed upon its absence[85–88]. Our data suggest that RNase Y might have similar effects in Gram-negative *C. jejuni*, which likewise lacks RNase E, as approximately 5% of genes were affected in Δ*rny* (Supplementary Dataset 5). This points to a possible broad regulatory role for *C. jejuni* RNase Y in transcriptome stability and sRNA processing.

Overexpression of FlmE increases motility, and we have identified two mRNA targets repressed by FlmE that might underlie this phenotype: Cj1387c and *flgM*. Cj1387c (CfiR, *Campylobacter* flagella

interaction regulator) encodes a putative transcriptional regulator with a PAS domain that was reported to modulate flagella-flagella interactions via post-translational flagellar modification, leading to reduced autoagglutination of ΔCj1387c bacteria[31]. In *C. jejuni* strain 81-176, the Cj1387c ortholog (HeuR, heme uptake regulator) promotes chicken colonization and regulates iron metabolism and methionine biosynthesis, possibly via direct binding to promoter regions[89,90]. Although the contribution of Cj1387c to the phenotypes presented here is not completely clear, this points to an additional role for FlmE not only in regulating flagellar biosynthesis, but potentially also in important metabolic pathways via repression of Cj1387c, thereby connecting these infection-relevant processes[91–93].

The second validated target of FlmE repression is the anti-$\sigma^{28}$ factor FlgM, a key hub in the transcriptional checkpoint between class II and class III genes. In line with the role of FlgM in sequestering FliA activity[20], we measured increased FlaA levels upon overexpression of FlmE (Figs. 4a and 5a). We also observed longer flagella in the FlmE overexpression strain, suggesting that an increase in FlaA levels might increase filament length (Fig. 4c–e), which can affect the secretion of virulence-associated factors[94]. Additional predicted FlmE/FlmR sRNA targets (Supplementary Datasets 1 and 2)[30] that could impact flagella/motility functions, including beyond biogenesis (e.g., rotation, chemotaxis), might also be part of the complex flagellar regulatory network, potentially leading to feedback and unpredictable effects. These include for example the fT3SS component FlhB (Cj0335) or RpoN-dependent hypothetical proteins Cj0243c and Cj0428. However, it is challenging to dissect on the molecular level the effects on flagellar biogenesis and/or function, which remains to be studied in more specific experimental setups to monitor also, e.g. swimming speed or directionality, as well as measure these parameters under different conditions or at the single-cell level. Identification of conditions or additional regulators controlling FlmE and FlmR, or additional targets outside of flagella regulation, might point to a role for these sRNAs beyond fine-tuning the biogenesis cascade to modulating flagellar morphology in response to different environments. This would expand the complexity of the system, whose temporal and spatial dynamics in individual bacteria might be disentangled by future studies with single-cell transcriptomics or imaging[95–97].

Although our combined strong modulation of relative FlmE and FlmR levels overcame potential feedback mechanisms and revealed phenotypes not observed in single deletion mutants before (Fig. 6a, b), testing under different experimental conditions might increase the relatively subtle effects on motility of sRNA single mutants that we observed, and potentially under conditions that more closely resemble a natural situation. Our preliminary screen under different growth phases and in a panel of regulator mutants has not yet revealed any regulators or conditions that strongly impact sRNA expression (Supplementary Fig. 12). However, we cannot rule out that some as-of-yet unknown conditions affect sRNA levels to modulate motility. Further, we hypothesize that strong phenotypes arising from sRNA-mediated fine-tuning of the tightly controlled flagellar biogenesis cascade will only be visible in single bacterial cells or in synchronized cultures, in particular as flagella biogenesis is coupled to cell-division in *C. jejuni*[98]. As it is not yet possible to synchronize flagellar assembly in *C. jejuni* via inducible promoters, for example, determining the exact temporal modulation of each flagellar building block by the sRNAs awaits future study.

In *C. jejuni*, *flgM* is both co-transcribed with the upstream RpoN-dependent genes *flgI* and *flgJ*, as well as independently transcribed from its own FliA promoter[29,99]. Temperature also affects *flgM* transcription and the interaction of FlgM protein with FliA[20,100]. Here, we further expand the regulatory complexity of the FlgM-FliA checkpoint inhibition strategy in *C. jejuni* with a layer of post-transcriptional control mediated by RpoN-dependent FlmE. The timing of regulation under native conditions, with *flgM* transcribed from two distinct promoters, is still unclear. This layer also likely interacts with the CsrA-FliW partner-switch mechanism that controls *flaA* mRNA translation and localization in *C. jejuni*[57]. This mechanism is distinct from Gram-negatives such as *E. coli*, where CsrA is titrated by sRNAs CsrB/C and activates the motility cascade through binding to *flhDC* mRNA[101], and more closely reflects Gram-positives. Together with the unique architecture of transcriptional control (e.g., lack of the master regulator FlhDC)[14,15], our results further set *C. jejuni* and the Epsilonproteobacteria apart as distinct models for flagella regulation.

Overall, our study establishes regulatory RNAs as potent post-transcriptional modulators of *C. jejuni* flagellar filament assembly and motility. In the absence of a dedicated master regulator of flagellar gene expression, *C. jejuni* has evolved sophisticated transcriptional mechanisms to grant proper biogenesis of its flagellar machine, with alternative sigma factors RpoN and FliA at the forefront. Our presented work expands the regulatory complexity of the system by sRNA-mediated control of the hierarchical cascade, which fine-tunes *C. jejuni* motility, an important virulence trait of this human pathogen.

## Methods

### Bacterial strains, plasmids, and oligonucleotides
Bacterial strains used in this study are listed in Supplementary Dataset 6, while plasmids are in Supplementary Dataset 7. DNA oligonucleotides used for cloning, T7 transcription template generation, and as northern blot probes are listed in Supplementary Dataset 8.

### Bacterial standard growth
*E. coli* strains were grown aerobically at 37 °C in Luria-Bertani (LB) broth or on LB agar, where necessary supplemented with 100 μg/ml ampicillin (Amp), 20 μg/ml chloramphenicol (Cm), 20 μg/ml gentamicin (Gm), or 20 μg/ml kanamycin (Kan) for marker selection. *E. coli* strains were stored at -80 °C in LB media containing 10% DMSO.

*C. jejuni* was grown on Mueller-Hinton (MH) agar plates supplemented with 10 μg/ml vancomycin. For selection of transformants and growth of mutant strains, 50 μg/ml Kan, 20 μg/ml Cm, 20 μg/ml Gm, or 250 μg/ml hygromycin B (Hyg) were added to the plates. For liquid cultures, 10 ml or 15 ml Brucella broth (BB) medium supplemented with 10 μg/ml vancomycin was inoculated with *Campylobacter* cells from plate to a final $OD_{600\,nm}$ of 0.002-0.005. Cultures were grown in 25 $cm^3$ cell culture flasks shaking at 140 rpm until bacteria reached exponential/mid-log phase ($OD_{600\,nm}$ of 0.4-0.5), or back-diluted to a final $OD_{600\,nm}$ of 0.05 in 50 ml fresh BB media (75 $cm^3$ flasks) for further growth as indicated in the text or figure legends. Plates and liquid cultures were incubated at 37 °C in a HERAcell 150i incubator (Thermo Fisher Scientific) providing a microaerobic atmosphere (10% $CO_2$, 5% $O_2$, 85% $N_2$). *C. jejuni* strains were stored at -80 °C in BB media containing 25% glycerol.

For examination of *C. jejuni* sRNA expression over growth in different media, bacteria from overnight cultures were washed three times in 1 x PBS and then inoculated into 50 ml media (BB, MEM, or MEM+Fe). MEM supplemented with 20 mM glutamate as a carbon source and 10 μg/ml vancomycin was used as a defined medium ("MEM"). For iron replete conditions (MEM+Fe), fresh $Fe_2SO_4$ was added to the defined medium at a final concentration of 40 μM based on previous studies[102]. Cultures were grown with shaking at 37 °C until the respective growth phase was reached (lag phase, $OD_{600\,nm}$ ~ 0.1; early exponential phase, $OD_{600\,nm}$ ~ 0.25; exponential phase, $OD_{600\,nm}$ ~ 0.5; and stationary phase, $OD_{600\,nm}$ ~ 0.8) and cells were harvested for RNA in stop mix and frozen in liquid nitrogen.

### Construction of *C. jejuni* deletion mutants
Deletion mutants of *C. jejuni* used in this study were constructed by double-crossover homologous recombination replacing the respective genomic region by an antibiotic resistance cassette. Cassettes used for cloning were either *aphA-3* (Kan[R])[103], *C. coli cat* (Cm[R])[104],

$aac(3)$-$IV$ (Gm[R])[105], or $aph(7'')$ (Hyg[R])[106]. By overlap PCR, these resistance cassettes were fused to ~500 bp homologous sequences up- and downstream of the gene intended to be deleted.

As an example, deletion of the CJnc230 sRNA in *C. jejuni* strain NCTC11168 (CSS-5604) is described in detail. First, approximately 500 bp upstream of the CJnc230 5′ end and ~500 bp downstream of the sRNA were amplified from NCTC11168 WT genomic DNA (gDNA) using CSO-3988 × 3989 and CSO-3991 × 3993, respectively. A polar Kan[R] cassette (*aphA-3*) was amplified from plasmid pGG1[57] using JVO-5068 × HPK2-term. This cassette contained the promoter and terminator sequence from the *H. pylori* sRNA RepG[107]. The antisense oligonucleotide (CSO-3989) for amplification of the CJnc230 upstream region and the sense oligonucleotide (CSO-3991) for amplification of the downstream region included overhangs (32 nt and 21 nt, respectively) to the resistance cassette in order to fuse all three fragments together by overlap PCR. Therefore, the purified PCR amplicons (Macherey-Nagel NucleoSpin Gel and PCR Clean-up Kit) of CJnc230 up- and downstream regions, as well as the polar Kan[R] cassette were mixed in a 20:20:200 ng ratio and amplified using CSO-3988 × 3993 (final concentration 60 nM). After purification, the product was transformed into *C. jejuni* NCTC11168 WT strain (CSS-5295) via electroporation (described below). The resulting CJnc230::*aphA-3* transformants were checked by colony PCR using CSO-3995 × HPK2-term. A similar approach was chosen also for other deletion mutants used in this study, except for the use of non-polar resistance cassettes (amplified with HPK1 × HPK2) when replacing CDSs instead of sRNAs, and oligonucleotides are listed in Supplementary Dataset 8.

## Construction of *C. jejuni* sRNA complementation and over-expression strains

For complementation or overexpression of CJnc170 and CJnc230, a copy of the respective sRNA was either introduced into the *rdxA* (Cj1066) or the Cj0046 pseudogene locus of *C. jejuni*, both of which were previously used for heterologous gene expression in this bacterium[51,108]. For complementation of CJnc230, the sRNA was fused with its 5′ end to the unrelated *metK* (Cj1096c) promoter, as the majority of CJnc230 expression originated from cleavage of the *flgE* mRNA. Strong overexpression of either CJnc170 or CJnc230 was achieved by fusing it to the promoter of the major outer membrane protein (*porA*, Cj1259). Constructs were generated via sub-cloning of existing plasmids (pSE59.1 and pST1.1[47], or pSSv54.3[73]) in *E. coli* TOP10 using homologous regions of the respective *C. jejuni* loci and *C. coli cat* (Cm[R])[104], *aphA-3* (Kan[R])[103], or *aac(3)-IV* (Gm[R])[105] cassettes with promoter and terminator.

As an example, construction of the *C. jejuni* NCTC11168 CJnc230 complementation strain (CSS-6172) is described in detail. The CJnc230 sRNA starting from its 5′ end until 124 bp after its annotated 3′ end[28] was amplified with CSO-4254 × 4257 from NCTC11168 WT (CSS-5295) gDNA and subsequently digested with *Cla*I. The plasmid backbone containing homologous parts to *rdxA*, a Cm[R] (*cat*) cassette, and the *metK* promoter was amplified by inverse PCR with CSO-0347 × 1956 from pSE59.1[47] and also *Cla*I digested. Vector and insert were ligated overnight at 16 °C using T4 DNA ligase (New England Biolabs) and transformed into *E. coli* TOP10. Positive clones were confirmed by colony PCR using CSO-0644 × 4257 and sequencing (Microsynth). The resulting plasmid was named pFK5.4. The purified PCR product, amplified from pFK5.4 with CSO-2276 × 2277, was then transformed into *C. jejuni* NCTC11168 ΔCJnc230 (CSS-5604) via electroporation. Clones were verified by colony PCR using CSO-0644 × 0349 and sequencing. A similar strategy was used to construct CJnc230 or CJnc170 overexpression mutants and respective oligonucleotides can be found in Supplementary Dataset 8. In order to combine existing insertions in the *rdxA* locus (e.g. translational reporters; described below) with sRNA overexpression, plasmids based on pSSv54.3[73] with regions homologous to the Cj0046 pseudogene locus were constructed. Site-directed mutagenesis of the CJnc230 interaction region

was performed by inverse PCR on pFK16.7 and *Dpn*I digestion using primers CSO-5216 × 5217, resulting in pFK37.1.

## Construction of 3xFLAG epitope-tagged proteins in *C. jejuni*

In order to measure protein expression in vivo, the C-terminus of a protein of interest was chromosomally tagged with a 3xFLAG epitope. This was achieved by overlap PCR and is briefly explained for the *C. jejuni* NCTC11168 FlgM-3xFLAG strain (CSS-5798).

For homologous recombination, approximately 500 bp upstream of the *flgM* stop codon and a region containing roughly the last 80 bp and 400 bp downstream of *flgM* CDS including parts of Cj1465 were amplified from NCTC11168 WT (CSS-5295) gDNA using oligonucleotides CSO-4431 × 4432 and CSO-4433 × 4434, respectively. The *aac(3)-IV* cassette (Gm[R]) fused downstream to the 3xFLAG sequence was amplified with primers CSO-0065 × HPK2 from gDNA of PtmG-3xFLAG (CSS-1252)[73]. Oligonucleotides CSO-4432 and CSO-4433 generated overhangs to the 3xFLAG tag 5′ end and the Gm resistance cassette 3′ end, respectively. All three purified fragments were mixed in an equimolar ratio and used for overlap PCR (CSO-4431 × 4434, final concentration 60 nM). The purified PCR product was electroporated into NCTC11168 WT (CSS-5295) and correct insertion in resulting clones was validated by colony PCR using oligonucleotides CSO-4430 × HPK2 and sequencing. Analogous approaches were chosen for C-terminal epitope-tagging of Cj1387c, FlaA, FlaB[57], RpoN, and FliA. The 3xFLAG fused to a kanamycin resistance cassette was amplified from pGG1[57]. Respective oligonucleotides for cloning of epitope-tagged strains can be found in Supplementary Dataset 8.

## Construction of *C. jejuni* sfGFP reporter fusions

For the construction of translational and transcriptional sfGFP reporters in NCTC11168, the 5′UTR or promoter region of interest was fused to an unrelated promoter or 5′UTR (*metK*), respectively. These constructs were linked to *sfgfp*[50] and inserted into the *rdxA* (Cj1066) or the Cj0046 pseudogene locus. Reporter fusions were generated by cloning in *E. coli* TOP10 and/or overlap PCR.

As an example for the overlap PCR approach, construction of the *flgM* (Cj1464) 5′UTR-sfGFP translational fusion in *C. jejuni* NCTC11168 (CSS-6300) is explained in more detail. First, the *rdxA* upstream region for homologous recombination including the *C. coli cat* cassette (Cm[R]) and the *metK* promoter (P*metK*), as well as *sfgfp* (starting from its 2nd codon) with the *rdxA* downstream region were amplified from pKF1.1 using oligonucleotides CSO-2276 × 1956 and CSO-3279 × 2277, respectively. This plasmid contained the *sfgfp* sequence amplified with CSO-3279 × 3717 on pXG10-SF[109], which was ligated into pSE59.1[47] after backbone amplification with CSO-4207 × 0347 and *Cla*I digestion. Positive *E. coli* TOP10 clones were confirmed after transformation by colony PCR using CSO-0644 × 3270 and sequencing. Second, the *flgM* 5′UTR including the predicted interaction region with CJnc230 and its first 10 codons were amplified from NCTC11168 WT (CSS-5295) gDNA using primers CSO-4721 × 4722. These oligonucleotides created overhangs to the P*metK* 3′ end (CSO-4721) or to the *sfgfp* 5′ end (CSO-4722). The resulting products were mixed in an equimolar ratio and used for overlap PCR (CSO-2276 × 2277, final primer concentration 60 nM). The purified PCR product was electroporated into NCTC11168 WT (CSS-5295) and correct integration was confirmed by colony PCR using oligonucleotides CSO-0644 × 0349 and sequencing. A similar approach was followed in order to generate the Cj1387c 5′UTR-sfGFP translational fusion for CJnc230 interaction validation.

Transcriptional reporter fusions were generated by a combination of cloning in *E. coli* TOP10 and subsequent overlap PCR. As example, construction of the *C. jejuni* NCTC11168 P*flaA*-sfGFP strain (CSS-7086) is explained. First, the promoter region of *flaA* (position -171 to +34 relative to TSS) was amplified with CSO-5246 × 5247 from NCTC11168 WT (CSS-5295) gDNA and subsequently digested with *Xma*I. The vector backbone containing flanking regions of the *rdxA* locus for

homologous recombination, the *C. coli cat* cassette (Cm^R), the *metK* 5'UTR (24 nt, including RBS), as well as *sfgfp* was amplified from pKF1.1 using oligonucleotides CSO-2989 × 5096 and similarly digested. Vector and insert were ligated overnight at 16 °C using T4 DNA ligase and transformed into *E. coli* TOP10. Positive clones were confirmed by colony PCR using CSO-0644 × 3270 and sequencing. The resulting plasmid was named pFK38.15. Next, the region containing the *flaA* promoter, *metK* 5'UTR, and *sfgfp* was amplified from pFK38.15 using oligonucleotides CSO-5248 × 5100, thereby generating overhangs to the 3' end of the Gm^R cassette terminator and the Cj0046 flanking region, respectively. The Cj0046 up- and downstream regions for homologous recombination including the Gm^R cassette were amplified from pSSv54.3[73] with CSO-1402 × 0484 and CSO-2751 × 1405, respectively. All three fragments were mixed in an equimolar ratio and used for overlap PCR (CSO-1402 × 1405, final primer concentration 60 nM). The purified PCR product was electroporated into NCTC11168 WT (CSS-5295) and correct integration was confirmed by colony PCR using oligonucleotides CSO-0833 × 3197 and sequencing. A similar approach was used for cloning of transcriptional reporter fusions with CJnc170, CJnc230, and *flgE* promoters. Respective oligonucleotides for cloning of translational and transcriptional sfGFP reporters are listed in Supplementary Dataset 8 and reporter sequences in Supplementary Dataset 9.

## Construction of the FlaA (S395C) mutant for flagella labeling

For imaging of flagellar filaments upon maleimide staining (see below), a *C. jejuni* NCTC11168 mutant bearing a second copy of *flaA* with an introduced cysteine residue (S395C) was generated (CSS-7487). The *flaA* locus including its FliA-dependent promoter was amplified from *C. jejuni* NCTC11168 WT (CSS-5295) gDNA using oligonucleotides CSO-4744 × 4745 and ligated into pGEM-T Easy (Promega) according to the manufacturer's recommendations. Site-directed mutagenesis was performed on the resulting plasmid pFK43.1 by inverse PCR with primers CSO-5344 × 5345 followed by *Dpn*I digestion in order to mutate serine at position 395 to cysteine (S395C). A similar residue was previously used for mutation and subsequent flagellar staining in the related *C. jejuni* strain 81-176[110], as this is assumed to be surface-exposed according to its glycosylation status[111]. After colony PCR and sequencing with primers REV and UNI-61, this plasmid was named pFK44.3 and served as template for amplification of the mutated *flaA* (S395C) locus using oligonucleotides CSO-4744 × 4745 for subsequent overlap PCR. The 5' end of this amplicon had complementarity to the upstream fragment for overlap PCR, containing ~500 bp of the *rdxA* locus and the *C. coli cat* (Cm^R) cassette, while the 3' end of the amplicon overlapped with the downstream fragment of the *rdxA* locus for homologous recombination. The up- and downstream fragments for overlap PCR were amplified from gDNA of the NCTC11168 *flaA* complementation mutant (CSS-6208)[112] with primers CSO-0345 × 0573 and CSO-0347 × 0348, respectively. All three products were mixed in an equimolar ratio and used for overlap PCR (CSO-2276 × 2277, final primer concentration 60 nM). The purified PCR product was electroporated into NCTC11168 WT (CSS-5295) and correct integration was confirmed by colony PCR using oligonucleotides CSO-0643 × 0349 and sequencing.

## Transformation of *C. jejuni* by electroporation

All mutant strains used in this study were generated by double-crossover homologous recombination in the chromosome after introducing DNA via electroporation. Therefore, *C. jejuni* was streaked out from cryostocks on MH agar with vancomycin, passaged once or maximum twice, and bacterial cells were resuspended in cold electroporation solution (272 mM sucrose, 15% (w/v) glycerol). Bacteria were pelleted by centrifugation at 4 °C and 6500 x *g* for 5 min, and cells were washed twice with the same solution. Depending on the pellet size, cells were taken up in 100-300 μl of cold electroporation solution and 50 μl of this suspension was mixed on ice with 200-400 ng of

purified PCR product. Subsequently, the cells were electroporated in a 1 mm gap cuvette (Cell Projects) at 2.5 kV, 200 Ω, and 25 μF for 4-6 ms on a Biorad MicroPulser™ instrument. Afterwards, the electroporation reaction was mixed with 100 μl fresh BB medium at room temperature and transferred onto a non-selective MH agar plate with vancomycin only to allow recovery overnight at 37 °C in the microaerobic incubator. On the next day, cells were restreaked on an MH agar plate containing the appropriate antibiotics for marker selection and incubated microaerobically at 37 °C until colonies were visible (2-5 days). Clones were verified by colony PCR after gene deletion and, in addition, by sequencing after gene insertions in *rdxA* or Cj0046 loci and C-terminal 3xFLAG-tagging.

## Motility assay

*C. jejuni* was grown to mid-log phase (OD_{600 nm} 0.4–0.5) and 1 μl of each bacterial culture was stab-inoculated into a soft agar plate (BB + 0.4% Difco agar). Plates were incubated right-side-up at 37 °C for 24 hrs under microaerobic conditions and, afterwards, the halo radius was measured at three different positions using ImageJ (NIH, USA; v 1.53j), thus averaging the mean swimming distance for each strain on one plate. Motility assays were performed in six biological replicates (six independent cultures stab-inoculated into six different plates) and the average halo radius for each mutant strain was used to compare motility to WT bacteria. Statistical tests as indicated were performed with GraphPad Prism (GraphPad Software, CA, USA; v 9.2.0).

## Transmission electron microscopy and flagellar length measurement

*C. jejuni* NCTC11168 cells were harvested from plate in passage one, gently soaked in PBS and centrifuged for 3 min at 2500 x *g*. Pellets were carefully resuspended in 500 μl 2.5% glutaraldehyde including 0.1 M cacodylate and fixed overnight at 4 °C. After staining with 2% uranylacetate, bacteria were imaged on 300-mesh grids using a JEOL-2100 transmission electron microscope. Flagellar length measurements were performed with ImageJ software (NIH, USA; v 1.53j) and the ridge detection plug-in applying the following settings: line width: 12; high contrast: 150; low contrast: 50; sigma: 3.96; lower threshold: 0.34; upper threshold: 0.85; minimum line length: 100; maximum line length: 6000. For every strain, at least 30 seemingly intact flagella attached to the bacterial cell body were measured. Statistical tests were performed with GraphPad Prism (GraphPad Software, CA, USA; v 9.2.0).

## Maleimide-staining of flagella and confocal microscopy

In order to visualize flagellar filaments by confocal microscopy, *C. jejuni* NCTC11168 strains bearing a cysteine residue in a second copy of the major flagellin (FlaA S395C; described above) were labeled with the DyLight™ 488 maleimide stain (Thermo Scientific), adapted from a previously published protocol[110]. Therefore, bacteria grown on MH agar plates (passage one) corresponding to an OD_{600 nm} of ~1 were gently resuspended in 1 ml PBS and 1 μl of the maleimide-conjugated dye (stock concentration: 10 μg/μl in DMF) was added. After incubation at 37 °C for 30 min, cells were centrifuged for 3 min at 2500 x *g* and the pellet was carefully resuspended in 1 ml of PBS containing 1 μM of the eBioscience™ Cell Proliferation Dye eFluor™ 670 (Thermo Fisher Scientific) to counterstain the cell body. Following incubation at 37 °C for 10 min and subsequent centrifugation, cells were washed once in PBS and then fixed for 2 hrs at room temperature in 4% paraformaldehyde/PBS. After two washes with PBS, samples were stored light-protected in the same buffer at 4 °C until imaging with a Leica TCS SP5 II laser scanning confocal microscope (Leica Microsystems, LAS AF software version 2.7.3.9723) using sequential scanning mode.

## SDS-PAGE and western blotting

*C. jejuni* cells grown to mid-log phase (OD_{600 nm} 0.4-0.5) were collected by centrifugation at 7500 x *g* and 4 °C for 5 min. Pellets were dissolved

in 1 x protein loading buffer (62.5 mM Tris-HCl, pH 6.8, 100 mM DTT, 10% (v/v) glycerol, 2% (w/v) SDS, 0.01% (w/v) bromophenol blue) and boiled for 8 min shaking at 1000 rpm and 95 °C.

For total protein analysis, samples corresponding to 0.1 $OD_{600\,nm}$ were separated on vertical SDS-polyacrylamide gels (10 or 12% PAA) and stained by PageBlue protein staining solution (Thermo Scientific). Pictures of protein gels were taken on an Amersham ImageQuant 800 instrument (GE; 06340134, version 1.2.0). For western blot analysis, protein samples corresponding to an $OD_{600\,nm}$ of 0.05–0.2 were separated by SDS-PAGE (12 or 15% PAA) and transferred to a nitro-cellulose membrane (GE) by semidry blotting. Membranes were blocked for 1 h with 10% (w/v) milk powder in TBS-T (Tris-buffered saline-Tween-20) and incubated overnight with primary antibody (monoclonal anti-FLAG, 1:1000; Sigma-Aldrich, #F1804-1MG; or anti-GFP, 1:1000, Roche #11814460001 in 3% bovine serum albumin (BSA)/TBS-T) at 4 °C. After three washes for 20 min with TBS-T, membranes were incubated for 1 h at room temperature with secondary antibody (anti-mouse IgG, 1:10,000 in 3% BSA/TBS-T; GE Healthcare, #RPN4201) linked to horseradish peroxidase. After washing, chemiluminescence was detected using ECL-reagent (2 ml solution A (0.1 M Tris-HCl pH 8.6 and 0.25 mg/ml luminol sodium salt) and 200 μl solution B (1.1 mg/ml para-hydroxycoumaric acid in DMSO)) with 0.6 μl 30% $H_2O_2$ and imaged on a ImageQuant LAS-4000 device (GE; version 1.3, build 1.3.0.134). A monoclonal antibody against GroEL (1:10,000; Sigma-Aldrich, #G6532-5ML) with an anti-rabbit IgG (1:10,000; GE Healthcare, #RPN4301) secondary antibody was used for normalization after FLAG/GFP detection. Bands were quantified using AIDA Image Analysis Software (v5.0, build 1182, Raytest, Germany). Statistical tests as indicated were performed with GraphPad Prism (GraphPad Software, CA, USA; v 9.2.0).

### Flow cytometry

For single cell analysis of sfGFP reporters in *C. jejuni*, bacteria grown to mid-log phase ($OD_{600\,nm}$ 0.4-0.5) were collected by centrifugation at 7500 x *g* and 4 °C for 5 min. Pellets were resuspended in 500 μl 4% paraformaldehyde/PBS and fixed overnight at 4 °C. After two washes with PBS, cells were resuspended in PBS and 100,000 events per sample were measured on a BD Accuri™ C6 instrument (version 1.0.23.1, build 20151211.23.1), applying a lower cut-off of 2000 for the forward scatter (FSC-H). Analysis was done using FlowJo software (FlowJo, OR, USA; v10) and statistical tests were performed with GraphPad Prism (GraphPad Software, CA, USA; v 9.2.0).

### RNA preparation from bacterial cells

Unless stated otherwise, *C. jejuni* was grown to mid-log phase ($OD_{600\,nm}$ 0.4-0.5) and cells corresponding to an $OD_{600\,nm}$ of ~4 were mixed with 0.2 volumes stop mix (95% EtOH and 5% phenol, v/v), and immediately snap-frozen in liquid nitrogen. Frozen cell pellets stored at −80 °C were thawed on ice, spun down for 20 min at 4500 x *g* and 4 °C, and resuspended in 600 μl Tris-EDTA buffer (pH 8.0) containing 0.5 mg/ml lysozyme and 1% SDS. Cells were lysed upon incubation at 64 °C for 2 min and, afterwards, total RNA was extracted using the hot phenol method as described previously for the related Epsilonpro-teobacterium *Helicobacter pylori*[113]. After RNA isolation, residual DNA was removed by treatment with DNase I (Thermo Scientific) according to the manufacturer's instructions.

### Northern blot analysis

For RNA expression analysis, 5–10 μg of total RNA in Gel Loading Buffer II (GLII, Ambion; 95% (v/v) formamide, 18 mM EDTA, and 0.025% (w/v) SDS, xylene cyanol, and bromophenol blue) was separated on 6% PAA/7 M urea denaturing gels in 1 x TBE buffer. Afterwards, RNA was transferred to Hybond-XL membranes (GE) by electroblotting and cross-linked to the membrane with UV light. Then, $\gamma^{32}$P-ATP end-labeled DNA oligonucleotides in Roti Hybri-quick (Roth) were

hybridized overnight at 42 °C. After washing at 42 °C for 20 min each in 5 x, 1 x, and 0.5 x SSC (saline-sodium citrate) + 0.1% SDS, the membrane was dried and exposed to a PhosphorImager screen. Screens were scanned using a Typhoon FLA-7000 Series PhosphorImager (GE; version 1.3, build 1.3.0.105). Bands were quantified using AIDA Image Analysis Software (v5.0, build 1182, Raytest, Germany).

### Primer extension analysis

For identification of CJnc230 sRNA 5′ ends, 4 μg of DNase I-treated RNA from *C. jejuni* NCTC11168 WT and mutant strains was used for primer extension. RNA was diluted in $H_2O$ to a total volume of 5.5 μl, denatured, and snap-cooled on ice. The 5′-radiolabeled DNA oligonucleotide complementary to CJnc230 (CSO-0537) was added and annealed by heating to 80 °C and slow cooling to 42 °C. Afterwards, 3.5 μl of master mix containing reverse transcriptase (RT) buffer, dNTPs (1 mM final concentration) and 20 U Maxima RT (Thermo Scientific) was added and RNA was reverse transcribed for 1 h at 50 °C. Reactions were stopped with 10 μl GLII. A sequencing ladder was generated using the DNA Cycle Sequencing kit (Jena Bioscience) according to the manufacturer's recommendations with the CJnc230 sRNA region amplified from NCTC11168 WT (CSS-5295) gDNA using oligonucleotides CSO-3995 × 3993. The radioactively labeled CSO-0537 was also used for ladder preparation and reactions were stopped with GLII. Samples were separated on 6% PAA/7 M urea sequencing gels, which were dried afterwards and exposed to a PhosphorImager screen. Screens were scanned using a Typhoon FLA-7000 Series PhosphorImager (GE; version 1.3, build 1.3.0.105).

### Reverse transcription-polymerase chain reaction (RT-PCR)

For determination of *flgE*-CJnc230 co-transcription, DNase I-digested RNA samples of *C. jejuni* NCTC11168 WT and RNase deletion strains were reverse transcribed using the High Capacity cDNA Reverse Transcription kit (Thermo Fisher Scientific). One microgram of RNA was denatured, snap-cooled on ice, and reactions were started with or without (+/-) MultiScribe™ RT using the following cycling conditions: 25 °C, 10 min; 37 °C, 2 hrs; 85 °C, 5 min. Afterwards, PCR was performed using gene-specific oligonucleotides for CJnc230 sRNA (CSO-4254 × 5138), as well as primer sets for amplification of CJnc230 together with the upstream gene *flgE* or with the downstream gene *metB* (CSO-5342 × 5138 or CSO-4254 × 5341, respectively). All three genes together were amplified using oligonucleotides CSO-5342 × 5341, and CJnc230 with 5′- or 3′-overhangs using primers CSO-4255 × 5138 or CSO-4254 × 4297, respectively. PCR on *C. jejuni* NCTC11168 WT (CSS-5295) gDNA served as positive control. Amplicons were visualized on 1-2% agarose gels, dependent on the expected fragment lengths.

### Circular rapid amplification of cDNA ends (cRACE)

To simultaneously map CJnc230 transcript 5′ and 3′ ends in *C. jejuni* NCTC11168 WT and the Δ*rny* mutant, Rapid Amplification of cDNA Ends (RACE) after self-ligation of RNA transcripts was performed.

First, 5 μg of DNase I-digested RNA was taken up in a total volume of 16.5 μl $H_2O$, denatured, and snap-cooled on ice. A mix of 2 μl 10 x Antarctic Phosphatase buffer (NEB), 1 μl Antarctic Phosphatase (5 U; NEB), and 0.5 μl Ribonuclease Inhibitor (10 U; Molox) was added and incubated for 1 h at 37 °C to dephosphorylate the RNA. The reaction was then made up to 100 μl and extracted with an equal volume of phenol:chloroform:isoamyl (PCI) alcohol in a Phase-Lock gel tube (5PRIME). Afterwards, RNA was precipitated with 1 μl GlycoBlue™ (Thermo Fisher Scientific) and 2.5 volumes 30:1 mix (100% EtOH:3 M sodium acetate, pH 6.5). The RNA was dissolved in 11.5 μl $H_2O$ and 8.5 μl of a mix containing 2 μl 10 x T4 RNA ligase buffer (NEB), 3 μl DMSO, 2 μl ATP (1 mM final concentration; NEB), 1 μl T4 RNA ligase 1 (10 U; NEB), and 0.5 μl Ribonuclease Inhibitor (10 U; Molox) was added. Following ligation at 37 °C for 30 min, RNA was extracted with PCI and pre-cipitated as described above. Then, it was dissolved in 30 μl $H_2O$ and

10 µL of circularized RNA, together with 2 µl dNTPs (1 mM final concentration) and 1 µl reverse transcription (RT) primer (CSO-0537; 5 µM final concentration), were denatured and snap-cooled on ice. The RT reaction was performed with 4 µl 5 x RT buffer (Thermo Scientific), 1 µl Ribonuclease Inhibitor (20 U; Molox), 1 µl 0.1 M DTT, and 1 µL Maxima RT (200 U; Thermo Scientific) for 5 min at 50 °C, 1 h at 55 °C, and 15 min at 70 °C. RNA was removed by digestion with 1 µl RNase H (5 U; NEB) for 22 min at 37 °C.

One microliter of this reaction was used as a template for PCR using *Taq* DNA polymerase (NEB) with oligonucleotides CSO-0537 × 5192 and 3% DMSO. Cycling conditions were as follows: 95 °C for 5 min and 35 cycles of [95 °C, 10 s; 56 °C, 30 s; 72 °C, 60 s], followed by 72 °C for 10 min. Amplification was checked on an 8% PAA gel, reactions were cleaned up with the NucleoSpin Gel and PCR Clean-up kit (Macherey-Nagel), and ligated into pGEM-T Easy (Promega) according to the manufacturer's instructions. For the WT background, the inserts of sixteen white clones were sequenced, and for Δ*rny* bacteria, the inserts of twenty-one white clones were sequenced with primers REV or UNI-61.

### In-vitro transcription and 5' end-labeling of RNAs
DNA templates containing the T7 promoter sequence were generated by PCR with S7 Fusion DNA polymerase (MOBIDING) using oligonucleotides listed in Supplementary Dataset 8. In-vitro transcription of RNAs by T7 RNA polymerase was carried out using the MEGAscript™ T7 kit (Thermo Fisher Scientific) according to the manufacturer's instructions and RNA quality was checked by electrophoresis on 6 or 10% PAA/7 M urea gels. Afterwards, transcripts were dephosphorylated with Antarctic Phosphatase (NEB), 5' end-labeled ($\gamma^{32}P$) with polynucleotide kinase (PNK; Thermo Scientific), and purified by gel extraction as described previously[114]. Sequences of the resulting T7 transcripts and DNA templates are listed in Supplementary Dataset 10.

### In-line probing
In order to determine RNA structure and binding interactions, in-line probing assay[54] were carried out as described previously[107]. Five prime end-labeled CJnc230 (0.2 pmol, 20 nM final concentration) in the absence or presence of 2, 20, or 200 nM of *flgM* or Cj1387c mRNA leaders was incubated in 1 x in-line probing buffer (50 mM Tris-HCl, pH 8.3, 20 mM $MgCl_2$, and 100 mM KCl) for 40 hrs at room temperature to allow spontaneous cleavage. RNA ladders were prepared using alkaline hydrolysis buffer (OH ladder) or sequencing buffer (RNase T1 ladder) according to the manufacturer's instructions (Ambion). Reactions were stopped on ice with an equal volume of 2 x colorless loading buffer (10 M urea and 1.5 mM EDTA, pH 8.0), separated on 10% PAA/7 M Urea sequencing gels, and products were visualized after drying and exposure to PhosphorImager screens using a Typhoon FLA-7000 Series PhosphorImager (GE; version 1.3, build 1.3.0.105).

### In-vitro translation assay
Reporter fusions of sRNA targets were in-vitro translated in the absence and presence of sRNA using the PURExpress® In Vitro Protein Synthesis kit (NEB). Four picomoles of in-vitro transcribed 5'UTR reporters (including the RBS/sRNA interaction site and the first 10 codons of *flgM*, Cj1387c, or *flaA* fused to *sfgfp*; Supplementary Dataset 10) were denatured in the absence or presence of 20, 80, or 200 pmol of CJnc230 sRNA for 1 min at 95 °C and cooled on ice for 5 min. A *flaA* reporter was used as a negative control. The mRNA-sRNA mixture was pre-incubated for 10 min at 37 °C before addition of the kit components, and translation was performed at 37 °C for 30 min. Reactions were stopped with an equal volume of 2 x protein loading buffer. Fifteen microlitres were loaded on 12% PAA gels and protein expression was analyzed by western blotting with an antibody against GFP as described above. After blotting, residual protein in the gel was stained with PageBlue (Thermo Scientific) as a loading control.

### RNA-RNA interaction predictions
Predictions were computed genome-wide with IntaRNA[53] version 3.2.0 (linking Vienna RNA package 2.4.14) using default parameters, except for seed size of 6 nt. The 3'-truncated (88 nt) and full-length (98 nt) version of *C. jejuni* NCTC11168 CJnc230 were used as input and NC_002163.1 as genome accession number for strain NCTC11168.

### Total RNA sequencing (RNA-seq)
*C. jejuni* NCTC11168 WT and *rny* deletion strains were grown in biological triplicates in BB media to mid-log phase and cells were harvested for RNA isolation and subsequent DNase I digestion. Ribosomal RNA depletion and cDNA library preparation were performed at Vertis Biotechnologie AG, Germany. The adapter ligation protocol for construction of cDNA libraries is briefly summarized. After ultrasound fragmentation, an adapter was ligated to RNA 3' ends and reverse transcription was performed, followed by ligation of 5' Illumina TruSeq sequencing adapters to the 3' ends of the antisense cDNAs. The resulting cDNA was PCR-amplified, purified with the Agencourt AMPure XP kit (Beckman Coulter Genomics), and analyzed by capillary electrophoresis. Samples were sequenced on an Illumina NextSeq 500 platform in single-end mode.

After sequencing, demultiplexing of reads was performed using bcl2fastq (Illumina; version: 2.20.0.422). Reads were quality and adapter trimmed with Cutadapt (versions: 1.17 and 2.5)[115] using a cutoff Phred score of 20 in NextSeq mode. READemption (version: 0.4.5)[116] was used to align all reads longer than 11 nt to the reference genome of *C. jejuni* NCTC11168 (NCBI accession number: NC_002163.1; Refseq assembly accession GCF_000009085.1) using segemehl (version: 0.2.0)[117] with an accuracy cutoff of 95%. Coverage plots representing the number of mapped reads per nucleotide were also generated by READemption and normalized for sequencing depth. NCBI gene annotations were complemented with annotations of previously determined 5'UTRs and sRNAs as described previously[33] and READemption was used to quantify aligned reads overlapping genomic features by at least 10 nt. Finally, differential gene expression analysis in replicates was performed by DESeq2 (version: 1.24.0)[118] without fold-change shrinkage. Modeling of batch effects between experiments in the regression step was conducted by including a batch variable in the design formula. Genes with a log2FC ≥ |1| and an adjusted (Benjamini-Hochberg corrected) *p*-value ≤ 0.05 were considered as differentially expressed.

### Differential RNA sequencing (dRNA-seq)
For comparative TSS analysis in *C. jejuni* NCTC11168 WT and RNase III-deficient bacteria, bacterial total RNA samples were harvested at mid-log phase after growth in rich BB medium. After digestion of residual genomic DNA (gDNA) by DNase I treatment, the sample was equally divided into two parts and Terminator 5'-phosphate-dependent exonuclease (TEX) (Epicentre) was used to deplete processed transcripts in one half as previously described[113]. Libraries for Solexa sequencing (HiSeq) were constructed at Vertis Biotechnologie AG, Germany, as previously described[119], but without RNA fractionation prior to cDNA synthesis. In brief, equal amounts of RNA were poly(A)-tailed using poly(A) polymerase and 5'-triphosphate groups were digested with tobacco acid pyrophosphatase (TAP). Then, an RNA adapter was ligated to the 5' phosphate and first-strand cDNA synthesized with an oligo(dT)-adapter primer and M-MLV (Moloney Murine Leukemia Virus) reverse transcriptase. A high-fidelity DNA polymerase was used to PCR-amplify the resulting cDNA to a concentration of ~20-30 ng/µl. Finally, cDNA was purified with the Agencourt AMPure XP kit (Beckman Coulter Genomics), analyzed by capillary electrophoresis, and sequenced on an Illumina HiSeq 2000 platform in single-end mode.

The FASTQ-format reads were trimmed with a cutoff Phred score of 20 using Cutadapt (version: 4.1)[115]. After poly(A)-tails were removed,

sequences shorter than 12 nt were omitted and the remaining reads mapped to the *C. jejuni* NCTC11168 reference genome (NCBI accession number: NC_002163.1, RefSeq assembly accession number: GCF_000009085.1) using READemption (version: 2.0.1)[116] and sege-mehl (version: 0.3.4)[117] with an accuracy cutoff of 95%. Coverage plots containing the number of mapped reads per nucleotide were also generated with READemption and visualized in the Integrated Genome Browser[120]. The coverage was normalized by the total number of aligned reads of a given library and multiplied by the minimum number of aligned reads calculated over all libraries. Gene annotations were retrieved from NCBI and extended as described for total RNA sequencing experiments.

### Termination site sequencing (term-seq)
In order to determine transcript 3′ ends on a global scale in *C. jejuni* strain NCTC11168, WT and *rnc*, *rny*, *pnp*, and *rnr* deletion mutant bacteria were harvested in biological duplicates for RNA isolation, followed by DNase I digestion. Ribosomal RNAs were depleted using the RiboCop™ rRNA Depletion Kit for Gram Negative Bacteria (Lexogen). Afterwards, cDNA libraries were constructed at Vertis Bio-technologie AG, Germany, using the 3′Term protocol. First, the 5′ Illumina sequencing adapter was ligated to the 3′-OH of the RNA molecules and cDNA synthesis was performed using M-MLV reverse transcriptase. Then, first-strand cDNA was fragmented and the 3′ Illu-mina sequencing adapter was ligated to the 3′ ends of the single-stranded cDNA fragments. Finally, the 3′ cDNA fragments were PCR-amplified and purified as described for total RNA-seq libraries above. Sequencing was performed using an Illumina NextSeq 500 platform in single-end mode.

Processing of sequencing reads, alignments, and coverage calcu-lation were performed as described for total RNA-seq experiments. Then, the READemption pipeline (version: 0.4.5)[116] was applied to generate two kinds of positional coverage files: default total coverage based on full-length alignments and last base coverage mapping of only the 3′-end base of each alignment. In both cases, sequencing depth-normalized plots were used for visualization in the Integrated Genome Browser[120].

### Conservation analysis and multiple sequence alignments
Putative CJnc230 homologs in different *Campylobacter* species were identified by blastn[121], GLASSgo[37], and manual inspection of the inter-genic region between Cj1729c (*flgE*) and Cj1727c (*metB*) homologs. Putative CJnc230 regions downstream of Cj1729c (*flgE*) homologs were retrieved from KEGG (Kyoto Encyclopedia of Genes and Genomes) and used for subsequent alignment with MultAlin[122].

### Statistical analysis
All data for western blot, motility assay, flow cytometry, and growth behavior are presented as mean ± standard deviation. Flagellar length measurements are depicted as Violin plots including median and quartiles. Exact sample sizes of each experiment can be found in respective figure legends or in the main text. For statistical analysis, a two-tailed unpaired Student's *t*-test was used.

### Reporting summary
Further information on research design is available in the Nature Portfolio Reporting Summary linked to this article.

## Data availability
The total RNA-seq, dRNA-seq, and term-seq datasets have been deposited at the NCBI Gene Expression Omnibus (GEO; https://www.ncbi.nlm.nih.gov/geo/)[123] under the accessions GSE230835, GSE230836, and GSE230837, respectively. The reference genome sequence NC_002163.1 (ASM908v1) and annotation was recovered from NCBI (2014-03-20). The sRNA and 5′UTR annotation was generated from

published differential RNA-seq data retrieved from GEO (accession GSE38883). All other data supporting the findings of this study are available within the article and its supplementary files. Source data are provided with this paper.

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

## Acknowledgements

This work was supported by the DFG Research Training Group GRK2157 "3D-Infect" (Deutsche Forschungsgemeinschaft; www.dfg.de), a "CampyRNA/031L0003A" junior consortium grant within the 2nd call of Infect-ERA (ERA-NET; www.infect-era.eu)/Bundesministerium für Bildung und Forschung (BMBF; www.bmbf.de), and a "StressRegNet" grant within the Bavarian Research Network (bayresq.net; www.bayresq.net) awarded to C.M.S.. We thank Gaurav Dugar, Elisabetta Fiore, Kirsten Kucka, Ke Ma, and Sahil Sharma for help with cloning of ribonuclease and transcription factor deletion strains of *C. jejuni* NCTC11168, as well as the FlaB- and RpoN-3xFLAG strains. We thank Gaurav Dugar for performing dRNA-seq experiments with NCTC11168 wild-type and Δ*rnc* strains, members of the Core Unit Systems Medicine (University of Würzburg and University Hospital Würzburg) for support with deep sequencing and bioinformatic analysis, and Johannes Kullmann for help with mapping the dRNA-seq data. We thank the Imaging Core Facility at the Theodor-Boveri-Institute of Bioscience of the University of Würzburg for help with electron microscopy and Sharma lab members for critical comments and discussion on the manuscript.

## Author contributions

F.K. & C.M.S. designed the experiments; F.K. & S.L.S. performed the experiments; F.K., S.L.S. & C.M.S. analyzed data; F.K., S.L.S. & C.M.S. wrote the manuscript; C.M.S. provided resources, acquired funding, and supervised the project.

## Funding

## Competing interests

The authors declare no competing interests.
