## [Peer Review File · Nature Communications]

Interplay of two small RNAs fine-tunes hierarchical flagella gene expression in *Campylobacter jejuni*REVIEWER COMMENTS

Reviewer #1 (Remarks to the Author):

This is an interesting and well-written paper that describes how two *C. jejuni* sRNAs influence flagella length and cell motility. One of the strengths of this manuscript is the high standard of the presented data. The amount of effort put into validating the translational repression of mRNA targets by CJnc230 was particularly impressive, and I cannot praise it enough. I appreciated that the authors employed a variety of approaches to test their hypotheses and validate their findings. Their thoroughness in conducting the study sets a high standard.

However, there are some possible weaknesses worth mentioning. All the phenotypically relevant results were obtained using sRNA overexpression strains, showing a 10-30-fold increase, while the sRNA deletions did not significantly affect flagellar biosynthesis. This raises the question of whether these sRNAs are expressed at such levels under physiologically relevant conditions. It would be valuable to explore in more detail the conditions under which these sRNAs are naturally expressed. Additionally, in some cases, the effects observed were rather small. Although not a major concern for me, the manuscript could be further improved by discussing these potential pitfalls in more detail.

Overall, this paper makes a significant contribution to the field, and the authors' diligent work and comprehensive approach set an example for future research.

Below I provide some additional suggestions on how the manuscript could be further improved.

Comments:

1. Lines 131 and 132: When mentioning RNase III as a regulator of sRNA processing, it would be prudent to cite the McKellar et al. RNase III CLASH paper, which was published back-to-back with the Mediati et al. manuscript.
2. It appears that Extended Data Fig. 10 is referenced before Extended Data Fig. 9.
3. Figure 4b: Given the degree of over-expression of CJnc230 in this experiment, the impact on motility is modest, as the authors state in the manuscript. The experiments have been repeated three times. However, considering that the values for the three datapoints in these experiments are very close to each other, it could be assumed that these are technical replicates. The legend states three independent replicates, but how this is defined is not clear. Three independent experiments (different days with different cultures) would be preferred. As shown in Fig. 4a, (n=8), I would recommend including more replicates as smaller differences are more convincing if supported by multiple replicates.
4. Figure 4d: To make a statement that you observed a larger portion of flagella with two turns in the OE mutant, it would be necessary to provide quantification of the imaging data. Fig. 4c is convincing and sufficient to make a claim about flagellar length, so perhaps this figure could be removed in case the quantification is challenging.
5. Fig. 6a: Again, n=3, and in some cases, the differences were small. This would also benefit from additional replicates. It would have been helpful to include a CJnc230 overexpression positive control in this experiment.
6. Fig. 6a: The observation that overexpressing CJnc230 in the CJnc170 background does not show an increased motility is surprising. I would have expected that the increase in motility would be more significant compared to just overexpressing CJnc230.

Reviewer #2 (Remarks to the Author):

I read with significant interest this manuscript which overall is an elegant work that attempts to dissect the molecular mechanisms of the role two sRNAs might play in the key virulence attribute of flagellation and motility in *C. jejuni*. Overall, the manuscript is very well written and the science is sound; there are several areas primarily in the interpretation of the data that would benefit from further clarification. I look forward to reviewing a revised version of the manuscript following the authors addressing the comments and requested edits below.

Specific comments:

Line 80-81 – After reading the entire manuscript, this statement may be too strong. These sRNAs certainly have the potential to impact a major virulence trait, however, the data does not support a direct strong link as only significant overexpression could demonstrate any changes and no further work associated with colonization or virulence was performed. Please modify accordingly.

Line 171 – Should this say ~89-98 nt for the longer transcripts as 88 is the same as the most abundant?

Extended data Fig 1a - How would the different downstream genes in 81116 and IA3902 affect the longer length transcripts that are demonstrated? Do you think processing differs in these strains for the 3' end given that you demonstrated the co-transcription with *metB* downstream? Does this affect the activity of the sRNA in those strains of *C. jejuni*?

Extended data Fig 5a – Label for *Cjnc230* is unclear, please adjust.

Extended data Fig 5c – Please clearly state that the labels for each lane are the same as 5b, or relabel again (this would be ideal).

Extended data Fig 6 – What timepoint was used for the protein analysis from the growth curve? OD and CFU/mL can differ for *C. jejuni* and affect the evaluation of growth phases. Was CFU/mL collected as well or just OD?

Extended data Fig 10 comes before extended data Fig 9 in text, this should be corrected.

Figure 4b and Extended Fig 10 – Please specify here what time point was this data collected (I see later that 24 hrs is listed in the Methods but would be helpful to the reader to have available here). It appears that the 11186 isolate that was used is quite low motility to begin with compared to other papers (ex. ref #30, 87) which also utilized 0.4% agar. Has the laboratory isolate utilized been further characterized? It is well documented in the literature that acquired mutations over time in laboratory strains of *C. jejuni* can affect motility.

Lines 376-381 – This section seems unclear – if *flgM* deletion does not lead to altered flagellar length, how can an argument be made that *Cjnc230* mediated repression of *flgM* is the cause of the observed altered flagellar length when *Cjnc230* is mutated? Please clarify further.

Lines 443-449 – Can the authors provide a better explanation for both deletion and overexpression

of CJnc170 returning OE CJnc230 to WT levels, especially as flagellar length doesn't change? It looks like there are clearly additional effects of CJnc230 beyond FlgM control that lead to what is seen. Also, is line 450-451 correct? Flagellar length was not correlated with the OE CJnc170 mutant in combination with OE and deletion of CJnc230. Maybe this should specify single knockout mutants?

Line 475-478 – Is stating that CJnc230 increases motility and CJnc170 decreases appropriate? This was only observed with significant overexpression of both sRNAs? Perhaps this should be clarified by stating that regulation by these sRNAs has the potential to affect motility with high levels of expression, the conditions in which this naturally occurs are unclear. Have you explored or considered exploring what conditions might lead to this extremely high of overexpression (25-30x) (ie – oxidative stress, temperature changes, iron levels, etc)? Are there other putative promoters (Fur, CosR, PerR, etc) that might help explain a circumstance where expression might increase? Have you explored how these sRNAs affect in vivo measures of virulence either in tissue culture or animal models? Have other studies that have reported altered in vivo expression of *C. jejuni* sRNAs reported increased expression of CJnc230 that might help bolster the argument for a strong physiologic role in vivo? It appears that *C. jejuni* can exhibit normal motility without CJnc230 and while it may fine tune the process as aptly suggested in the manuscript title CJnc230 does not appear to be a critical regulator of this process at least under the conditions studied.

General comments:

NCTC11168 has 2 annotations of flgE (Cj1729c and Cj0043) that differ somewhat in sequence –does the affect the analysis at all?

The CJnc230 story is a very elegant explanation of one mechanism of interaction, however, the CJnc170 story is a bit harder to follow and not as well developed, partly because some of this work was done by another lab. Given that a lot of this manuscript focuses on the processing of CJnc230, does CJnc170 require any processing, or is it expressed in active form? It would be good to clarify this for the reader. Where is CJnc170 located on the genome (near other flagellar genes?), what is its length and structure (consider placing in supplemental)? Has CJnc170 also been confirmed to be non-coding via ribosome profiling (ref #33)? There is also a discussion missing on why the previous lab was unable to demonstrate a change in motility (ref#30 states the overexpression mutants were generated and was successful for CJnc170 - was it a failure to overexpress at high enough levels?). This discussion should be added to the paper.

PmtG is on the list of potential targets for CJnc230 – your previous work has shown this to be regulated by CJnc190/CJnc180 – do you believe CJnc230 also plays a role in regulation of this gene? This is also flagellar modification gene - how might that affect motility?

Was autoregulation of the flgE/CJnc230 transcript by CJnc230 investigated as flgE is listed on the IntaRNA list as a potential binding partner?

How is regulation of Cj1387c by CJnc230 related to the rest of the story? The authors demonstrate an interaction, but then do not seem to tie it back to the phenotypes of interest nearly as well as for flgM and it is not included in the final model in Figure 6? There is only a brief mention of this in the Discussion (lines 537-547). I struggle a bit with the inclusion of that data – on one hand it provides evidence of an additional target that might be tied to the phenotype of interest, but because this

connection isn't explored further it detracts somewhat from the rest of the story.

Recent publications have shown a more substantial physiologic role for additional sRNAs (Cjnc110, Cjnc140) in controlling motility in *C. jejuni* albeit a molecular mechanism for motility control for these sRNAs has not been established. The authors might consider an updated literature review and a brief discussion on the potential for a larger regulatory circuit to control flagellar motility using sRNAs in *C. jejuni* which might be beneficial to the reader. Lines 459-461 in Results hint at this but this is not discussed further in the discussion. Have you looked at screening other sRNAs to look for control via RpoN that could help explain some of the missing data for this regulatory cascade? If so, this could be helpful to report here as well.

Reviewer #3 (Remarks to the Author):

In this important study, the authors characterize two sRNAs (FlmE and FlmR) that regulate flagellar filament assembly and motility in *C. jejuni*, adding another level of regulation to a pathway that is already tightly controlled. The authors thoroughly study the generation of FlmE from a downstream region to flgE. The results of this study document positive effects of FlmE overexpression on flagella synthesis and bacterial motility, while opposite effects are presented for FlmR sRNA overexpression. This interplay between FlmE and FlmR results in an interesting fine tuning of flagella biosynthesis. The manuscript is well-written and easy to read. The experiments are well executed and of high quality, although some of the effects resulted from the manipulations of sRNAs levels are modest. Below are some comments and suggestions that could improve the manuscript:

Major comments:

1. While statistically significant, many of the effects resulted from the sRNAs overexpression or deletions are modest. The authors should better explain why these effects are still of biological significant for the bacteria. This can be done by revision of the text or by additional experiments that will support their importance.
2. Pages 5-6, lines 126-151: In figure 2A it does not seem that Cjnc230 is affected by PNPase deletion while in Fig 2B we do see an effect in that strain. Can the authors clarify this? The effect of rny/pnp double deletion is very strong. Is this the first time they are documented to cooperate?
3. Page 6, lines 152-166: Can the authors elaborate on the allegedly contradicting results of Fig 2A and 2C?
4. Page 11, lines 314-322: The experiments that were done to support direct base-pairing between Cjnc230 were all done in-vitro, is there a reason that the authors didn't try the mutagenesis in-vivo in the translational reporter assay? Also, can the authors comment on why they have not tried to complement and restore the interaction between the sRNA and the targets? (e.g. testing both M1 mutants together)
5. Page 12, lines 344-347: The results presented in Fig 4A are not convincing. It seems that only in 50% of the cases FlaA levels increase upon CJ230 overexpression. Can the authors address it or repeat on the experiment?
6. Page 15, lines 444-449: Can the authors explain and discuss how the results of the single and double deletion strains fit the rest of the data presented in the manuscript?
7. The timing is very important in flagella biosynthesis. One may hypothesize that FlmE will be more

important earlier in flagella synthesis while FlmR will be important later on when the synthesis is no longer required. Additional experiments testing if FlmE and FlmR are affecting the timing of flagella synthesis can be a nice addition to this work.

8. In this work the authors carried out global 5' end and 3' end mapping by RNA-seq based methodologies. Uploading these data to online genome browsers will be extremely valuable for the scientific community.

Minor comments:

1. Page 2, lines 46-56: In many gram-negative bacteria (e.g. *E. coli*, *Salmonella*) there is a mixed class 2 + class 3 genes in flagella biosynthesis. Does this mixed class exist in *C. jejuni*? Please refer to it in the text.
2. Page 4, lines 106-109: This sentence is unclear, please revise.
3. I find it confusing that the authors use CJnc230 and CJnc170 names in most places in the text and then transition to FlmE and FlmR, respectively. Consider renaming these two sRNAs FlmE and FlmR from the beginning of the manuscript.
4. As the authors mention in their discussion, a recent paper (<https://doi.org/10.7554/eLife.87151.1>) describes a similar type of fine-tuning of flagella biosynthesis by two sRNAs (MotR and FliX) in *E. coli*. In both cases, one sRNA has a positive role and one sRNA has a negative role. It can be valuable to discuss in more detail the similarities and the differences between the regulation in *E. coli* and *C. jejuni* by these analog sRNAs.

Point-to-point reply NCOMMS-23-30341-T:

We thank the three reviewers for their positive feedback and comments on our manuscript. In the enclosed detailed point-by-point response to the reviewers' comments, we have addressed the specific suggestions and points raised by the referees. In particular, we have i) investigated the expression of flagella sRNAs during growth in more detail, ii) added additional biological replicates to motility assays to strengthen our observations, and iii) revised manuscript text and softened conclusions as suggested.

Answers to the reviewers' comments:

Reviewer #1 (Remarks to the Author)

This is an interesting and well-written paper that describes how two *C. jejuni* sRNAs influence flagella length and cell motility. One of the strengths of this manuscript is the high standard of the presented data. The amount of effort put into validating the translational repression of mRNA targets by CJnc230 was particularly impressive, and I cannot praise it enough. I appreciated that the authors employed a variety of approaches to test their hypotheses and validate their findings. Their thoroughness in conducting the study sets a high standard.

However, there are some possible weaknesses worth mentioning. All the phenotypically relevant results were obtained using sRNA overexpression strains, showing a 10-30-fold increase, while the sRNA deletions did not significantly affect flagellar biosynthesis. This raises the question of whether these sRNAs are expressed at such levels under physiologically relevant conditions. It would be valuable to explore in more detail the conditions under which these sRNAs are naturally expressed.

Additionally, in some cases, the effects observed were rather small. Although not a major concern for me, the manuscript could be further improved by discussing these potential pitfalls in more detail.

Overall, this paper makes a significant contribution to the field, and the authors' diligent work and comprehensive approach set an example for future research.

Below I provide some additional suggestions on how the manuscript could be further improved.

We thank this reviewer for the very positive and endorsing comments on our manuscript. We have added several text clarifications as well as additional experiments and replicates to address the points listed above.

To address whether the sRNAs are expressed at levels comparable to the overexpression under native conditions, we have performed additional experiments to test for expression changes (see **Rebuttal Fig. 1 below**). To check for potential conditions and additional regulators that affect their expression, we measured levels of CJnc230 and CJnc170 at different growth phases in rich and minimal media (**Rebuttal Fig. 1a**) and in regulator deletion mutants by northern blotting (**Rebuttal Fig. 1b**). Both sRNAs were slightly upregulated in lag phase in minimal media (MEM) compared to rich Brucella broth (BB). They showed decreased

abundance at stationary phase, which could be rescued by iron supplementation (MEM+Fe) (**Rebuttal Fig. 1a**). In contrast to strong control by the flagellar regulators (**Manuscript Fig. 1b**), neither sRNA was substantially affected by deletion of several transcriptional regulators (**Rebuttal Fig. 1b**).

Thus, at this point we assume that they are mainly controlled in a temporal manner along the flagellar regulatory cascade rather than by environmental signals. Unlike, *e.g.*, *Salmonella*, *Vibrio* sp. lateral flagella that increase in number during, *e.g.*, swarmer cell differentiation, *C. jejuni* only ever has at most one flagellum at each end. In the aforementioned gammaproteobacteria, it makes sense that environmental signals might feed into the cascade to increase or fine-tune flagellar number. This is further supported by the fact that the *C. jejuni* sRNA expression is controlled by two different flagellar sigma factors that are themselves part of the cascade, regulating expression of middle and late flagellar genes, and also controlling FlgM, which times the cascade. Thus, the sRNAs are likely expressed in a hierarchical temporal manner during flagellar biogenesis. Moreover, we would like to emphasize that while sRNA single deletions did not have a strong effect on flagellar biogenesis under the tested conditions, the combination of CJnc170 overexpression with CJnc230 deletion unmasked a motility phenotype, thus supporting our hypothesis that the two sRNAs fine-tune the cascade, in opposite directions (**Manuscript Fig. 6a**).

We would also like to point out that overexpression is a common means to identify targets and phenotypes in other bacteria, such as Gammaproteobacteria, as sRNA single deletion mutants often have little or no phenotype (Goldberger et al., 2021). For example, a recent study of flagella-related sRNAs in *E. coli* also required overexpression to observe phenotypes for these sRNAs (Melamed et al., 2023). Moreover, our data on CJnc230 and *flgE* expression (**Manuscript Fig. 5a and Extended Data Fig. 5a**) indicate that there is strong feedback control in the system, necessitating the performed uncoupling of CJnc230 from the cascade by overexpression from a sigma-70 promoter.

a**b**
Rebuttal Figure 1. Northern blot analyses of Cjnc230 and Cjnc170 expression. **a**, Total RNA of *C. jejuni* NCTC11168 wildtype (WT) grown in rich Brucella broth (BB) or minimal media (MEM) +/- iron (Fe) was harvested at different growth phases: lag phase, $OD_{600\text{ nm}} \sim 0.1$ (lag); early exponential phase, $OD_{600\text{ nm}} \sim 0.25$ (early exp); exponential phase, $OD_{600\text{ nm}} \sim 0.5$ (exp); and stationary phase, $OD_{600\text{ nm}} \sim 0.8$ (stat). **b**, Total RNA from *C. jejuni* NCTC11168 WT and several transcriptional regulator mutant strains grown to exponential phase ($OD_{600\text{ nm}} \sim 0.5$). Fold changes of sRNA expression relative to WT and normalized to 5S rRNA are indicated. Cjnc230 sRNA was detected with CSO-0537 and Cjnc170 sRNA with CSO-0182. 5S rRNA (CSO-0192) served as loading control.

Comments:

1. Lines 131 and 132: When mentioning RNase III as a regulator of sRNA processing, it would be prudent to cite the McKellar et al. RNase III CLASH paper, which was published back-to-back with the Mediati et al. manuscript.

As suggested, this reference has now been added to the sequence of citations in line 132.

2. It appears that Extended Data Fig. 10 is referenced before Extended Data Fig. 9.

Thank you for pointing this out. This was accidentally switched and is now corrected (line 360).

3. Figure 4b: Given the degree of over-expression of CJnc230 in this experiment, the impact on motility is modest, as the authors state in the manuscript. The experiments have been repeated three times. However, considering that the values for the three data points in these experiments are very close to each other, it could be assumed that these are technical replicates. The legend states three independent replicates, but how this is defined is not clear. Three independent experiments (different days with different cultures) would be preferred. As shown in Fig. 4a, (n=8), I would recommend including more replicates as smaller differences are more convincing if supported by multiple replicates.

The data points in **Manuscript Figure 4b** indeed represent biological replicates from independent cultures of the indicated *C. jejuni* strains. Three independent cultures per strain (biological replicates) were stab-inoculated into three different plates (technical replicates) as specified in the Methods section. For the technical replicates, halo radii were also measured at three different points each and the mean of these values was used. The mean swimming distance with standard deviations presented in the figure is calculated based on the three biological replicates. We have clarified the type of replicate in the legends.

To further support the conclusions, we have now performed additional independent biological replicates with CJnc230 deletion and overexpression mutants, for a total of six biological replicates. These are now included in **Manuscript Figure 6a** and confirm the previously observed phenotype.

4. Figure 4d: To make a statement that you observed a larger portion of flagella with two turns in the OE mutant, it would be necessary to provide quantification of the imaging data. Fig. 4c is convincing and sufficient to make a claim about flagellar length, so perhaps this figure could be removed in case the quantification is challenging.

Manuscript Figure 4d was meant to further support the observation of increased filament length in the CJnc230 overexpression mutant. We agree with the reviewer that quantification would be necessary to base a hypothesis solely on imaging data from confocal microscopy. However, as already anticipated by the referee, unbiased measurements of filament turns were challenging due to uneven fluorescent intensities hampering automated detection. Since the transmission electron microscopy pictures (**Manuscript Fig. 4c**) allowed for more reliable quantification, we based our conclusions mainly on those and only added the flagella-labeling experiments to back up our observation based on additional visualization. We prefer to keep this visualization of our flagella phenotype in the manuscript, but have changed our statement about the number of turns as this has not been quantified (lines 370-372):

“Consistent with electron microscopy, we generally observed that CJnc230 overexpression mutants expressed longer flagella than Δ CJnc230 or WT cells using confocal microscopy (**Fig. 4d**).”

5. Fig. 6a: Again, n=3, and in some cases, the differences were small. This would also benefit from additional replicates. It would have been helpful to include a CJnc230 overexpression positive control in this experiment.

We have now repeated the motility assays in **Manuscript Figure 6a** another three times (independent biological replicates) with all strains, including the CJnc230 deletion and overexpression mutants (as requested by the reviewer). We have added these new data to the previous set of three independent biological replicates. The panel in **Manuscript Figure 6a** has been updated accordingly:

Manuscript Figure 6. RpoN-dependent CJnc230 and FliA-dependent CJnc170 have opposing effects on motility and flagellar biogenesis. a, Motility assays of *C. jejuni* WT and CJnc170 deletion (Δ) or overexpression (OE) mutants with or without deletion or overexpression of CJnc230 in 0.4% soft agar BB plates quantified after 24 hrs of incubation. Δ flaA: non-motile control. Error bars: standard deviation of the mean of independent biological replicates (n = 3 for CJnc230 deletion and overexpression, n = 6 for all other strains). ****: $p < 0.0001$, ns: not significant, two-tailed Student's *t*-test vs. WT.

6. Fig. 6a: The observation that overexpressing CJnc230 in the CJnc170 background does not show an increased motility is surprising. I would have expected that the increase in motility would be more significant compared to just overexpressing CJnc230.

We assume that referee #1 was predicting an even stronger increase in motility upon deletion of CJnc170 in combination with CJnc230 overexpression compared to overexpressing CJnc230 alone, which would be expected based on the WT phenotype of the strain overexpressing both sRNAs. This cannot be conclusively explained at this point. However, we can speculate on several possibilities.

First, both sRNAs might have yet unknown targets that also influence motility. We had proposed that other mRNA targets encoding proteins or regulators might additionally impact flagellar assembly in lines 463-466 of the manuscript. However, deciphering the full targetome of the flagellar sRNAs would be beyond the scope of this study. Second, although we were not able to strongly detect CJnc10 on northern blots (Dugar et al., 2013), this sRNA paralog of CJnc170 could provide some redundancy regarding CJnc170 function. Lastly, flagella

length measurements for sRNA double mutants (**Manuscript Fig. 6b**) revealed that the effect of the two sRNAs on filament length and motility does not always correlate, as overexpressing CJnc230 in combination with any CJnc170 mutation led to longer flagella but WT motility. For this reason, we believe there are yet unknown complexities in the effects of the sRNAs on filament length and motility that might underlie the unexpected phenotype mentioned by the reviewer.

While the mechanism underlying the effects on motility and filament length is so far not clear, we still believe our data support an interplay between the two sRNAs. Nonetheless, we have re-assessed our phrasing regarding related effects of the sRNAs on motility and filament length throughout the manuscript (*e.g.*, line 12, lines 72-74, lines 78-80, lines 377-382, lines 454-455, lines 463-466, lines 577-585, lines 617-618).

Reviewer #2 (Remarks to the Author)

I read with significant interest this manuscript which overall is an elegant work that attempts to dissect the molecular mechanisms of the role two sRNAs might play in the key virulence attribute of flagellation and motility in *C. jejuni*. Overall, the manuscript is very well written and the science is sound; there are several areas primarily in the interpretation of the data that would benefit from further clarification. I look forward to reviewing a revised version of the manuscript following the authors addressing the comments and requested edits below.

We thank referee #2 for the positive feedback and appreciate the useful comments, which helped us to more clearly present the underlying post-transcriptional regulatory pathways of flagellar biogenesis in *C. jejuni*.

Specific comments:

Line 80-81 – After reading the entire manuscript, this statement may be too strong. These sRNAs certainly have the potential to impact a major virulence trait, however, the data does not support a direct strong link as only significant overexpression could demonstrate any changes and no further work associated with colonization or virulence was performed. Please modify accordingly.

As suggested, this sentence in lines 78-80 is now softened: “Due to their impact on the assembly of the flagellar filament and motility, they might ultimately also affect the virulence of this bacterium, as motility is crucial for *C. jejuni* host colonization (Burnham and Hendrixson, 2018).”

Line 171 – Should this say ~89-98 nt for the longer transcripts as 88 is the same as the most abundant?

Yes, this is now changed in line 171 to make it more clear.

Extended data Fig 1a - How would the different downstream genes in 81116 and IA3902 affect the longer length transcripts that are demonstrated? Do you think processing differs in these strains for the 3' end given that you demonstrated the co-transcription with *metB* downstream? Does this affect the activity of the sRNA in those strains of *C. jejuni*?

RNA structure and sequence are well-characterized determinants of nuclease activity in multiple bacteria (Bechhofer and Deutscher, 2019). Thus, the sequence variation downstream of CJnc230 in *C. jejuni* 81116 and IA3902 strains (**Rebuttal Fig. 2**) might potentially affect transcript termination and/or processing at the sRNA 3' end, as pointed out by the reviewer.

and boxes mark the sequence variation detected in *C. jejuni* 81116 and IA3902 due to different sRNA downstream genes in these strains (**Extended Data Fig. 1a**).

However, we do not think that CJnc230 processing and target regulation is different in these two strains compared to NCTC11168 for the following reasons. First, RNase Y in NCTC11168 only had a minor role in CJnc230 maturation/processing and full-length/88-nt CJnc230 looks the same as in the WT (**Manuscript Fig. 2a**). Therefore, the 3'-end processing might not have a huge impact on target regulation, especially as sequence variation starts downstream of the annotated 3' end of CJnc230. In addition, our previous work demonstrated that CJnc230 expression on northern blots looks very similar between NCTC11168 WT and 81116 (Dugar et al., 2013) (see Fig. 4A pasted below from (Dugar et al. 2013)). We anticipate similar processing and expression in strain IA3902, which displays highly similar sequence and genome organization downstream of CJnc230 like 81116 (**Rebuttal Figure 2 and Extended Data Figure 1b**). Nevertheless, evolution of *Campylobacter* sRNA biogenesis and function is definitely an interesting topic (Svensson and Sharma, 2022), which could be further explored in different *Campylobacter* species in the future.

A

Figure 4A and respective legend from (Dugar et al. 2013): Small RNAs in *Campylobacter jejuni*. (Left) Left and right flanking genes and orientation of sRNA candidates in *C. jejuni* NCTC11168. Arrows indicating the genes are not drawn to scale. Small RNA candidates were termed “CjncXXX” and numbered in steps of ten according to their genome position. (Right) Expression analysis of the housekeeping and candidate sRNAs during different growth phases in the four *C. jejuni* strains. Specifically, total RNA (15 µg per lane) was extracted at mid-exponential (E), stationary (S), and overnight (O) growth phase and analyzed by Northern blot using labeled DNA probes complementary to the sRNAs (see Table S14). The blots were probed for the housekeeping 5S rRNA as loading control.

Extended data Fig 5a – Label for Cjnc230 is unclear, please adjust.

To clarify the illustration of the reporter construction in **Extended Data Figure 5a**, we have rephrased the corresponding sentence in the figure legend (lines 1624-1627) explaining the different lengths of promoter-upstream regions that were used.

Extended data Fig 5c – Please clearly state that the labels for each lane are the same as 5b, or relabel again (this would be ideal).

The figure arrangement was changed and the labeling added to **Extended Data Figure 5c**.

Extended data Fig 6 – What timepoint was used for the protein analysis from the growth curve? OD and CFU/mL can differ for *C. jejuni* and affect the evaluation of growth phases. Was CFU/mL collected as well or just OD?

In the growth curve analysis in **Extended Data Figure 6a**, the protein samples for the experiment in panel (b) were taken at exponential phase, which corresponds to an $OD_{600\text{ nm}} \sim 0.4-0.5$, as indicated in the figure legend and the Methods section. We have now also added this information about the time point to the figure legend (lines 1650-1652). Only the absorbance (optical density at 600 nm wavelength, $OD_{600\text{ nm}}$) was measured in **Extended Data Figure 6a**. As the strains showed no obvious growth defect, we reasoned that measuring viability (by plating CFUs) was not necessary.

On the SDS-PAGE image (**Extended Data Fig. 6b**), the PorA loading control suggests the same $OD_{600\text{ nm}}$ has no differences in cell number between the strains. Moreover, for the reviewer's information we now provide colony forming unit (CFU) quantification for CJnc230 mutant strains in exponential phase below (**Rebuttal Fig. 3**). As these did not substantially differ from WT, and the strains showed no defect based on optical density, we have chosen not to include a full growth curve with CFU information added to measure viability/culturability in the manuscript.

Rebuttal Figure 3. Recovery of viable WT and CJnc230 mutants from cultures at comparable optical densities is similar. Colony forming unit (CFU) counts of *C. jejuni* NCTC11168 WT and CJnc230 deletion (Δ), complementation (C), and overexpression (OE) strains plated at exponential phase ($OD_{600\text{ nm}} \sim 0.4-0.5$) on Mueller-Hinton agar plates. Colonies were counted after 48 hrs of incubation in a microaerobic atmosphere (10% CO_2 , 5% O_2 , 85% N_2). Error bars depict the standard deviation (SD) of independent biological replicates ($n = 3$).

Extended data Fig 10 comes before extended data Fig 9 in text, this should be corrected.

Yes, this was accidentally switched and is now corrected (line 360).

Figure 4b and Extended Fig 10 – Please specify here what time point was this data collected (I see later that 24 hrs is listed in the Methods but would be helpful to the reader to have available here). It appears that the 11186 isolate that was used is quite low motility to begin with compared to other papers (ex. ref #30, 87) which also utilized 0.4% agar. Has the laboratory isolate utilized been further characterized? It is well documented in the literature that acquired mutations over time in laboratory strains of *C. jejuni* can affect motility.

As suggested, we have added the information on the incubation time of motility plates, which was 24 hrs, now to the figure legends of **Manuscript Figures 4b and 6a** (line 1501 and line 1543, respectively) and **Extended Data Figure 9** (line 1680). In reference #30 (Le et al., 2015), we could not find any information about which time point was used for motility assays. Furthermore, the bacteria in that study were not stab-inoculated into Brucella broth soft agar plates but spotted onto them, and swarming motility (on the surface) instead of swimming motility (through the agar) appears to have been examined. While in reference #87 (Ruddell et al., 2020; now reference #90) the same incubation time as ours was assayed, highly motile *C. jejuni* strains (IA3902 and W7 variant of NCTC11168) and different growth conditions were used (42°C and Mueller-Hinton soft agar plates for motility assays). These differences might have led to different experimental outcomes compared to the results described in our manuscript. Several previous publications from our lab using the same *C. jejuni* WT strain showed comparable swimming radii after 24 hrs to the data presented in this study, supporting the reproducibility of our protocol with this particular isolate and suggesting its motility phenotype is relatively stable in our lab (Alzheimer et al., 2020; Dugar et al., 2016).

Regarding the isolate characterization: The NCTC11168 isolate we used was originally received from the lab of Arnoud van Vliet (University of Surrey). Besides initial phenotypic characterization of this strain in our lab, including motility (Dugar et al., 2016) and infection (Alzheimer et al., 2020), we also mapped its transcriptome and compared it to other *C. jejuni* strains (Dugar et al., 2013). While *C. jejuni* NCTC11168 is well known to show wide variation between labs (Gaynor et al., 2004; Pascoe et al., 2019), our own genome sequencing data for our isolate revealed only a few single-nucleotide polymorphisms (SNPs) in genes with no obvious links to reduced motility (data not shown). Since we compare the motility and filament phenotypes of our sRNA mutant strains (**Manuscript Figs. 4b and 6a, Extended Data Fig. 9**) to the exact same WT background strain that the deletion and overexpression mutants were constructed in, we think that our conclusions are valid.

Lines 376-381 – This section seems unclear – if *flgM* deletion does not lead to altered flagellar length, how can an argument be made that Cjnc230 mediated repression of *flgM* is the cause of the observed altered flagellar length when Cjnc230 is mutated? Please clarify further.

As stated in these two sentences, there could be other yet unknown molecular mechanisms involved affecting flagellar length in Cjnc230 mutants that compensate for the complete loss of *flgM* in the deletion strain. For example, we propose that there are additional Cjnc230

targets that might play a role in flagellar assembly but are independent of the FlgM repression that our data show.

To clarify, we have rephrased this section: "This indicates that there might be additional mechanisms compensating for the complete loss of FlgM in the $\Delta flgM$ deletion mutant, potentially leading to stronger feedback, that are absent in the CJnc230 overexpression strain, or that additional CJnc230 targets affect flagellum length. Taken together, our data suggest a connection between CJnc230 regulation of *flgM* and major flagellin protein expression and an increase in filament length" (lines 377-382).

Lines 443-449 – Can the authors provide a better explanation for both deletion and overexpression of CJnc170 returning OE CJnc230 to WT levels, especially as flagellar length doesn't change? It looks like there are clearly additional effects of CJnc230 beyond FlgM control that lead to what is seen.

We certainly agree with the reviewer, and proposed that other mRNA targets encoding motility factors or regulators of motility might additionally impact flagellar assembly and/or function in lines 463-466 of the manuscript. Moreover, lines 577-581 also suggested that regulation of flagellar biogenesis may be even more complex than we are currently aware of: for example, additional unidentified feedback control and compensatory mechanisms within the cascade could explain these unexpected results, also with regard to similar filament lengths of $\Delta flgM$ mutants compared to WT bacteria. This complexity is further supported by the lack of flagella- and motility-related phenotypes in previous work on CJnc10 and CJnc170 sRNAs (Le et al., 2015) and the CsrA-FliW partner switch mechanism involved in major flagellin mRNA translation and localization in *C. jejuni* NCTC11168 (Dugar et al., 2016). Our study adds another layer to this complexity, but likely more remains to be discovered.

Also, is line 450-451 correct? Flagellar length was not correlated with the OE CJnc170 mutant in combination with OE and deletion of CJnc230. Maybe this should specify single knockout mutants?

We agree that the motility phenotypes do not fully correlate with the measured flagellar lengths. However, for sRNA single deletion or overexpression mutants and for deletion of CJnc230 combined with deletion or overexpression of CJnc170, the changes in motility did seem to be proportional to the observed filament lengths. For clarity, we have now changed the phrasing "strongly correlate" to "partially correlate" (line 454) and extended the description of flagella length observations in lines 456-462. Our experiments have shown that the relationship between flagella length and swimming speed is not yet clear and remains to be studied in more controlled experimental setups such as single-cell tracking in different conditions, e.g. media viscosities, which we now mention in the Discussion in lines 582-585.

Line 475-478 – Is stating that CJnc230 increases motility and CJnc170 decreases appropriate? This was only observed with significant overexpression of both sRNAs? Perhaps this should be clarified by stating that regulation by these sRNAs has the potential to affect motility with high levels of expression, the conditions in which this naturally occurs are unclear. Have you explored or considered exploring what conditions might lead to this extremely high of overexpression (25-30x) (ie – oxidative stress, temperature changes, iron levels, etc)? Are

there other putative promoters (Fur, CosR, PerR, etc) that might help explain a circumstance where expression might increase?

We have now changed the phrasing in the first paragraph of the Discussion section and pointed out that overexpression of CJnc230 and CJnc170 led to these reported phenotypes in lines 478-484. However, in their native setting, we believe they do not necessarily act to activate or repress motility. We hypothesize that CJnc230 and CJnc170 might be an internal checkpoint or fine-tuning system for the highly regulated flagellar biosynthesis cascade, which only reveals a phenotype upon strong overexpression and uncoupling of the sRNAs from their native regulation. The phenotype might just reflect the disruption of the regulation and not necessarily a real environmental condition. Physiological expression dynamics of the sRNAs, including their promoters, will be further investigated in the future using single-cell approaches to address cellular heterogeneity but are beyond the scope of this study.

As noted above for reviewer #1, we examined CJnc170 and CJnc230 sRNA expression levels under different growth conditions and in selected transcriptional regulator mutant strains by northern blotting (**Rebuttal Figure 1**). However, we observed only very minor changes at different growth or media conditions (**Rebuttal Figure 1a**) or in some of the regulator mutants compared to the effects of *rpoN* or *fliA* deletion (**Rebuttal Figure 1b and Manuscript Figure 1b**). Thus, at the moment we are not aware of any environmental conditions that strongly affect expression of the two sRNAs.

Have you explored how these sRNAs affect *in vivo* measures of virulence either in tissue culture or animal models? Have other studies that have reported altered *in vivo* expression of *C. jejuni* sRNAs reported increased expression of CJnc230 that might help bolster the argument for a strong physiologic role *in vivo*?

Pathogenesis was not the focus of our study, and we believe that the main role of the CJnc230 and CJnc170 sRNAs is a housekeeping function during flagellar biogenesis. However, we cannot rule out that the sRNAs might impact pathogenesis, as motility and flagella are well-defined virulence factors for *C. jejuni* (Burnham and Hendrixson, 2018).

We did inspect publicly available infection transcriptomics data. However, there is little information available on the expression levels of *C. jejuni* sRNAs in *in-vivo* models. In two studies of *C. jejuni* transcriptomes during colonization (chickens and sheep, (Kreuder et al., 2017; Taveirne et al., 2013)) with isolates other than NCTC11168, both sRNAs were detected, but were not significantly regulated compared to *in vitro*.

In unpublished data from our lab, we have performed a dual RNA sequencing experiment of host and pathogen transcriptomes during infection. Upon infection of Caco-2 cells in 2D monolayers and a 3D tissue model (Alzheimer et al., 2020), several sRNAs of *C. jejuni* NCTC11168 were differentially expressed (Alzheimer and König et al., unpublished). Interestingly, CJnc170 and CJnc230 sRNAs were specifically upregulated in internalized bacteria compared to growth in cell culture media alone, which points at their potential importance during infection. Since this is very preliminary data, it requires future follow-up work including validation of the expression levels and virulence phenotypes of selected sRNAs in additional infection studies.

It appears that *C. jejuni* can exhibit normal motility without CJnc230 and while it may fine tune the process as aptly suggested in the manuscript title CJnc230 does not appear to be a critical regulator of this process at least under the conditions studied.

We agree that under the examined conditions, CJnc230 might not be an essential regulator. However, we think just because a strong phenotype is not observed in a specific laboratory condition, this does not exclude that a gene indeed has an important function, especially if it is maintained through evolution. Nevertheless, we have softened our description of the sRNAs as “crucial” or “key” regulators of flagellar biosynthesis (e.g., line 617).

General comments:

NCTC11168 has 2 annotations of *flgE* (Cj1729c and Cj0043) that differ somewhat in sequence –does the affect the analysis at all?

Cj1729c (*flgE*) encodes the structural component of the *C. jejuni* flagellar hook (Lüneberg et al., 1998) and deletion has a strong impact on motility and flagellar assembly (Hendrixson and DiRita, 2003). In contrast, deletion of Cj0043 does not affect flagellar biogenesis and motility, although it is encoded directly downstream of a flagellar hook assembly protein (Cj0042, *flgD*) (Hendrixson and DiRita, 2003). The relevance of Cj0043 for flagellar biogenesis is not yet clear and according to our previous study, there does not seem to be another sRNA candidate encoded in the vicinity of Cj0043 (Dugar et al., 2013). For simplicity, we have decided to exclude this information from the submitted manuscript, also since CJnc170 is only predicted to target one *flgE* gene (Cj1729c), upstream of CJnc230 (Le et al., 2015).

The CJnc230 story is a very elegant explanation of one mechanism of interaction, however, the CJnc170 story is a bit harder to follow and not as well developed, partly because some of this work was done by another lab. Given that a lot of this manuscript focuses on the processing of CJnc230, does CJnc170 require any processing, or is it expressed in active form? It would be good to clarify this for the reader. Where is CJnc170 located on the genome (near other flagellar genes?), what is its length and structure (consider placing in supplemental)? Has CJnc170 also been confirmed to be non-coding via ribosome profiling (ref #33)? There is also a discussion missing on why the previous lab was unable to demonstrate a change in motility (ref#30 states the overexpression mutants were generated and was successful for CJnc170 - was it a failure to overexpress at high enough levels?). This discussion should be added to the paper.

Thank you for pointing this out. CJnc170 sRNA is not encoded adjacent to other flagellar genes and was confirmed to be non-coding by our Ribo-seq data (Froschauer et al., 2022). We have now placed its genomic location and structure in **Extended Data Figure 10** and expanded the information on CJnc170 biogenesis in the Results section in lines 390-396:

“In addition, we also examined levels of the CJnc170 sRNA (**Extended Data Fig. 10**), which was previously shown to be an unprocessed FliA-dependent transcript (Dugar et al., 2013; Porcelli et al., 2013) (**Fig. 1b**) and suggested to control multiple RpoN-dependent genes, such as *flgE* (Cj1729c), *flaB* (Cj1338c), and *flgP* (Cj1026c) (Le et al., 2015). Further, ribosome

profiling of *C. jejuni* strain NCTC11168, the same isolate, grown in rich media confirmed the 47-nt long CJnc170 sRNA is non-coding (Froschauer et al., 2022).”

We also added a section (see below) on the possible reasons why the previous study on CJnc10/CJnc170 (Le et al., 2015) might have missed a motility phenotype to the Discussion. Since the previous work did not detect a phenotype for CJnc170, our results add an important piece of information about the function of this sRNA.

“In *C. jejuni*, paralogous CJnc10 and CJnc170 (FlmR) sRNAs were previously suggested to regulate class II flagellar genes but motility phenotypes associated with the sRNAs were not previously observed (Le et al., 2015). Due to differences in motility assay protocols and lower sRNA overexpression levels compared to our study, a motility phenotype might have been previously missed for FlmR (Le et al., 2015)” (lines 507-512).

PtmG is on the list of potential targets for CJnc230 – your previous work has shown this to be regulated by CJnc190/CJnc180 – do you believe CJnc230 also plays a role in regulation of this gene? This is also flagellar modification gene - how might that affect motility?

CJnc230 is predicted to target several motility-associated genes (**Supplementary Tables 1 and 2**). However, we decided to focus on top-ranked putative targets where interaction was predicted to occur close to the translation initiation region. The *ptmG* mRNA, or CJnc180/190 sRNAs, are not highly significant predicted targets of either CJnc230 or CJnc170. Therefore, we have not further explored regulation of *ptmG* or CJnc180/190 sRNAs by CJnc170 or CJnc230. The *ptmG* gene is in a flagellin glycosylation island that is present in only a subset of strains (Champion et al., 2005). It would be interesting to see in future studies if the conserved sRNAs CJnc230 and CJnc170 regulated an accessory gene, but here we decided to focus on conserved targets.

To answer the question about a role of PtmG on motility: we have previously shown that deletion or overexpression of *ptmG* and deletion of the CJnc180/190 sRNA pair do not affect motility in NCTC11168 (Alzheimer et al., 2020).

Was autoregulation of the *flgE*/CJnc230 transcript by CJnc230 investigated as *flgE* is listed on the IntaRNA list as a potential binding partner?

The *flgE* (Cj1729c) mRNA was previously predicted to be targeted at the translation initiation region by CJnc170 sRNA (Le et al., 2015). As pointed out by the reviewer, we also noticed that CJnc230 sRNA is predicted to interact with the *flgE* (Cj1729c) mRNA (**Supplementary Tables 1 and 2**), suggesting a potential auto-regulation. The predicted interaction site of CJnc230 on *flgE* mRNA is, in contrast to the RBS-overlapping CJnc170 site, approximately 70 nucleotides downstream of the start codon. To test whether a potential interaction might affect RNA stability, we performed rifampicin assays for *flgE* in the absence/presence of CJnc230 overexpression. However, the half-life of *flgE* mRNA was not affected by CJnc230 overexpression (**Rebuttal Fig. 4**). Thus, so far we do not see an effect on *flgE* mRNA stability by CJnc230. However, we cannot rule out that there might be post-transcriptional effects on *flgE* translation, which needs to be explored in future studies using mutational analyses and translational reporters to uncouple *flgE* and CJnc230 expression.

Rebuttal Figure 4. Stability of *flgE* mRNA is not affected by CJnc230. Quantifications of northern blots after rifampicin assays performed in duplicates. Error bars depict the standard error of the mean (SEM) of independent biological replicates (n = 2). Signal intensities of *flgE* mRNA were first normalized to the 5S rRNA signal and time point 0 was then set to 100%. The gray dotted line indicates 50% remaining transcript.

How is regulation of Cj1387c by CJnc230 related to the rest of the story? The authors demonstrate an interaction, but then do not seem to tie it back to the phenotypes of interest nearly as well as for *flgM* and it is not included in the final model in Figure 6? There is only a brief mention of this in the Discussion (lines 537-547). I struggle a bit with the inclusion of that data – on one hand it provides evidence of an additional target that might be tied to the phenotype of interest, but because this connection isn't explored further it detracts somewhat from the rest of the story.

The *flgM* and Cj1387c mRNAs were both among the top ranked predicted targets with a putative interaction with CJnc230 at their ribosome binding sites (**Supplementary Tables 1 and 2**). Both *flgM* and Cj1387c (flagella-flagella interactions (Reuter et al., 2015)) are related to motility. However, as FlgM is a regulator itself, suggesting this might represent an interesting regulatory feature, we chose to focus more of our efforts on this target than Cj1387c, also as the function of Cj1387c is less clear. We validated the interaction with Cj1387c because it has been linked previously both to motility in NCTC11168 and to other cell processes in 81-176 (iron acquisition (Johnson et al., 2016) and methionine biosynthesis (Kelley et al., 2021)), potentially expanding the phenotypic space of CJnc230. We would like to keep the Cj1387c data in the manuscript, as it might be useful for other researchers. However, we have changed the coloring in the in-line probing figures (**Manuscript Fig. 3b and Extended Data Fig. 7a**) to draw the reader's attention more to *flgM*, which was our main focus. If and how Cj1387c regulation contributes to the observed phenotypes, we unfortunately cannot say at the moment, which we mention now in the Discussion (lines 566-570):

“Although the contribution of Cj1387c to the phenotypes presented here is not completely clear, this points to an additional role for FlmE not only in regulating flagellar biosynthesis, but potentially also in important metabolic pathways via repression of Cj1387c, thereby connecting these infection-relevant processes (Crofts et al., 2018; Liu et al., 2018; Ruddell et al., 2020).”

Recent publications have shown a more substantial physiologic role for additional sRNAs (CJnc110, CJnc140) in controlling motility in *C. jejuni* albeit a molecular mechanism for motility control for these sRNAs has not been established. The authors might consider an updated literature review and a brief discussion on the potential for a larger regulatory circuit to control flagellar motility using sRNAs in *C. jejuni* which might be beneficial to the reader. Lines 459-461 in Results hint at this but this is not discussed further in the discussion. Have you looked at screening other sRNAs to look for control via RpoN that could help explain some of the missing data for this regulatory cascade? If so, this could be helpful to report here as well.

We thank referee #2 for the recommendation to mention additional studies on *Campylobacter* sRNAs in our manuscript. As pointed out by the reviewer, CJnc110 and CJnc140 sRNAs were recently suggested to regulate a variety of physiological processes in the highly virulent sheep abortion clone of *C. jejuni* IA3902 (Kreuder et al., 2020; Ruddell et al., 2023). Motility was increased upon deletion of each of the two sRNAs. However, as molecular targets were not identified, the described phenotypes could be in part due to indirect effects. Furthermore, as CJnc110 harbors two validated small open reading frames (sORFs) according to our own ribosome profiling data (Froschauer et al., 2022), its effect on motility could be independent of any base-pairing activity. We have added information on CJnc110 and CJnc140 to the Discussion in lines 514-518:

“*C. jejuni* motility also appears to be subject to environmental control mediated by sRNAs. Deletion of CJnc110 and CJnc140, which are themselves not regulated by flagellar sigma factors (Dugar et al., 2013; Porcelli et al., 2013), increased bacterial swimming behavior of the highly virulent *C. jejuni* sheep abortion clone IA3902, although direct molecular targets were not identified (Kreuder et al., 2020; Ruddell et al., 2023).”

CJnc110 and CJnc140, and possibly also other *C. jejuni* sRNAs, might affect motility in response to environmental signals as observed for several enterobacterial sRNAs (De Lay and Gottesman, 2012; Romilly et al., 2020; Thomason et al., 2012). However, since these two sRNAs do not have RpoN or FliA promoter motifs, unlike CJnc10/170 and CJnc230 (Dugar et al., 2013; Le et al., 2015; Porcelli et al., 2013), they most probably are not direct members of the hierarchical control of flagellar biogenesis and were not investigated in detail in our study on the interplay of two flagellar sRNAs. Furthermore, we had screened our flagellar regulator mutants (including *fliA* and *rpoN*) for differences in expression of several sRNAs annotated previously (Dugar et al., 2013; Porcelli et al., 2013), which is how we became interested in CJnc230. No other of the tested sRNAs showed marked differences in expression other than CJnc230 and CJnc170. However, we cannot rule out that additional unannotated sRNAs, which we did not probe for, might also be regulated in this pathway and feed into the cascade. Future RNA-seq in, e.g., *rpoN* and *fliA* mutants might reveal additional sRNA candidates.

Reviewer #3 (Remarks to the Author)

In this important study, the authors characterize two sRNAs (FlmE and FlmR) that regulate flagellar filament assembly and motility in *C. jejuni*, adding another level of regulation to a pathway that is already tightly controlled. The authors thoroughly study the generation of FlmE from a downstream region to *flgE*. The results of this study document positive effects of FlmE overexpression on flagella synthesis and bacterial motility, while opposite effects are presented for FlmR sRNA overexpression. This interplay between FlmE and FlmR results in an interesting fine tuning of flagella biosynthesis. The manuscript is well-written and easy to read. The experiments are well executed and of high quality, although some of the effects resulted from the manipulations of sRNAs levels are modest. Below are some comments and suggestions that could improve the manuscript:

Major comments:

1. While statistically significant, many of the effects resulted from the sRNAs overexpression or deletions are modest. The authors should better explain why these effects are still of biological significant for the bacteria. This can be done by revision of the text or by additional experiments that will support their importance.

To further support our conclusions and subtle phenotypes, we have added another three independent biological replicates to the motility assays with all presented sRNA mutants. This data is now added to **Manuscript Figure 6a** and agrees with our previous conclusions.

Regarding the question on modest regulation: Bacterial sRNAs are often reported as regulators that fine-tune physiological processes. In many cases this leads to subtle phenotypes and makes overexpression and uncoupling of the respective sRNAs from their native regulation necessary, as also exemplified in a recent study on flagella regulation in the model organism *E. coli* (Melamed et al., 2023). Phenotypes are sometimes also only visible upon double or triple deletion of sRNAs (Goldberger et al., 2021). In line with this, we observed stronger effects on motility and filament assembly upon combining deletion and overexpression of CJnc230 and CJnc170 sRNAs, as shown in **Manuscript Figures 6a and b**, indicating that the interplay of these sRNAs is important for flagella regulation.

Moreover, we think the mild motility phenotype observed upon overexpression of CJnc230 reflects the tight regulation of flagellar assembly and motility, which appears to be hard to interfere with because of possible feedback and compensatory mechanisms (**Manuscript Fig. 5a and Extended Data Fig. 5a**). Such subtle phenotypes could still be relevant under long-term selection as these sRNAs are highly conserved, or could be enhanced in competition with other microbes (as shown for *E. coli* sRNAs in flagellar motility (Melamed et al., 2023)). Moreover, flagellar biogenesis is coupled to cell division in *C. jejuni* (Balaban and Hendrixson, 2011) and cells in batch culture cannot be synchronized. This probably also leads to a dilution of the phenotype, *i.e.* changes in temporal expression/building of the individual components, which requires future single-cell live-cell analyses. In addition, these sRNAs might become more relevant under so far unknown stress or growth conditions. We have expanded the discussion on these points in the revised manuscript at lines 577-602:

“Additional predicted FlmE/FlmR sRNA targets (**Supplementary Tables 1 and 2**) (Le et al., 2015) that could impact flagella/motility functions, including beyond biogenesis (*e.g.*, rotation, chemotaxis), might also be part of the complex flagellar regulatory network, potentially leading to feedback and unpredictable effects. These include for example the FT3SS component FlhB (Cj0335) or RpoN-dependent hypothetical proteins Cj0243c and Cj0428. However, it is challenging to dissect molecularly the effects on flagellar biogenesis and/or function, which remains to be studied in more specific experimental setups to monitor also, *e.g.* swimming speed or directionality, as well as measure these parameters under different conditions or at the single-cell level. Identification of conditions or additional regulators controlling FlmE and FlmR, or additional targets outside of flagella regulation, might point to a role for these sRNAs beyond fine-tuning the biogenesis cascade to modulating flagellar morphology in response to different environments. This would expand the complexity of the system, whose temporal and spatial dynamics in individual bacteria might be disentangled by future studies with single-cell transcriptomics or imaging (Dar et al., 2021; Endesfelder, 2019; Homberger et al., 2022).

Although our combined strong modulation of relative FlmE and FlmR levels overcame potential feedback mechanisms and revealed phenotypes not observed in single deletion mutants before (**Manuscript Fig. 6a, b**), testing under different experimental conditions might increase the relatively subtle effects on motility of sRNA single mutants that we observed, and potentially under conditions that more closely resemble a natural situation. Further, we hypothesize that strong phenotypes arising from sRNA-mediated fine-tuning of the tightly controlled flagellar biogenesis cascade will only be visible in single bacterial cells or in synchronized cultures, in particular as flagella biogenesis is coupled to cell-division in *C. jejuni* (Balaban and Hendrixson 2011). As it is not yet possible to synchronize flagellar assembly in *C. jejuni* via inducible promoters for example, determining the exact temporal modulation of each flagellar building block by the sRNAs awaits future study.”

2. Pages 5-6, lines 126-151: In figure 2A it does not seem that Cjnc230 is affected by PNPase deletion while in Fig 2B we do see an effect in that strain. Can the authors clarify this? The effect of *rny/pnp* double deletion is very strong. Is this the first time they are documented to cooperate?

PNPase deletion does increase levels of the 88 nt species in **Manuscript Figure 2a** (~16-fold) and **Extended Data Figure 1b** (~9-fold) on northern blots, even though its length does not appear to be affected. This increase is likely responsible for the increased 5' end signal observed in primer extension (**Manuscript Fig. 2b**). We used the same RNA samples and the same probe for both northern blotting and primer extension in **Manuscript Figures 2a and b**. As these are different methods, there could be slight differences in the signal from each. As stated in the Results section (lines 126-151), expression of the 88-nt Cjnc230 is not lost in Δpnp , as observed for Δrnc bacteria, but increases its abundance. In combination with Δrny , many longer species are detected. We interpret this as evidence that PNPase is involved in degradation/maturation of the *flgE*-Cjnc230-*metB* transcript in conjunction with RNase Y.

We agree with the reviewer that the effect of double *rny/pnp* deletion is very strong. In Gram-positives lacking RNase E, and it also appears to be the case in Epsilonproteobacteria, RNase Y serves a similar function as RNase E in enterobacteria. It is present in a degradosome-like complex together with PNPase (Cho, 2017; Lehnik-Habrink et al., 2011). These enzymes are well known in Gram-positives to cooperate (Broglia et al., 2020; Salvo et al., 2016; Taggart et

al., 2023), with RNase Y generating decay intermediates that are degraded by PNPase, which we stated in the Discussion at lines 550-552. To our knowledge, this is the first demonstration of this in Epsilonproteobacteria, although RNase Y is not well characterized. Our RNA-seq/term-seq datasets in PNPase and RNase Y mutants, which we provide publicly, should allow future studies on this matter.

3. Page 6, lines 152-166: Can the authors elaborate on the allegedly contradicting results of Fig 2A and 2C?

Manuscript Figures 2a and c are not contradictory to us, but are rather complementary. While **Manuscript Figure 2c** is explained in more detail in lines 168-199 of the submitted manuscript dealing with the 3'-mapping of observed transcripts using term-seq, lines 152-166 describe the location of CJnc230 5' ends in different mutant strains (**Manuscript Fig. 2b**). In lines 152-166, we refer to the different 5' ends of CJnc230 detected in the WT and an RNase III deletion background. Backed by dRNA-seq data displayed in **Extended Data Figure 4**, our results indicate CJnc230 to be a processed sRNA. **Manuscript Figures 2a and c** collectively show that CJnc230 sRNA is dependent on RNase III and further matured at its 3' end by RNase Y and PNPase. **Manuscript Figure 2b** confirms that RNase III cleaves the 5' end of the sRNA.

4. Page 11, lines 314-322: The experiments that were done to support direct base-pairing between CJnc230 were all done *in-vitro*, is there a reason that the authors didn't try the mutagenesis *in-vivo* in the translational reporter assay? Also, can the authors comment on why they have not tried to complement and restore the interaction between the sRNA and the targets? (e.g. testing both M1 mutants together)

In addition to *in-vitro* translation experiments (**Manuscript Fig. 3c**) described in lines 314-322 of the submitted manuscript, we did use *C. jejuni* translational reporter systems also *in vivo* (**Manuscript Fig. 3e**). To show that regulation is specific, we used either negative controls (*flaA* reporter *in vitro*) or mutated versions of CJnc230 *in vivo*. In both cases, we did not observe CJnc230-dependent target regulation.

We did try to mutate the binding regions in the mRNAs to show with compensatory mutations that the interaction with the sRNA was specific. However, since the interaction occurred right at the ribosome binding site of the target mRNAs (**Manuscript Fig. 3a**), this interfered with translation of the respective reporters and was consequently excluded from the manuscript. To support CJnc230-mRNA interactions, in-line probing experiments were also performed reciprocally (**Extended Data Fig. 7a**). We feel that our data, taken together, sufficiently support that a direct base-pairing interaction underlies this regulation.

5. Page 12, lines 344-347: The results presented in Fig 4A are not convincing. It seems that only in 50% of the cases FlaA levels increase upon CJ230 overexpression. Can the authors address it or repeat on the experiment?

Western blotting is oftentimes prone to technical biases as processes like blotting, antibody binding, and the chemiluminescence-based detection reaction can vary between replicates. As the samples represent independent biological replicates, slight differences in media composition and growth of the bacteria might have also contributed to differences in FlaA

protein abundance. However, in all cases the mutants and WT within each independent biological replicate were analyzed on one blot.

We have now re-plotted the same data (**Rebuttal Fig. 5a/same as Manuscript Fig. 4a**) for each of the eight independent biological replicates separately (**Rebuttal Fig. 5b**). All eight independent biological replicates display an increase in FlaA-3xFLAG protein levels in the CJnc230 overexpression mutant, but some show a very strong upregulation (280-340%) compared to others (125-162%) (**Rebuttal Fig. 5b**). Together with the observed increase in *flaA* transcription upon CJnc230 overexpression (**Manuscript Fig. 5**), we think our data is overall conclusive.

Rebuttal Figure 5. CJnc230 overexpression increases major flagellin levels. FlaA-3xFLAG levels in *C. jejuni* NCTC11168 wild-type (WT, black bars), CJnc230 deletion (Δ , gray bars), and overexpression (OE, white bars) strains harvested at exponential growth phase and quantified by western blotting. **a**, Mean of independent biological replicates ($n = 8$) with error bars depicting the standard deviation. **: $p < 0.01$, ns: not significant, two-tailed Student's *t*-test was used to compare the respective mutant to WT. Same data representation as in **Manuscript Figure 4a**. **b**, Eight independent biological replicates (bio- rep) from (**a**) plotted separately with indicated fold changes of FlaA-3xFLAG expression in OE CJnc230 compared to WT.

6. Page 15, lines 444-449: Can the authors explain and discuss how the results of the single and double deletion strains fit the rest of the data presented in the manuscript?

Lines 447 to 453 of the revised manuscript describe the motility assays performed with *C. jejuni* CJnc170 single deletion or overexpression mutants combined with deletion or overexpression of CJnc230 (**Manuscript Fig. 6a**). This experiment showed that while deletion of CJnc230 alone did not cause a significant motility phenotype (**Manuscript Fig. 4b**), its combination with CJnc170 overexpression had a strong effect on motility.

These results, taken together, led us to conclude that CJnc230 and CJnc170 have a balancing effect on motility, which can also be observed in the WT motility phenotype when the sRNAs are overexpressed simultaneously. However, these data also revealed that the sRNAs integrate additional levels of control, as the Δ CJnc170/OE CJnc230 strain did not show even further increased motility (as one might expect). By providing these results at the end of the story, we underline the complexity of the flagellar biogenesis, with post-transcriptional regulators fine-tuning the tightly regulated cascade.

7. The timing is very important in flagella biosynthesis. One may hypothesize that FlmE will be more important earlier in flagella synthesis while FlmR will be important later on when the synthesis is no longer required. Additional experiments testing if FlmE and FlmR are affecting the timing of flagella synthesis can be a nice addition to this work.

We share the reviewer's enthusiasm about investigating the timing of flagella sRNA function within the biogenesis cascade and we agree that FlmE and FlmR are likely involved in temporal regulation of flagellar biogenesis. While we integrate our results using overexpression of CJnc170 (FliA-dependent) and CJnc230 (RpoN-dependent) sRNAs into the previously established transcriptional hierarchy of flagella biosynthesis in *C. jejuni*, examining the timing of flagella synthesis would require live-cell single-cell analysis of flagellar components and the sRNAs, which unfortunately at the moment is technically not feasible for *C. jejuni*. One major challenge is that in *C. jejuni*, flagellar biogenesis is coupled to cell division (Balaban and Hendrixson, 2011) and, thus, cells in batch culture are all in different stages of cell division and flagella assembly. So far, we are not aware of any method to synchronize *C. jejuni* cells in batch culture, which would allow tracking the same step of flagellar biogenesis across all cells. Unlike in *Salmonella* (Guse et al., 2021; Karlinsey et al., 2000), there is no tightly-controllable inducible promoter system available for *C. jejuni* so far, which would allow us to induce the flagellar cascade, e.g., by pulse-expression of a flagellar regulator at the start of the transcriptional cascade. We also consulted with our collaborator Dr. Morgan Beeby (Imperial College London), an expert in *C. jejuni* flagellar biogenesis and function (Drobnič et al., 2023), who confirmed that tackling this question would require non-trivial establishment of new methods for *C. jejuni*, which would be beyond the scope of this study.

8. In this work the authors carried out global 5' end and 3' end mapping by RNA-seq based methodologies. Uploading these data to online genome browsers will be extremely valuable for the scientific community.

We agree with referee #3 and are currently working on the establishment of an online browser for RNA-seq data visualization. While this will take a while and will only be available in the future, we have deposited all sequencing data of this manuscript on the Gene Expression Omnibus platform (NCBI) with accession numbers GSE230835, GSE230836, and GSE230837.

Minor comments:

1. Page 2, lines 46-56: In many gram-negative bacteria (e.g. *E. coli*, *Salmonella*) there is a mixed class 2 + class 3 genes in flagella biosynthesis. Does this mixed class exist in *C. jejuni*? Please refer to it in the text.

Lines 42-44 of the revised manuscript state that two dedicated alternative sigma factors in Epsilonproteobacteria promote transcription of middle and late genes. For clarity, we have now added more info to the respective section (lines 52-54) to distinguish middle and late flagellar genes.

To the best of our knowledge, a mixed class of middle and late flagellar genes is not described for *C. jejuni* so far. However, *flgM* for example, similar to *flgKL* in *E. coli* (Fitzgerald et al.,

2014), is transcribed in an operon from RpoN and FliA promoters (Dugar et al., 2013; Porcelli et al., 2013; Wösten et al., 2004) (lines 603-609).

2. Page 4, lines 106-109: This sentence is unclear, please revise.

We admit that this sentence was slightly confusing. We have changed the phrasing in lines 106-109: “Moreover, in contrast to the strong upregulation of FliA-dependent CJnc170, CJnc230 and *flgE* levels slightly decreased in a mutant lacking the anti- σ^{28} factor FlgM. This uniform downregulation further supports the co-expression of CJnc230 with upstream *flgE*.”

3. I find it confusing that the authors use CJnc230 and CJnc170 names in most places in the text and then transition to FlmE and FlmR, respectively. Consider renaming these two sRNAs FlmE and FlmR from the beginning of the manuscript.

Since most publications referring to *C. jejuni* sRNAs with their identifier prefix “CJnc”, we decided to mainly use these throughout the manuscript. Besides their introduction at the end of the Introduction and Results sections, “FlmE” and “FlmR” are extensively used in the Discussion section. This was intended to not confuse the reader, who might be used to *Campylobacter* sRNA identifiers instead of newly termed abbreviations.

4. As the authors mention in their discussion, a recent paper (<https://doi.org/10.7554/eLife.87151.1>) describes a similar type of fine-tuning of flagella biosynthesis by two sRNAs (MotR and FliX) in *E. coli*. In both cases, one sRNA has a positive role and one sRNA has a negative role. It can be valuable to discuss in more detail the similarities and the differences between the regulation in *E. coli* and *C. jejuni* by these analog sRNAs.

We thank the reviewer for this suggestion. We have now extended the respective section in the Discussion to underline functional similarities between *E. coli* and *C. jejuni* flagella sRNAs in lines 495-499:

“While there is no sequence homology between the *E. coli* and *C. jejuni* flagella sRNAs and there are differences in their targets as well as their biogenesis and hierarchical order of transcription, post-transcriptional fine-tuning of the flagellar biogenesis cascade involving counteracting regulatory RNAs appears to be a functionally conserved concept.”

References

- Alzheimer M, Svensson SL, König F, Schweinlin M, Metzger M, Walles H, Sharma CM. 2020. A three-dimensional intestinal tissue model reveals factors and small regulatory RNAs important for colonization with *Campylobacter jejuni*. *PLoS Pathog* **16**:e1008304. doi:10.1371/journal.ppat.1008304
- Balaban M, Hendrixson DR. 2011. Polar flagellar biosynthesis and a regulator of flagellar number influence spatial parameters of cell division in *Campylobacter jejuni*. *PLoS Pathog* **7**:e1002420. doi:10.1371/journal.ppat.1002420
- Bechhofer DH, Deutscher MP. 2019. Bacterial ribonucleases and their roles in RNA metabolism. *Crit Rev Biochem Mol Biol* **54**:242–300. doi:10.1080/10409238.2019.1651816
- Brogia L, Lécrivain A-L, Renault TT, Hahnke K, Ahmed-Begrich R, Le Rhun A, Charpentier E. 2020. An RNA-seq based comparative approach reveals the transcriptome-wide interplay between 3'-to-5' exoRNases and RNase Y. *Nat Commun* **11**:1587. doi:10.1038/s41467-020-15387-6
- Burnham PM, Hendrixson DR. 2018. *Campylobacter jejuni*: collective components promoting a successful enteric lifestyle. *Nat Rev Microbiol* **16**:551–565. doi:10.1038/s41579-018-0037-9
- Champion OL, Gaunt MW, Gundogdu O, Elmi A, Witney AA, Hinds J, Dorrell N, Wren BW. 2005. Comparative phylogenomics of the food-borne pathogen *Campylobacter jejuni* reveals genetic markers predictive of infection source. *Proc Natl Acad Sci USA* **102**:16043–16048. doi:10.1073/pnas.0503252102
- Cho KH. 2017. The Structure and Function of the Gram-Positive Bacterial RNA Degradosome. *Front Microbiol* **8**:154. doi:10.3389/fmicb.2017.00154
- Corpet F. 1988. Multiple sequence alignment with hierarchical clustering. *Nucleic Acids Res* **16**:10881–10890. doi:10.1093/nar/16.22.10881
- Crofts AA, Poly FM, Ewing CP, Kuroiwa JM, Rimmer JE, Harro C, Sack D, Talaat KR, Porter CK, Gutierrez RL, DeNearing B, Brubaker J, Laird RM, Maue AC, Jaep K, Alcalá A, Tribble DR, Riddle MS, Ramakrishnan A, McCoy AJ, Trent MS. 2018. *Campylobacter jejuni* transcriptional and genetic adaptation during human infection. *Nat Microbiol* **3**:494–502. doi:10.1038/s41564-018-0133-7
- Dar D, Dar N, Cai L, Newman DK. 2021. Spatial transcriptomics of planktonic and sessile bacterial populations at single-cell resolution. *Science* **373**:abi4882. doi:10.1126/science.abi4882
- De Lay N, Gottesman S. 2012. A complex network of small non-coding RNAs regulate motility in *Escherichia coli*. *Mol Microbiol* **86**:524–538. doi:10.1111/j.1365-2958.2012.08209.x
- Drobnič T, Cohen EJ, Alzheimer M, Froschauer K, Svensson S, Singh N, Garg SG, Henderson L, Umrekar T, Nans A, Ribardo D, Hochberg G, Hendrixson DR, Sharma CM, Rosenthal P, Beeby M. 2023. Molecular model of a bacterial flagellar motor *in situ* reveals a “parts-list” of protein adaptations to increase torque. *BioRxiv*. doi:10.1101/2023.09.08.556779
- Dugar G, Herbig A, Förstner KU, Heidrich N, Reinhardt R, Nieselt K, Sharma CM. 2013. High-resolution transcriptome maps reveal strain-specific regulatory features of multiple *Campylobacter jejuni* isolates. *PLoS Genet* **9**:e1003495. doi:10.1371/journal.pgen.1003495
- Dugar G, Svensson SL, Bischler T, Wäldchen S, Reinhardt R, Sauer M, Sharma CM. 2016.

- The CsrA-FliW network controls polar localization of the dual-function flagellin mRNA in *Campylobacter jejuni*. *Nat Commun* **7**:11667. doi:10.1038/ncomms11667
- Endesfelder U. 2019. From single bacterial cell imaging towards *in vivo* single-molecule biochemistry studies. *Essays Biochem* **63**:187–196. doi:10.1042/EBC20190002
- Fitzgerald DM, Bonocora RP, Wade JT. 2014. Comprehensive mapping of the *Escherichia coli* flagellar regulatory network. *PLoS Genet* **10**:e1004649. doi:10.1371/journal.pgen.1004649
- Froschauer K, Svensson SL, Gelhausen R, Fiore E, Kible P, Klaude A, Kucklick M, Fuchs S, Eggenhofer F, Engelmann S, Backofen R, Sharma CM. 2022. Complementary Ribo-seq approaches map the translome and provide a small protein census in the foodborne pathogen *Campylobacter jejuni*. *BioRxiv*. doi:10.1101/2022.11.09.515450
- Gaynor EC, Cawthraw S, Manning G, MacKichan JK, Falkow S, Newell DG. 2004. The genome-sequenced variant of *Campylobacter jejuni* NCTC 11168 and the original clonal clinical isolate differ markedly in colonization, gene expression, and virulence-associated phenotypes. *J Bacteriol* **186**:503–517. doi:10.1128/JB.186.2.503-517.2004
- Goldberger O, Livny J, Bhattacharyya R, Amster-Choder O. 2021. Wisdom of the crowds: A suggested polygenic plan for small-RNA-mediated regulation in bacteria. *iScience* **24**:103096. doi:10.1016/j.isci.2021.103096
- Guse A, Halte M, Hüsing S, Erhardt M. 2021. Hook-basal-body assembly state dictates substrate specificity of the flagellar type-III secretion system. *Mol Microbiol* **116**:1189–1200. doi:10.1111/mmi.14805
- Hendrixson DR, DiRita VJ. 2003. Transcription of sigma54-dependent but not sigma28-dependent flagellar genes in *Campylobacter jejuni* is associated with formation of the flagellar secretory apparatus. *Mol Microbiol* **50**:687–702. doi:10.1046/j.1365-2958.2003.03731.x
- Homberger C, Barquist L, Vogel J. 2022. Ushering in a new era of single-cell transcriptomics in bacteria. *microLife* **3**:uqac020. doi:10.1093/femsml/uqac020
- Johnson JG, Gaddy JA, DiRita VJ. 2016. The PAS Domain-Containing Protein HeuR Regulates Heme Uptake in *Campylobacter jejuni*. *MBio* **7**. doi:10.1128/mBio.01691-16
- Karlinsey JE, Tanaka S, Bettenworth V, Yamaguchi S, Boos W, Aizawa SI, Hughes KT. 2000. Completion of the hook-basal body complex of the *Salmonella typhimurium* flagellum is coupled to FlgM secretion and *fliC* transcription. *Mol Microbiol* **37**:1220–1231. doi:10.1046/j.1365-2958.2000.02081.x
- Kelley BR, Callahan SM, Johnson JG. 2021. Transcription of Cystathionine β -Lyase (MetC) Is Repressed by HeuR in *Campylobacter jejuni*, and Methionine Biosynthesis Facilitates Colonocyte Invasion. *J Bacteriol* **203**:e0016421. doi:10.1128/JB.00164-21
- Kreuder AJ, Ruddell B, Mou K, Hassall A, Zhang Q, Plummer PJ. 2020. Small Noncoding RNA CjNC110 Influences Motility, Autoagglutination, AI-2 Localization, Hydrogen Peroxide Sensitivity, and Chicken Colonization in *Campylobacter jejuni*. *Infect Immun* **88**. doi:10.1128/IAI.00245-20
- Kreuder AJ, Schleining JA, Yaeger M, Zhang Q, Plummer PJ. 2017. RNAseq Reveals Complex Response of *Campylobacter jejuni* to Ovine Bile and *In vivo* Gallbladder Environment. *Front Microbiol* **8**:940. doi:10.3389/fmicb.2017.00940
- Le MT, van Veldhuizen M, Porcelli I, Bongaerts RJ, Gaskin DJH, Pearson BM, van Vliet AHM. 2015. Conservation of σ 28-Dependent Non-Coding RNA Paralogs and Predicted σ 54-Dependent Targets in Thermophilic *Campylobacter* Species. *PLoS ONE* **10**:e0141627. doi:10.1371/journal.pone.0141627
- Lehnik-Habrink M, Newman J, Rothe FM, Solovyova AS, Rodrigues C, Herzberg C,

- Commichau FM, Lewis RJ, Stülke J. 2011. RNase Y in *Bacillus subtilis*: a Natively disordered protein that is the functional equivalent of RNase E from *Escherichia coli*. *J Bacteriol* **193**:5431–5441. doi:10.1128/JB.05500-11
- Liu MM, Boinett CJ, Chan ACK, Parkhill J, Murphy MEP, Gaynor EC. 2018. Investigating the *Campylobacter jejuni* Transcriptional Response to Host Intestinal Extracts Reveals the Involvement of a Widely Conserved Iron Uptake System. *MBio* **9**. doi:10.1128/mBio.01347-18
- Lüneberg E, Glenn-Calvo E, Hartmann M, Bär W, Frosch M. 1998. The central, surface-exposed region of the flagellar hook protein FlgE of *Campylobacter jejuni* shows hypervariability among strains. *J Bacteriol* **180**:3711–3714. doi:10.1128/JB.180.14.3711-3714.1998
- Melamed S, Zhang A, Jarnik M, Mills J, Silverman A, Zhang H, Storz G. 2023. σ 28-dependent small RNA regulation of flagella biosynthesis. *eLife* **12**. doi:10.7554/eLife.87151
- Pascoe B, Williams LK, Calland JK, Meric G, Hitchings MD, Dyer M, Ryder J, Shaw S, Lopes BS, Chintoan-Uta C, Allan E, Vidal A, Fearnley C, Everest P, Pachebat JA, Cogan TA, Stevens MP, Humphrey TJ, Wilkinson TS, Cody AJ, Sheppard SK. 2019. Domestication of *Campylobacter jejuni* NCTC 11168. *Microb Genom* **5**. doi:10.1099/mgen.0.000279
- Porcelli I, Reuter M, Pearson BM, Wilhelm T, van Vliet AHM. 2013. Parallel evolution of genome structure and transcriptional landscape in the Epsilonproteobacteria. *BMC Genomics* **14**:616. doi:10.1186/1471-2164-14-616
- Reuter M, Periago PM, Mulholland F, Brown HL, van Vliet AHM. 2015. A PAS domain-containing regulator controls flagella-flagella interactions in *Campylobacter jejuni*. *Front Microbiol* **6**:770. doi:10.3389/fmicb.2015.00770
- Romilly C, Hoekzema M, Holmqvist E, Wagner EGH. 2020. Small RNAs OmrA and OmrB promote class III flagellar gene expression by inhibiting the synthesis of anti-Sigma factor FlgM. *RNA Biol* **17**:872–880. doi:10.1080/15476286.2020.1733801
- Ruddell B, Hassall A, Moss WN, Sahin O, Plummer PJ, Zhang Q, Kreuder AJ. 2023. Direct interaction of small non-coding RNAs CjNC140 and CjNC110 optimizes expression of key pathogenic phenotypes of *Campylobacter jejuni*. *MBio* e0083323. doi:10.1128/mbio.00833-23
- Ruddell B, Hassall A, Sahin O, Zhang Q, Plummer PJ, Kreuder AJ. 2020. Role of *metAB* in Methionine Metabolism and Optimal Chicken Colonization in *Campylobacter jejuni*. *Infect Immun* **89**. doi:10.1128/IAI.00542-20
- Salvo E, Alabi S, Liu B, Schlessinger A, Bechhofer DH. 2016. Interaction of *Bacillus subtilis* Polynucleotide Phosphorylase and RNase Y: STRUCTURAL MAPPING AND EFFECT ON mRNA TURNOVER. *J Biol Chem* **291**:6655–6663. doi:10.1074/jbc.M115.711044
- Svensson SL, Sharma CM. 2022. Small RNAs that target G-rich sequences are generated by diverse biogenesis pathways in Epsilonproteobacteria. *Mol Microbiol* **117**:215–233. doi:10.1111/mmi.14850
- Taggart JC, Lalanne J-B, Durand S, Braun F, Condon C, Li G-W. 2023. A high-resolution view of RNA endonuclease cleavage in *Bacillus subtilis*. *BioRxiv*. doi:10.1101/2023.03.12.532304
- Taveirne ME, Theriot CM, Livny J, DiRita VJ. 2013. The complete *Campylobacter jejuni* transcriptome during colonization of a natural host determined by RNAseq. *PLoS ONE* **8**:e73586. doi:10.1371/journal.pone.0073586
- Thomason MK, Fontaine F, De Lay N, Storz G. 2012. A small RNA that regulates motility

- and biofilm formation in response to changes in nutrient availability in *Escherichia coli*. *Mol Microbiol* **84**:17–35. doi:10.1111/j.1365-2958.2012.07965.x
- Westermann AJ, Gorski SA, Vogel J. 2012. Dual RNA-seq of pathogen and host. *Nat Rev Microbiol* **10**:618–630. doi:10.1038/nrmicro2852
- Wösten MMSM, Wagenaar JA, van Putten JPM. 2004. The FlgS/FlgR two-component signal transduction system regulates the *fla* regulon in *Campylobacter jejuni*. *J Biol Chem* **279**:16214–16222. doi:10.1074/jbc.M400357200

REVIEWER COMMENTS

Reviewer #1 (Remarks to the Author):

The authors have done a thorough job in addressing my concerns. I have no additional comments.

Reviewer #2 (Remarks to the Author):

I thank the authors for the very thorough response to the initial review and for the extensive additional data that was provided to address the reviewers' comments. This paper represents very extensive, well thought-out work that will have significant impact in the field. I have only a few minor comments for the authors to consider prior to final acceptance.

The rebuttal figure 1 and the information that was generated from this additional work is substantial and represents data that adds value to the manuscript. I have read through the manuscript and supplementary material twice, and do not find a references to this additional work. I believe it would add value to the publication and recommend that it be included as all three reviewers had concerns about the lack of a phenotype except for in OE mutants – readers are likely to have this concern as well. While this work does not answer the question directly of what conditions might lead to overexpression at the level tested, this is important information regarding how tightly tied to the flagellar regulatory cascade expression of these sRNAs seem to be. I realize there may not be a perfect place to introduce, but perhaps a few sentences somewhere in the last two sections of the results... "Even though we were not able to demonstrate in vitro the growth conditions that might lead to the expression levels demonstrated in our OE mutant (extended data ___)... The other option is to introduce in the Discussion around lines 592-600 which were added to address concerns about a lack of a motility phenotype.

In the rebuttal to Reviewer #1 regarding figure 4b, the authors state: "To further support the conclusions, we have now performed additional independent biological replicates with CJnc230 deletion and overexpression mutants, for a total of six biological replicates." However, the figure legend for 6b, which is where the authors state the additional data was placed states "n = 3 for CJnc230 deletion and overexpression, n = 6 for all other strains." Is this legend correct in that the CJnc230 deletion and OE mutants did not have additional replicates performed? In comparing figures between the original version and the revision, the CJnc230 mutant and OE were not original in this 6a figure – does this represent the same data as in Figure 4b, or was it repeated to address reviewer #1's concerns about the CJ230 OE positive control? After looking at this, I am guessing that the replicates in Figure 4b and 6a are additive and equal 6 total, however, nowhere is this data presented together. Assuming it is 3+3 replicates, in the original data (Fig 4b) the WT and mutant look to be around 12mm, with the OE around 14.5mm, while in the new figure 6a WT and mutant are now 13.5mm and OE is maybe 16mm – if all 6 replicates are included (which is now the case for all other strains), is there still statistical significance for the OE?

The addition of wording associated with other sRNAs that have been implicated in motility in *Campylobacter* to the discussion is helpful for readers to put the results of the present study in context to other published knowledge in the field. However, the information provided might be misinterpreted, and I believe could benefit from some additional clarification. The changes state "...Deletion of CJnc110 and CJnc140, which are themselves not regulated by flagellar sigma factors (Dugar et al 2013, Porcelli et al 2013)...". Both Dugar et al 2013 and Porcelli et al 2013 generated TSS data which predicted RpoD promoters for CJnc140 and CJnc110 but did not experimentally confirm this observation; thus without the further information you provided in the rebuttal, changing to "...which themselves are not predicted to be regulated by dedicated flagellar sigma factors..." would seem more appropriate – especially as even in the work presented here you

discovered additional alternative TSS from multiple different sigma factors for CJnc230. Your observation that you screened your flagellar regulator mutants (FliA and RpoN) for differences in expression of other known sRNAs which did not demonstrate differential expression is interesting, particularly as the intergenic region where CJnc110 is located has been experimentally shown to exhibit significant down regulation in an RpoN mutant noted via RNAseq (Chaudhuri et al 2011) – your results seem to contradict this. Would you consider adding this additional information to the supplemental materials (could be referenced briefly in the text around lines 88-90)? This would likely be helpful to other researchers in the field and add strength to the argument that CJnc230 and CJnc170 play a more critical role in the process. Also of possible interest, CJnc110 expression has been shown to be iron activated (Butcher and Stintzi, 2013), which would fit with the authors assertion that “C. jejuni motility also appears to be subject to environmental control mediated by sRNAs” and may thus be worth adding to the reference list.

Reviewer #3 (Remarks to the Author):

The authors have successfully addressed the majority of my comments. The revised text and the additional experimental data strengthen the manuscript. This research elucidates the intricate network governing flagella synthesis through sRNA-mediated regulation, presenting novel insights into cellular mechanisms underlying flagellar synthesis.

Point-to-point reply NCOMMS-23-30341B:

Answers to the reviewers' comments:

Reviewer #1 (Remarks to the Author):

The authors have done a thorough job in addressing my concerns. I have no additional comments.

We thank the reviewer again for the helpful comments on our manuscript, which has strengthened our work, and the support of our revised manuscript.

Reviewer #2 (Remarks to the Author):

I thank the authors for the very thorough response to the initial review and for the extensive additional data that was provided to address the reviewers' comments. This paper represents very extensive, well thought-out work that will have significant impact in the field. I have only a few minor comments for the authors to consider prior to final acceptance.

We thank this reviewer for again closely examining our study and providing additional feedback on the revised version of the manuscript.

The rebuttal figure 1 and the information that was generated from this additional work is substantial and represents data that adds value to the manuscript. I have read through the manuscript and supplementary material twice, and do not find a references to this additional work. I believe it would add value to the publication and recommend that it be included as all three reviewers had concerns about the lack of a phenotype except for in OE mutants – readers are likely to have this concern as well. While this work does not answer the question directly of what conditions might lead to overexpression at the level tested, this is important information regarding how tightly tied to the flagellar regulatory cascade expression of these sRNAs seem to be. I realize there may not be a perfect place to introduce, but perhaps a few sentences somewhere in the last two sections of the results... “Even though we were not able to demonstrate in vitro the growth conditions that might lead to the expression levels demonstrated in our OE mutant (extended data __).... The other option is to introduce in the Discussion around lines 592-600 which were added to address concerns about a lack of a motility phenotype.

As suggested by this reviewer, we have now added the **Rebuttal Figure 1** as a new figure to the Supplementary Information (**Supplementary Fig. 12**). We have also added information on how these additional transcriptional regulator deletion strains were cloned (updated **Supplementary Datasets 6 and 8**). At lines 466-476 of the results section, we have added the following:

“We have assayed steady-state expression levels of the two sRNAs under different growth phases in *C. jejuni* NCTC11168 WT as well as in a panel of non-essential transcriptional regulator deletion strains (**Supplementary Fig. 12a, b**). Both sRNAs were slightly upregulated in lag phase in minimal media (MEM) compared to rich Brucella broth (BB). They showed decreased abundance at stationary phase, which could be rescued by iron supplementation (MEM+Fe) (**Supplementary Fig. 12a**). In contrast to strong control by the flagellar regulators (**Fig. 1b**), neither sRNA was substantially affected by deletion of several transcriptional regulators (**Supplementary Fig. 12b**). Based on their apparently dedicated effects on filament length and bacterial motility, we propose renaming CJnc230/CJnc170 to FlmE/FlmR (flagellar

length and motility enhancer/repressor).”

As suggested by the review, we also refer to this new data in the Discussion (lines 605-608):

“Our preliminary screen under different growth phases and in a panel of regulator mutants has not yet revealed any regulators or conditions that strongly impact sRNA expression (**Supplementary Fig. 12**). However, we cannot rule out that some as of yet unknown conditions affect sRNA levels to modulate motility.”

In the rebuttal to Reviewer #1 regarding figure 4b, the authors state: “To further support the conclusions, we have now performed additional independent biological replicates with CJnc230 deletion and overexpression mutants, for a total of six biological replicates.” However, the figure legend for 6b, which is where the authors state the additional data was placed states “n = 3 for CJnc230 deletion and overexpression, n = 6 for all other strains.” Is this legend correct in that the CJnc230 deletion and OE mutants did not have additional replicates performed? In comparing figures between the original version and the revision, the CJnc230 mutant and OE were not original in this 6a figure – does this represent the same data as in Figure 4b, or was it repeated to address reviewer #1s concerns about the CJ230 OE positive control? After looking at this, I am guessing that the replicates in Figure 4b and 6a are additive and equal 6 total, however, nowhere is this data presented together. Assuming it is 3+3 replicates, in the original data (Fig 4b) the WT and mutant look to be around 12mm, with the OE around 14.5mm, while in the new figure 6a WT and mutant are now 13.5mm and OE is maybe 16mm – if all 6 replicates are included (which is now the case for all other strains), is there still statistical significance for the OE?

As stated in the response to reviewer #1, we have repeated the motility assays with all sRNA single and double mutants, including CJnc230 deletion and overexpression, an additional three times (in total, six independent biological replicates). The data from **Manuscript Figure 4b** was not just re-used for the revised version of **Manuscript Figure 6a**.

To clarify: **Manuscript Figure 4b** has not changed at all, even though this figure was commented on by the reviewer. We did three additional replicates for all relevant strains, but because of variability/batch effects between plates (*i.e.*, even WT was slightly different, although the overall trend for strains was the same), it was hard to include them in **Manuscript Figure 4b**. Therefore, we added the additional replicates to **Manuscript Figure 6a**, as swimming behavior and radii of *C. jejuni* WT and respective mutant strains of the new replicates were more similar to this previous set of independent experiments. This resulted in strongly significant differences for the relevant strains, which are in line with those in **Manuscript Figure 4b** (although in these data, the difference is not as marked). We think our data is sound and overall support the conclusions made, and hope it is now more clear why the new replicates were added to the paper as they were.

The addition of wording associated with other sRNAs that have been implicated in motility in *Campylobacter* to the discussion is helpful for readers to put the results of the present study in context to other published knowledge in the field. However, the information provided might be misinterpreted, and I believe could benefit from some additional clarification. The changes state “...Deletion of CJnc110 and CJnc140, which are themselves not regulated by flagellar sigma factors (Dugar et al 2013, Porcelli et al 2013)...”. Both Dugar et al 2013 and Porcelli et al 2013 generated TSS data which predicted RpoD promoters for CJnc140 and CJnc110 but did not experimentally confirm this observation; thus without the further information you provided in the rebuttal, changing to “...which themselves are not predicted to be regulated by dedicated flagellar sigma factors...” would seem more appropriate – especially as even in the work presented here you discovered additional alternative TSS from multiple different sigma factors for CJnc230.

Thanks for pointing this out. We changed our phrasing in lines 523-525 according to your suggestion:

“Deletion of CJnc110 and CJnc140, which do not appear to be directly regulated by flagellar sigma factors (based on promoter motif predictions),...”

Your observation that you screened your flagellar regulator mutants (FliA and RpoN) for differences in expression of other known sRNAs which did not demonstrate differential expression is interesting, particularly as the intergenic region where CJnc110 is located has been experimentally shown to exhibit significant down regulation in an RpoN mutant noted via RNAseq (Chaudhuri et al 2011) – your results seem to contradict this. Would you consider adding this additional information to the supplemental materials (could be referenced briefly in the text around lines 88-90)? This would likely be helpful to other researchers in the field and add strength to the argument that CJnc230 and CJnc170 play a more critical role in the process.

In order to not cause any potential confusion for the reader, we would like to exclude the information on potentially contradicting results regarding CJnc110 expression from the manuscript. Although the *rpoN* mutant the reviewer mentioned was constructed in the same strain background we used for our study (*C. jejuni* NCTC11168), the growth and media conditions for the experiments were totally different (Chaudhuri et al., 2011). Not only can varying culture conditions affect gene expression levels of bacteria, but also assay sensitivity varies widely between RNA sequencing and northern blotting. In addition, as flagellar deletion strains in *C. jejuni* grow much faster than WT, one has to be careful to not over-interpret expression differences that could be a result of growth phase differences.

Also of possible interest, CJnc110 expression has been shown to be iron activated (Butcher and Stintzi, 2013), which would fit with the authors assertion that “*C. jejuni* motility also appears to be subject to environmental control mediated by sRNAs” and may thus be worth adding to the reference list.

We added the suggested reference (Butcher and Stintzi, 2013) to connect sRNA expression to environmental signals in line 523.

Reviewer #3 (Remarks to the Author):

The authors have successfully addressed the majority of my comments. The revised text and the additional experimental data strengthen the manuscript. This research elucidates the intricate network governing flagella synthesis through sRNA-mediated regulation, presenting novel insights into cellular mechanisms underlying flagellar synthesis.

We thank the reviewer for the feedback, which helped to improve our study, and their support of our revised manuscript.

References

- Butcher J, Stintzi A. 2013. The transcriptional landscape of *Campylobacter jejuni* under iron replete and iron limited growth conditions. *PLoS ONE* **8**:e79475. doi:10.1371/journal.pone.0079475
- Chaudhuri RR, Yu L, Kanji A, Perkins TT, Gardner PP, Choudhary J, Maskell DJ, Grant AJ. 2011. Quantitative RNA-seq analysis of the *Campylobacter jejuni* transcriptome. *Microbiology (Reading, Engl)* **157**:2922–2932. doi:10.1099/mic.0.050278-0